# Relative Entropy Estimation in Function Space:
# Theory and Applications to Trajectory Inference

**Chao Wang** [* 1]   **Luca Nepote** [* 1]
**Giulio Franzese** [1]   **Pietro Michiardi** [1]

## Abstract

Trajectory Inference (TI) seeks to recover latent dynamical processes from snapshot data, where only independent samples from time-indexed marginals are observed. In applications such as single-cell genomics, destructive measurements make path-space laws non-identifiable from finitely many marginals, leaving held-out marginal prediction as the dominant but limited evaluation protocol. We introduce a general framework for estimating the Kullback–Leibler divergence (KL) between probability measures on function space, yielding a tractable, data-driven estimator that is scalable to realistic snapshot datasets. We validate the accuracy of our estimator on a benchmark suite, where the estimated functional KL closely matches the analytic KL. Applying this framework to synthetic and real scRNA-seq datasets, we show that current evaluation metrics often give inconsistent assessments, whereas path-space KL enables a coherent comparison of trajectory inference methods and exposes discrepancies in inferred dynamics, especially in regions with sparse or missing data. These results support functional KL as a principled criterion for evaluating trajectory inference under partial observability. Code available here:
https://github.com/eurecom-probai/functional-kl.

## 1. Introduction

Trajectory inference (TI) is a rapidly evolving field, especially in single-cell genomics applications that seek to reconstruct latent dynamical processes from observed omics data, typically formalized as probability distributions over trajectories. A major challenge is that omics measurements are destructive, providing only snapshots of cellular states at discrete time points rather than continuous trajectories (Trapnell et al., 2014; Nowakowski et al., 2017; Weinreb et al., 2018; La Manno, 2018; Trevino et al., 2021).

This setting has motivated stochastic-process formulations of TI, often borrowing from Optimal Transport (OT) (Peyré & Cuturi, 2019; Villani, 2021) and Schrödinger Bridge (SB) problems (Léonard, 2013; Chen et al., 2021). In this work, we set aside pseudo-time methods (Lange et al., 2022; Haghverdi et al., 2016; Weiler et al., 2024), since pseudo-orderings flatten real temporal structure and can obscure lineage relationships. We focus instead on model families that consider Ordinary Differential Equation (ODE) dynamics (Sha et al., 2024) and/or Stochastic Differential Equation (SDE)-based approaches (Tong et al., 2020; Neklyudov et al., 2023; Chizat et al., 2022), including the SB variants (Shi et al., 2023; Shen et al., 2025; Chen et al., 2023; Park & Lee, 2025). ODE-based methods assume noiseless flows and lead to deterministic optimal control, which under marginal constraints yields Monge-Kantorovich-type formulations (Peyré & Cuturi, 2019; Villani, 2021). SDE-based models relax determinism and solve a stochastic control problem that steers a Brownian prior with minimal control effort, typically optimizing only the drift while treating the marginal flow as fixed or estimated from data. SB-based models are less prescriptive: given a reference diffusion (e.g., Brownian or Ornstein-Uhlenbeck), they infer a path measure whose snapshot marginals match the data while remaining close to the reference, equivalently solving a stochastic optimal control problem (Chen et al., 2021). Yet, computational constraints often push SB methods away from native path-space representations, relying on discretization and finite-dimensional parameterizations that only approximately preserve path-space structure.

In the literature, most TI methods are evaluated via held-out marginals. Such metrics are intrinsically limited because marginals do not determine how states at different times are coupled. Indeed, while marginal evaluation only compares snapshot distributions in finite-dimensional space, the un-

---

[*]Equal contribution [1]Department of Data Science, EURECOM, Biot, France. Correspondence to: Chao Wang <chao.wang@eurecom.fr>, Luca Nepote <luca.nepote@eurecom.fr>.

*Proceedings of the $43^{rd}$ International Conference on Machine Learning*, Seoul, South Korea. PMLR 306, 2026. Copyright 2026 by the author(s).

derlying inference target is a probability measure over full trajectories in function space. Without constraints on temporal correlations, transition structure, or other path-wise properties, distinct trajectory models can attain similar marginal scores while inducing very different distributions over trajectories, differences that remain invisible to marginal-only criteria (see Figure 1). This motivates evaluating the inferred object itself: a probability measure on trajectory space. In this work, we propose a general framework for estimating the KL divergence between probability measures on function space, validate it on special cases where the analytic KL is available, and use it to assess TI methods in both synthetic settings (e.g., physical simulations) and real settings (single-cell omics). This work builds on the growing literature of in-

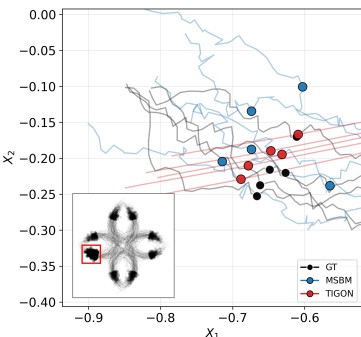

*Figure 1.* **Petal dataset** ($\tau = 0.75$). MSBM and TIGON yield similar $W_2$ scores (0.160 vs. 0.155) but qualitatively distinct trajectories. FKL distinguishes them, assigning significantly lower divergence to MSBM ($\mathrm{KL}(\nu^A \| \nu^B) = 9.641$, $\mathrm{KL}(\nu^B \| \nu^A) = 17.055$) than TIGON (96.144 and 35.505), correctly reflecting MSBM's better alignment with true dynamics.

finite dimensional generative models (Kerrigan et al., 2023; Hagemann et al., 2025; Pidstrigach et al., 2024; Lim et al., 2024; Yang et al., 2025; Baker et al., 2024; Park et al., 2024; Pieper-Sethmacher et al., 2025). In particular, our approach represents trajectory distributions through functional flows using Functional Flow Matching (FFM) (Kerrigan et al., 2024). FFM constructs a path of conditional Gaussian measures interpolating between a reference Gaussian law and a data-induced law; marginalizing these conditional laws over the data distribution yields a corresponding path of unconditional measures. Exploiting this structure, we derive a tractable estimator of the KL divergence between trajectory distributions, expressible, up to model- and approximation-dependent terms, in terms of squared discrepancies between the associated (approximate) velocity fields. The resulting estimator is practical and scales to realistic regimes in both snapshot count and trajectory resolution. Our work connects to recent progress on information-measure estimation (Belghazi et al., 2018; Franzese et al., 2024; Kong et al., 2024; Butakov et al., 2024; 2026; Gowri et al., 2024; Pinski et al., 2015a;b); to our knowledge, it is among the first practically tractable estimators designed to act directly on distributions

over functions.

Our contributions are fourfold. First, we give a general construction for estimating the KL divergence between distributions over functions, based on a careful treatment of absolute continuity and trace-class noise covariances, leading to a simple estimator. Second, we validate our functional KL estimator on an extensive set of special cases with known analytic KL divergence, and systematically assess its robustness across multiple estimation factors. Third, we conduct an extensive evaluation of prominent TI methods on a range of synthetic and real datasets, demonstrating that five widely used snapshot-based metrics can yield inconsistent rankings. Fourth, we show that our functional KL estimator provides a coherent comparison that exposes discrepancies in inferred dynamics, particularly under sparse or missing observations, supporting functional relative entropy as a principled criterion for evaluating TI under partial observability.

## 2. Preliminaries: Functional Flow Matching

We now revisit the FFM (Kerrigan et al., 2024) framework, which is instrumental for the derivation of our method.

Let $H$ be a real separable Hilbert space, equipped with inner product $\langle \cdot, \cdot \rangle_H$ (and, consequently, norm $\| \cdot \|_H$) and denote its Borel $\sigma$-field by $B(H)$. By separability, $H$ has a countable dense subset $\{d_k\}_{k \in \mathbb{N}}$, which allows us to define a countable orthonormal basis (ONB) $\{e_k\}_{k \in \mathbb{N}}$ in $H$. Given $\{e_k\}_{k \in \mathbb{N}}$, the canonical projection map is defined by $\forall K \in \mathbb{N}, \pi^K : H \to \mathbb{R}^K, \pi^K(x) = (\langle x, e_1 \rangle, \ldots, \langle x, e_K \rangle)$, and we write $\mathrm{Cyl}(H \times I)$ for the space of all smooth cylindrical test functions $\varphi : H \times I \to \mathbb{R}, \varphi(x, t) = \psi(\pi^K(x), t)$, where $K \in \mathbb{N}, I = (0, 1)$, and $\psi \in C_c^\infty(\mathbb{R}^K \times I; \mathbb{R})$ is an arbitrary smooth function.

Consider a probability space $(\Omega, \mathcal{F}, P)$ supporting two *independent* random variables $X_0, X_1 : \Omega \to H$, with laws respectively $P_{\#X_1} = \mu_1$ and $P_{\#X_0} = \mu_0$. Assume the $H$−valued random variable $X_0$ to be a Gaussian variable, fully described by its zero mean and its covariance operator $C$. We assume $C$ to be trace-class, self-adjoint, bounded, and strictly positive. We adopt the identification $\mu_0 = \mathcal{N}(0, C)$ when useful and use the notation $H_{\mu_0}$ to indicate the Cameron–Martin space associated with such measure.

We then define an $H$-valued random process $X = (X_t)_{t \in [0,1]}$ by the linear interpolation between the two *independent* random variables $X_0$ and $X_1$:

$$X_t = (1 - t)X_0 + tX_1, \ t \in [0, 1] \tag{1}$$

Denote the associated curve of finite positive Borel probability measures of $X$ by $(\mu_t)_{t \in [0,1]}$ with $\mu_t = P_{\#X_t}$. We say that the velocity field $v : H \times [0, 1] \to H$ *generates*

the path of measures $(\mu_t)_{t\in[0,1]}$ if the path $\mu_t$ is the push-forward of $\mu_0$ along the flow associated with $(v_t)_{t\in[0,1]}$, where flow refers to the mapping $\phi : [0,1] \times H \to H$ that solves *the initial value problem* (Kerrigan et al., 2024): $\partial_t\phi_t(x) = v_t(\phi_t(x))$, $\phi_0(x) = x$. A standard way to verify this is to check that the pair $(\mu_t, v_t)$ satisfies *the continuity equation in the sense of distributions (i.e., weakly)* (Zhang & Scott, 2025; Kerrigan et al., 2024): for arbitrary test function $\varphi \in \mathrm{Cyl}(H \times I)$,

$$
\int_H \varphi(x,1)d\mu_1(x) - \int_H \varphi(x,0)d\mu_0(x) \\
= \int_I \int_H \left( \partial_t\varphi(x,t) + \langle v_t(x), \nabla_x\varphi(x,t) \rangle \right) d\mu_t(x)dt
\tag{2}
$$

where $\nabla_x\varphi(x,t)$ is the unique vector $g$ such that the Fréchet derivative $D_x\varphi(\cdot,t)$ can be identified with $D_x\varphi(\cdot,t) = \langle \cdot, g \rangle$ by the Riesz representation theorem.

If known, the velocity field could be leveraged for generative modeling purposes (simulating an $H-$valued ODE). Since $\mu_1$ and $v$ are unknown in practice, the velocity field is approximated with neural networks optimized using the conditional flow-matching loss (Kerrigan et al., 2024).

## 3. Derivation of Functional KL

**To begin with, we clarify the probabilistic setting and state our objective.**

We extend the previous construction to accommodate *two different* probability spaces $(\Omega, \mathcal{F}, P^A)$ and $(\Omega, \mathcal{F}, P^B)$, both supporting independent random variables $X_0$ and $X_1$, such that $X_0$ has the same Gaussian law in both spaces whereas $X_1$ has laws $\mu_1^A = \nu^A$ and $\mu_1^B = \nu^B$ respectively. We then train one FFM model for each endpoint law, obtaining two parametrized pairs $(\mu_t^A, v_t^A)$ and $(\mu_t^B, v_t^B)$, each satisfying the weak continuity equation. We overload the notation previously introduced by considering superscripts to indicate whether the measures and fields of interest refer to the first or second probability space.

Our objective in this work is to estimate the KL divergence between two probability measures $\nu^A$ and $\nu^B$ on the Hilbert space $H$, which is formally defined as

$$
\mathrm{KL}\left(\nu^A \,\|\, \nu^B\right) := \begin{cases} \int_H \log\left(\frac{d\nu^A}{d\nu^B}\right) \, d\nu^A, & \text{if } \nu^A \ll \nu^B, \\ +\infty, & \text{otherwise.} \end{cases}
\tag{3}
$$

**Next, we state the assumptions needed to construct our framework.**

**Assumption 3.1** (Existence of Radon–Nikodym derivative). We assume that $\nu^A \ll \nu^B$, so that the Radon–Nikodym derivative $\frac{d\nu^A}{d\nu^B}$ exists.

*Remark.* Indeed, as the goal of this work is to obtain KL estimates via a novel estimator, we need to assume that the underlying *estimand* (the true KL) is well-defined.

**Assumption 3.2** (Cameron–Martin support). The measures are fully supported on the Cameron–Martin space of the noise ($\nu^A(H_{\mu_0}) = \nu^B(H_{\mu_0}) = 1$).

*Remarks.* First, Assumption 3.2 can be relaxed: together with Assumption 3.1, it suffices to require that $\nu^B$ be fully supported on $H_{\mu_0}$, since $\nu^B(H_{\mu_0}) = 1$ paired with Assumption 3.1 implies $\nu^A(H_{\mu_0}) = 1$. Indeed, by Assumption 3.1, $\nu^A$ is absolutely continuous with respect to $\nu^B$ on $B(H)$, i.e., $\nu^B(\Gamma) = 0$ implies $\nu^A(\Gamma) = 0$ for all $\Gamma \in B(H)$. If $\nu^B(H_{\mu_0}) = 1$, then $\nu^B(H \setminus H_{\mu_0}) = 0$, and since the Hilbert subspace $H_{\mu_0}$ is Borel, Assumption 3.1 yields $\nu^A(H \setminus H_{\mu_0}) = 0$, hence $\nu^A(H_{\mu_0}) = 1$.

Second, Assumption 3.2 can be satisfied by choosing noise that is rougher than the data distribution (so the data are fully supported on $H_{\mu_0}$) while still being trace-class (and hence in $H$). In practice, in $H := L^2(\mathbb{T}; \mathbb{R}^d)$ with Fourier orthonormal basis, we first estimate the data's Fourier spectrum using empirically determined variances for each Fourier coefficient, and then multiply each coefficient by the wavenumber magnitude $k = \|m\|_2$ to produce noise with a rougher spectrum while still in $H$ as discussed in (Chen & Vanden-Eijnden, 2025).

**Under these assumptions, we now develop a velocity-only representation of the KL divergence** $\mathrm{KL}\left(\nu^A \,\|\, \nu^B\right)$. The derivation is organized into three lemmas: Lemma 3.3 establishes the absolute continuity needed to ensure the Radon–Nikodym derivatives are well-defined; Lemma 3.4 applies weak continuity equation to express KL divergence with logarithmic Radon–Nikodym derivative and velocities. Lemma 3.5 links the logarithmic gradients to the velocity fields.

**Lemma 3.3.** *Under the Assumptions 3.1 and 3.2,*

(a) *$\mu_t^{A,B} \ll \rho_t$, where $\rho_t = ((1-t)\mathrm{Id})_\# \mu_0$ denotes the push-forward of $\mu_0$ by the map $(1-t)\mathrm{Id}$ for every $t \in [0,1)$.*

(b) *$\mu_t^A \ll \mu_t^B$ for every $t \in [0,1]$.*

*Proof.* The proof follows from a conditional Gaussian representation of $\mu_t^{A,B}$ and the Cameron-Martin theorem. See Appendix A for the full proof. $\square$

Given the Radon–Nikodym derivatives are well-defined, next we apply the weak continuity equation using the Radon–Nikodym derivative and its logarithm as test functions to derive an integral representation of the KL divergence.

**Lemma 3.4.** *Let $r_t := \frac{d\mu_t^A}{d\mu_t^B}$, which is a well-defined Radon-Nikodym derivative by Lemma 3.3 (b). Then, under mild*

*regularity conditions, we have that* $\mathrm{KL}\big(\nu^A \parallel \nu^B\big)$ *equals*

$$\int_I \int_H \big\langle v_t^A(x) - v_t^B(x), \, \nabla_x \log r_t(x) \big\rangle \, \mathrm{d}\mu_t^A(x). \quad (4)$$

*Proof.* The proof proceeds in two steps. First, $r_t$ and $\log r_t$ are shown to be admissible test functions, using Doob's martingale convergence theorem for the projected Radon–Nikodym derivatives and the density of $C_c^\infty$ functions in the $L^1$ spaces associated with finite-dimensional projections. Second, the weak continuity equation is tested with $\log r_t$ for $\big(\mu_t^A, v_t^A\big)$ and with $r_t$ for $\big(\mu_t^B, v_t^B\big)$, then combining the resulting identities with $\mu_t^A = r_t \mu_t^B$ gives the desired KL identity. See Appendix A for the full proof. □

By Lemma 3.3 $(a)$, we can rewrite the term $\nabla_x \log r_t(x)$ appearing in Lemma 3.4 as

$$\nabla_x \log r_t(x) = \nabla_x \log \frac{d\mu_t^A}{d\rho_t}(x) - \nabla_x \log \frac{d\mu_t^B}{d\rho_t}. \quad (5)$$

The next lemma links this logarithmic gradients mismatch to velocity fields mismatch, which is useful to derive a velocity-only representation of $\mathrm{KL}\big(\nu^A \parallel \nu^B\big)$.

**Lemma 3.5.** *For the linear interpolation used in FFM (Kerrigan et al., 2024), under the assumption that $v_t^A(x) - v_t^B(x) \in \mathrm{Dom}\,\big(C^{-1}\big) = \mathrm{Range}(C)$   $\mu_t^A(dx)\,dt$-a.e., we have*

$$\begin{aligned}
&\nabla_x \log \frac{d\mu_t^A}{d\rho_t}(x) - \nabla_x \log \frac{d\mu_t^B}{d\rho_t} \\
&= \frac{t}{1-t} C^{-1} \big( v_t^A(x) - v_t^B(x) \big)
\end{aligned} \quad (6)$$

*Proof.* The proof first uses the Gaussian-mixture representation of $\mu_t^{A,B}$, the Cameron-Martin theorem on translated Gaussian measures, and Bayes' rule to identify

$$\nabla_x \log \frac{d\mu_t^{A,B}}{d\rho_t}(x) = \frac{t}{(1-t)^2} C^{-1} \mathbb{E}_{P^{A,B}} \left[ X_1 \mid X_t = x \right]$$

and then combines this with the velocity formula given by linear interpolation in FFM

$$v_t^{A,B}(x) = \frac{\mathbb{E}_{P^{A,B}} \left[ X_1 \mid X_t = x \right] - x}{1-t}$$

to obtain the desired relation between score mismatch and velocity mismatch. See Appendix A for the full proof. □

Finally, with $\nabla_x \log r_t$ in Lemma 3.4 replaced by velocity fields mismatch in Lemma 3.5, we obtain a velocity-only expression for what we label the Function-space Kullback–Leibler divergence (FKL):

**Theorem 3.6.** *For the linear interpolation used in FFM (Kerrigan et al., 2024), under Assumptions 3.1–3.2 and mild regularity conditions, we have that $\mathrm{KL}\big(\nu^A \parallel \nu^B\big)$ equals*

$$\int_0^1 \int_H \frac{t}{1-t} \left\| v_t^A(x) - v_t^B(x) \right\|_{H_{\mu_0}}^2 \, d\mu_t^A(x)\, dt. \quad (7)$$

In the sequel, we use Equation (7) for FFM-based FKL estimation. To this end, we use a MINO-based conditional neural operator (Shi et al., 2025) to jointly approximate the two velocity fields $v^A$ and $v^B$, where a binary conditioning variable $c$ specifies which velocity field is being approximated. To cancel the singularity from $\frac{t}{1-t}$ in Equation (7) and thereby stabilize stable FKL estimation, we enforce the boundary condition $v_\theta(x, 1) = x$ through a subtraction-based parameterization (Hu et al., 2025), so that the difference between the two learned fields vanishes at $t = 1$. The model is trained with the conditional flow-matching objective, and the resulting FKL is estimated by Monte Carlo evaluation of Equation (7) from samples of $\nu^A$, $\mu_0$, and $t \sim \mathcal{U}[0, 1]$, without requiring simulating the full generative dynamics via ODE integration. Full implementation details of FKL estimation are provided in Appendix B.

## 4. Validation of Function KL Estimation on Special Cases

To validate our theory and KL estimation pipeline, we conduct an extensive comparison between analytic and estimated KL divergences in two special cases where the analytic KL is available. This enables a direct evaluation of the estimation accuracy. Specifically, we consider *(i) Gaussian measures:* $\nu^A = \mathcal{N}\big(s \sin(2\pi f x), \mathcal{R}\big)$, $\nu^B = \mathcal{N}(0, \mathcal{R})$, where $\mathcal{R}$ is a Matérn covariance operator with smoothness parameter $\alpha_1 = 3.5$, and we vary the codomain dimension $D$, frequency $f$, and scale $s$; and *(ii) linear SDEs:* $\nu^{A,B}$ : $dY_t = c^{A,B} Y_t \, dt + g \, dW_t$, where we vary the codomain dimension $D$, drift $c^{A,B}$, and diffusion $g$. Derivations of the closed-form ground truth KL for both cases are provided in Appendix C; all KL estimates are computed with both the training and estimation resolutions of FFM set to 256. For the centered noise measure $\mu_0$ used in FKL estimation, we choose the covariance operator $C$ as follows: in the *(i) Gaussian special case*, we use Matérn with smoothness $\alpha_0 = 0.5$; in the *(ii) SDE special case* and in all experiments in the next section, we estimate the data's Fourier spectrum using empirically determined variances for each Fourier coefficient, then scale each coefficient by the wavenumber magnitude $k = \|m\|_2$ to obtain a rougher noise spectrum. As shown in Table 1, across a range of settings, our estimated KL matches the closed-form ground truth very closely. This provides compelling empirical validation that our functional-space KL formulation and its mathematical derivation are correct, and that the proposed estimation procedure is reliable in practice.

We further ablate some important factors, discussed next.

**Sensitivity to estimation factors.** To evaluate the robustness of FKL estimation via Equation (7) to four key estimation factors, we conduct a systematic ablation study in a Gaussian special case (Case 3 in Table 1 (a)), where

$\text{KL}_{\text{Fwd}} = \text{KL}_{\text{Rev}} = 50.00$, and an analytic expression for the marginal velocity mismatch is also available (see Appendix C).

*(i) Basis truncation.* We discretize the Gaussian processes on $M = 128$ uniformly spaced time points and implement the method using Fourier representations with $8, 16, 32,$ and $64$ modes. Figure 3a shows that the estimation error decreases as more modes are retained. *(ii) Velocity-field estimation.* Although the analytic marginal velocity field is unavailable in general, the FFM conditional flow-matching loss provides a surrogate measure of velocity estimation quality and quickly converges to a very small value during training. In the Gaussian special case, with the analytic marginal velocity field being available, Figure 3b shows that the marginal velocity-mismatch error is uniformly small over $t \in [0,1]$, decreases as $t$ grows, and vanishes at $t = 1$ due to the boundary parameterization in Equation (21). This supports accurate FKL estimation, since Equation (7) emphasizes the regime near $t = 1$ through $\frac{t}{1-t}$, where the velocity-estimation error is smallest. *(iii) Monte Carlo approximation.* We vary the number of trajectories in the KL estimator from 10 to 2000. Figure 3c shows that the estimates remain highly precise and stable throughout. *(iv) Temporal discretization.* We vary the number $n$ of sampled time points used to approximate the integral in Equation (7). Over 5 random seeds, Figure 3d shows that the mean estimate remains close to the ground truth for all $n$, and the standard deviation decreases substantially with finer discretization and becomes negligible at $n = 80$, confirming the strong stability of our estimator.

Together, these results show that FKL estimation is accurate and numerically stable, supporting our discretization choices for FKL estimation.

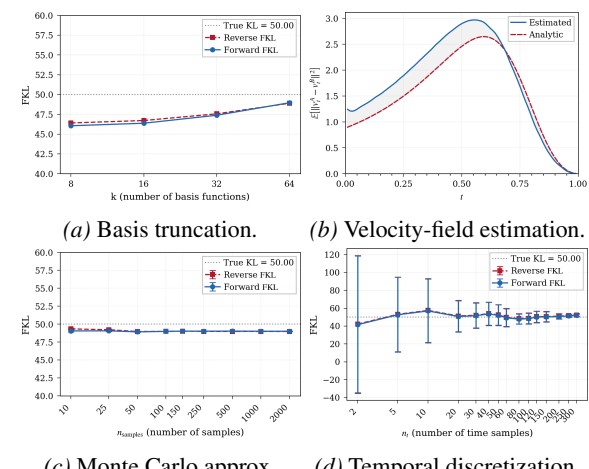

*(a)* Basis truncation.   *(b)* Velocity-field estimation.

*(c)* Monte Carlo approx.   *(d)* Temporal discretization.

*Figure 3.* Sensitivity analysis on a Gaussian special case.

**Choice of noise covariance operator $C$.** Accurate and resolution-invariant FKL estimation requires $C$ to be *(i)* trace-class on $H$ and to *(ii)* satisfy the Cameron–Martin support assumption (Assumption 3.2), i.e., the noise $\mathcal{N}(0, C)$ must be rougher than the data measure. An ablation on the Gaussian special case (Case 2 in Table 1 (a) with data smoothness $\alpha_1 = 3.5$) shows that white noise ($C = \text{Id}$) violates trace-class, causing the FKL to diverge as test resolution increases, while an overly smooth Matérn noise ($\alpha_0 = 6.0 > \alpha_1$) violates the support condition and yields unstable estimates; in contrast, Matérn $C$ with $\alpha_0 = 0.5 < \alpha_1$ satisfies both conditions, yielding accurate and resolution-invariant forward and reverse FKL estimates, which highlights the importance of choosing a noise covariance that meets both conditions for FKL evaluation. Details in Appendix D.1.

# 5. Application of Function KL on TI Evaluation

## 5.1. TI Benchmark Setup

**TI methods.** We consider several recent SDE-based methods. vanilla IPF-based baseline (vSB) (Chen et al., 2022), Schrödinger Bridge Iterative Reference Refinement (SBIRR) (Shen et al., 2025), and Multi-Marginal Schrödinger Bridge Matching (MSBM) (Park & Lee, 2025) formulate TI as a constrained stochastic optimal control problem, which via the Hopf-Cole transform reduces to

| Case | $D$ | $f$ | $s$ | $\text{KL}(\nu^A \| \nu^B)$ | | $\text{KL}(\nu^B \| \nu^A)$ | |
|------|-----|-----|-----|----------|-----------|----------|-----------|
| | | | | Analytic | Estimated | Analytic | Estimated |
| 1 | 1 | 1 | 0.5 | 3.64 | 3.95 | 3.64 | 3.88 |
| 2 | 1 | 1 | 1.5 | 32.79 | 32.61 | 32.79 | 33.31 |
| 3 | 1 | 3 | 1.5 | 50.00 | 49.58 | 50.00 | 49.00 |
| 4 | 1 | 5 | 1.5 | 103.74 | 104.46 | 103.74 | 100.58 |
| 5 | 2 | 1 | 0.5 | 7.29 | 6.77 | 7.29 | 6.72 |
| 6 | 3 | 1 | 0.5 | 10.93 | 10.48 | 10.93 | 10.49 |
| 7 | 5 | 1 | 0.5 | 18.22 | 22.32 | 18.22 | 22.28 |
| 8 | 10 | 1 | 0.5 | 36.43 | 45.12 | 36.43 | 45.73 |

*(a)* Gaussian case.

| Case | $D$ | $c_{\mathcal{A}}$ | $c_{\mathcal{B}}$ | $g$ | $\text{KL}(\nu^A \| \nu^B)$ | | $\text{KL}(\nu^B \| \nu^A)$ | |
|------|-----|------|------|-----|----------|-----------|----------|-----------|
| | | | | | Analytic | Estimated | Analytic | Estimated |
| 1 | 1 | 0.01 | 1.50 | 0.75 | 8.93 | 8.87 | 54.71 | 53.77 |
| 2 | 1 | 0.10 | 2.00 | 0.75 | 15.89 | 13.89 | 186.19 | 200.64 |
| 3 | 2 | 0.01 | 1.50 | 0.75 | 17.86 | 18.05 | 109.43 | 127.67 |
| 4 | 3 | 0.01 | 1.50 | 0.75 | 26.79 | 34.58 | 164.14 | 145.06 |
| 5 | 5 | 0.01 | 1.50 | 1.00 | 26.34 | 32.82 | 158.22 | 135.28 |

*(b)* SDE case.

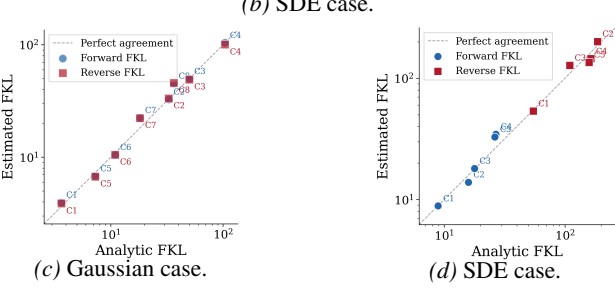

*(c)* Gaussian case.   *(d)* SDE case.

*Table 1.* **Comparison of analytic and estimated FKL.** Tables report numerical values; plots compare analytic vs estimated values.

learning forward and backward diffusion processes. Mean-Field Langevin in path space (MFL) (Chizat et al., 2022) relaxes the multi-marginal fitting constraints and solves the resulting entropy-regularized variational problem using mean-field Langevin dynamics. entropic Action Matching (AM) fixes the marginal densities and therefore optimizes only one variable. As an ODE-based baseline, we include Trajectory Inference with Growth via Optimal transport and Neural network (TIGON), which formulates TI as a deterministic optimal control problem governed by the continuity equation with multi-marginal constraints.

For each method, we sample trajectories after training. SBIRR and vSB simulate trajectory segments independently between consecutive training snapshots and concatenate them into full trajectories. TIGON, MSBM, and AM sample an initial state at an endpoint and then simulate the trajectory end-to-end. MFL connects optimized snapshots at training time points using conditional reference Brownian bridges. We denote by Validation (VAL) the validation trajectories resampled from the same distribution as the ground-truth trajectories.

**Datasets.** We use both synthetic and real single-cell datasets. The synthetic datasets, generated from known SDEs, include *(i) Lotka-Volterra*, modeling predator–prey dynamics; *(ii) Repressilator* (Shen et al., 2025), exhibiting cyclic behavior; and *(iii) Petal* (Neklyudov et al., 2023), capturing bifurcations and merges similar to those in cellular differentiation. For *Lotka–Volterra* and *Repressilator*, odd-indexed snapshots are used for training and even-indexed snapshots for validation. In contrast, for *Petal*, all snapshot times are used for both training and validation, but the validation set consists of samples different from those used for training. Real data consists of four scRNA-seq benchmarks, preprocessed following (Shen et al., 2025): *(i) Embryoid Body (EB)* (Moon et al., 2019), *(ii) Human Embryonic Stem Cell (hESC)* (Chu et al., 2016), *(iii) Mouse Erythroid (ME)* (Pijuan-Sala et al., 2019), and *(iv) Human Fibroblast (HF)* (Riba et al., 2022). Each dataset is projected to 5 dimensions and split into 3 equally spaced training snapshots and 2 validation snapshots.

**Evaluation metrics.** The TI literature typically evaluates performance via marginal reconstruction on held-out validation snapshots, using metrics such as Earth Mover's Distance (EMD) ($W_1$), Wasserstein-2 Distance (W2), Sliced Wasserstein Distance (SWD), Max-Sliced Wasserstein Distance (MWD), and RBF-Maximum Mean Discrepancy (MMD) (see Appendix E.1), all of which we report here. To the best of our knowledge, this is the first benchmark using such an extensive set of standard marginal metrics.

However, marginal metrics cannot detect discrepancies at the level of full trajectory distributions. We therefore estimate FKL between the ground-truth trajectory distribution and that produced by each method by learning velocity fields from complete trajectories. For synthetic datasets, ground-truth trajectories are simulated from the known SDEs, whereas for real scRNA-seq datasets, where only snapshot samples are observed, we use the trajectory measure inferred by SBIRR (Shen et al., 2025) from all snapshots as a biologically motivated reference. This is consistent with standard assumptions in cellular development modeling, since SBIRR learns stochastic dynamics aligned with Waddington's landscape view of differentiation; under this proxy, FKL quantifies how closely alternative methods match a biologically interpretable view of developmental dynamics.

### 5.2. Results with Marginal-Based Metrics

**Finding 1: Marginal-based rankings vary substantially with *(i)* validation time-point, *(ii)* metric choice, and *(iii)* metric hyperparameters.** Our benchmark highlights that inconsistencies are intrinsic to marginal-based evaluation. *(i) The same method can rank best at one validation snapshot and substantially worse at another, even though methods are trained to learn the full underlying dynamics.* This pattern is clear on both synthetic and real datasets. On Lotka-Volterra, SBIRR leads at early snapshots but is overtaken by vSB and TIGON at later ones. Repressilator shows even stronger reversals, with SBIRR leading at some snapshots and TIGON at others. Petal is more stable, but still not invariant across $\tau$ (Table 2). The same effect appears on real datasets, where the leading method at early snapshot often differs from that at later snapshot (Table 3). *(ii) Even at a fixed validation snapshot, different metrics on the same marginal can induce different rankings.* For example, at the last validation snapshot of Lotka-Volterra, TIGON ranks best under EMD and MMD, but not under the other metrics (Table 4a). Similar metric-dependent reversals are also observed at early snapshots on Petal, EB, and ME. *(iii) Some marginal metrics are themselves sensitive to hyperparameter choices, so rankings can vary even within a fixed metric family.* Kernel-based distances illustrate this effect: their usefulness depends on whether the kernel captures a notion of similarity aligned with the data. For MMD with RBF kernel, the statistic is informative only when Euclidean distances in data space are meaningful, and even then it can remain highly sensitive to bandwidth. With a fixed bandwidth, MMD may saturate and become insensitive to growing discrepancies. On Repressilator, this appears for AM: while OT-based metrics such as EMD and W2 increase steadily, reflecting transport over larger spatial scales, MMD remains nearly constant (Table 4b). Taken together, these results suggest that marginal-based rankings are highly sensitive to evaluation choices, making them difficult to interpret robustly.

**Finding 2: Matching marginals does not imply correct dynamics.** Marginal evaluation is fundamentally non-identifiable: many distinct path measures can share the same finite set of time-indexed marginals, so matching these marginals does not determine the underlying dynamics. Therefore, *(i) strong marginal agreement can mask incorrect temporal behavior.* A model may achieve competitive marginal scores without recovering the true dynamics. On Repressilator, for example, TIGON attains competitive marginal distances despite generating a smooth spiral rather than the characteristic three-gene oscillations (Table 4b). *(ii) Marginal comparisons are further affected by method-specific trajectory generation procedures.* Choices such as the numerical integration direction can improve agreement at some validation snapshots while degrading it at others. For example, on Lotka-Volterra, TIGON with backward integration performs well at the last validation snapshot (Table 4a). On Repressilator, SBIRR performs well at early snapshots but deteriorates later, whereas vSB scores well at the last snapshot despite missing the correct dynamics (Table 4b).

To further assess both the statistical robustness of marginal-metric rankings and their relation to dynamical fidelity, we examine the *Critical Difference (CD)* plots on the three synthetic datasets (Figure 5). On Lotka-Volterra and Repressilator, the CD plots show that, except for VAL, most differences are not statistically significant, underscoring the statistical weakness of marginal-metric rankings. On Petal, several differences become significant, yet CD ranks SBIRR ahead of VAL, suggesting that even statistically significant marginal-based rankings need not align with trajectory-level fidelity. Overall, these results *support our observations*: marginal-based metrics often yield rankings that are not only statistically indistinguishable but also misaligned with trajectory-level fidelity. As a result, the fact that no method consistently recovers the correct dynamics is largely invisible under marginal evaluation, which can give an incoherent and misleading view of performance.

### 5.3. Results with Functional KL

Next, we analyze the behavior of all methods for all datasets under a new light, by considering the divergence between trajectory distributions, which is unlocked by our estimator. For all the result tables, we report two additional columns: the forward $\mathrm{KL}(\nu^A \| \nu^B)$, and the reverse $\mathrm{KL}(\nu^B \| \nu^A)$, where $\nu^A$ stands for the reference, ground-truth path measure, and $\nu^B$ is the path-measure inferred by each TI method. Recall that these are learned through our estimation method, through the lenses of the parametric velocity fields trained on sampled trajectories.

**Finding 3: FKL yields more plausible rankings than marginal metrics.** Across datasets, FKL provides a con-sistent trajectory-level assessment of all methods, with lower values indicating smaller discrepancies between generated and ground-truth trajectory distributions. *(i) When marginal-metric-based CD diagrams agree with visual inspection, FKL agrees as well.* For example, on 2 of the 3 synthetic datasets (Lotka–Volterra and Repressilator), both marginal-metric-based CD diagrams (Figure 5) and FKL (Table 2) correctly identify VAL as the best-performing method. Likewise, both correctly assign poor performance to AM on Repressilator and to MFL on Lotka–Volterra and Petal. *(ii) More importantly, FKL remains consistent with visual inspection when marginal metrics become misleading, correcting the inconsistent rankings produced by existing methods.* Among all trajectory inference methods, FKL ranks SBIRR as the best method on Lotka–Volterra and Repressilator, and MSBM as the best on Petal. These conclusions match visual inspection but are not fully captured by marginal-metric-based CD plots. For instance, on Repressilator, marginal metrics rank TIGON above SBIRR, whereas FKL places SBIRR higher, in agreement with visual evidence. On Petal, the discrepancy is even more pronounced: although FKL correctly identifies VAL as the best and MSBM as the strongest trajectory inference method, marginal-metric-based CD plots rank VAL only second and place MSBM among the worst methods. On real-world dataset hESC, TIGON is consistently ranked best by all marginal metrics at the first validation timepoint ($\tau = 0.25$); in contrast, FKL reveals a mismatch in dynamics: TIGON generates smooth trajectories, whereas the reference paths are stochastic. Overall, these results suggest that marginal-based evaluation can be misaligned with trajectory-level quality, whereas our discrepancy measure FKL can recover the similarity in the dynamics between TI methods and produce rankings that are better aligned with visual evidence.

**Finding 4: Mode seeking (reverse KL) vs. mode coverage (forward KL).** Recall that $\nu^A$ denotes the reference, ground-truth path measure and $\nu^B$ the path measure inferred by a TI method. The asymmetry between $\mathrm{KL}(\nu^A \| \nu^B)$ and $\mathrm{KL}(\nu^B \| \nu^A)$ explains why the two divergences favor different methods empirically. *(i) The forward KL,* $\mathrm{KL}(\nu^A \| \nu^B)$, *favors mode coverage*: it strongly penalizes regions where the reference measure has nonzero mass but the inferred measure assigns little or no mass. Therefore, it favors methods that preserve broad support over the target distribution. This helps explain why forward KL ranks MFL best all real datasets (Table 3), as supported by both the optimization procedure of MFL and the trajectory visualizations. In MFL, a family of point clouds, one for each training snapshot, is initially sampled from a Gaussian distribution and then evolved toward the data by noisy gradient descent. Some particles that start far from the data region fail to move back to the data support, and the Brownian bridges connecting these outliers to correctly optimized particles then produce

trajectories that extend well beyond the data support, giving rise to a pronounced radiating pattern in the visualizations. As a result, the inferred trajectories cover not only the data region but also substantial off-support space. *(ii) By contrast, the reverse KL,* $\mathrm{KL}\!\left(\nu^B \,\|\, \nu^A\right)$*, favors mode-seeking*: it penalizes probability mass assigned to regions where the reference measure has little or no support, thereby favoring precision over coverage. Consistent with this property, reverse KL ranks MSBM best on all real datasets (Table 3), since its inferred trajectories stay closest to the data distribu-

tion, sometimes at the cost of missing lower-density modes. At the other extreme, MFL is among the worst methods on almost all synthetic and real datasets because it assigns substantial mass to regions far from the reference support. Taken together, these results reveal an asymmetry in the preferences of the two divergences that marginal metrics do not capture.

Figure 1 highlights the limitations of marginal-only evaluation. On the petal dataset, at validation time point $\tau = 0.75$,

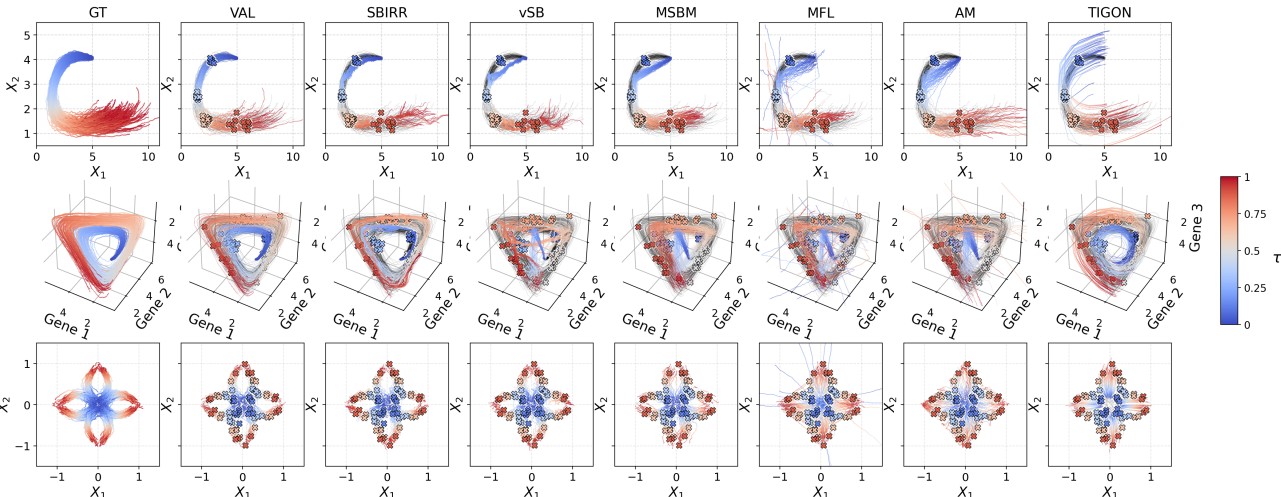

*Figure 4.* Trajectories generated by different TI methods across three synthetic datasets, from top to bottom: Lotka–Volterra, Repressilator, and Petal. Colored curves show generated trajectories, while black curves show Ground Truth (GT) trajectories. Colored crosses mark sampled validation snapshots used to compute marginal metrics.

| Method | $\tau = 0.125$ | | | | | $\tau = 0.375$ | | | | | $\tau = 0.625$ | | | | | $\tau = 0.875$ | | | | | FKL | |
|---|---|---|---|---|---|---|---|---|---|---|---|---|---|---|---|---|---|---|---|---|---|---|
| | $EMD$ | $W2$ | SWD | MWD | MMD | $EMD$ | $W2$ | SWD | MWD | MMD | $EMD$ | $W2$ | SWD | MWD | MMD | $EMD$ | $W2$ | SWD | MWD | MMD | $\mathrm{KL}(\nu^A\|\nu^B)$ | $\mathrm{KL}(\nu^B\|\nu^A)$ |
| VAL | 0.018 | 0.024 | 0.011 | 0.015 | 0.007 | 0.026 | 0.035 | 0.015 | 0.020 | 0.013 | 0.036 | 0.049 | 0.021 | 0.027 | 0.012 | 0.061 | 0.082 | 0.035 | 0.044 | 0.017 | 0.271 | 0.268 |
| SBIRR | **0.179** | **0.186** | **0.125** | **0.178** | **0.174** | **0.090** | **0.105** | **0.064** | **0.070** | **0.051** | **0.254** | **0.281** | **0.200** | **0.260** | 0.198 | 0.368 | 0.420 | 0.273 | 0.375 | 0.198 | **43.352** | **42.779** |
| vSB | 1.009 | 1.022 | 0.764 | 1.015 | 0.874 | 0.529 | 0.568 | 0.381 | 0.516 | 0.482 | 0.306 | 0.366 | 0.254 | 0.311 | 0.173 | **0.272** | **0.323** | **0.199** | **0.244** | 0.156 | 165.057 | 126.886 |
| MSBM | 0.784 | 0.786 | 0.595 | 0.784 | 0.716 | 0.321 | 0.325 | 0.232 | 0.323 | 0.306 | 0.486 | 0.503 | 0.361 | 0.501 | 0.426 | 0.554 | 0.637 | 0.452 | 0.625 | 0.388 | 79.872 | 46.023 |
| MFL | 1.004 | 1.140 | 0.801 | 0.905 | 0.615 | 0.393 | 0.639 | 0.458 | 0.590 | 0.247 | 0.402 | 0.573 | 0.396 | 0.490 | 0.265 | 0.659 | 0.958 | 0.698 | 0.875 | 0.240 | 43.929 | 130.579 |
| eAM | 0.862 | 0.864 | 0.655 | 0.863 | 0.763 | 0.342 | 0.406 | 0.299 | 0.381 | 0.277 | 0.569 | 1.079 | 0.804 | 1.072 | 0.239 | 1.134 | 1.958 | 1.449 | 1.945 | 0.203 | 44.914 | 55.488 |
| TIGON | 0.446 | 0.500 | 0.340 | 0.390 | 0.293 | 0.323 | 0.386 | 0.256 | 0.361 | 0.174 | 0.266 | 0.381 | 0.257 | 0.330 | **0.131** | **0.245** | 0.351 | 0.241 | 0.312 | **0.104** | 179.367 | 65.442 |

*(a)* Lotka–Volterra dataset.

| Method | $\tau = 0.1$ | | | | | $\tau = 0.3$ | | | | | $\tau = 0.5$ | | | | | $\tau = 0.7$ | | | | | $\tau = 0.9$ | | | | | FKL | |
|---|---|---|---|---|---|---|---|---|---|---|---|---|---|---|---|---|---|---|---|---|---|---|---|---|---|---|---|
| | $EMD$ | $W2$ | SWD | MWD | MMD | $EMD$ | $W2$ | SWD | MWD | MMD | $EMD$ | $W2$ | SWD | MWD | MMD | $EMD$ | $W2$ | SWD | MWD | MMD | $EMD$ | $W2$ | SWD | MWD | MMD | $\mathrm{KL}(\nu^A\|\nu^B)$ | $\mathrm{KL}(\nu^B\|\nu^A)$ |
| VAL | 0.036 | 0.044 | 0.014 | 0.019 | 0.014 | 0.056 | 0.070 | 0.023 | 0.034 | 0.011 | 0.080 | 0.099 | 0.033 | 0.052 | 0.014 | 0.094 | 0.114 | 0.042 | 0.060 | 0.023 | 0.116 | 0.147 | 0.055 | 0.081 | 0.030 | 0.015 | 0.014 |
| SBIRR | **0.390** | **0.413** | **0.256** | **0.401** | **0.366** | 0.865 | **0.899** | 0.558 | 0.868 | 0.688 | **0.442** | **0.475** | **0.263** | **0.345** | **0.291** | **0.742** | **0.815** | 0.477 | 0.750 | **0.390** | 1.832 | 1.894 | 1.055 | 1.852 | 0.545 | **23.519** | **25.242** |
| vSB | 1.884 | 1.889 | 1.174 | 1.884 | 1.270 | 1.227 | 1.248 | 0.724 | 1.156 | 0.880 | 0.983 | 1.017 | 0.596 | 0.972 | 0.611 | 1.059 | 1.122 | 0.640 | 0.948 | 0.605 | 0.944 | 1.078 | 0.576 | 0.901 | 0.537 | 82.933 | 79.014 |
| MSBM | 1.465 | 1.469 | 0.790 | 1.464 | 1.120 | 1.324 | 1.348 | 0.694 | 1.306 | 0.982 | 1.011 | 1.106 | 0.643 | 1.062 | 0.713 | 0.995 | 1.128 | 0.571 | 0.992 | 0.614 | 1.225 | 1.400 | 0.698 | 1.288 | 0.678 | 90.011 | 49.395 |
| MFL | 1.796 | 1.914 | 1.101 | 1.529 | 0.931 | 1.737 | 1.837 | 1.083 | 1.552 | 0.869 | 1.500 | 1.627 | 0.975 | 1.391 | 0.652 | 1.361 | 1.510 | 0.823 | 1.277 | 0.523 | 1.173 | 1.317 | 0.718 | 1.036 | 0.466 | 63.077 | 84.621 |
| eAM | 1.454 | 1.456 | 0.894 | 1.419 | 1.111 | 1.355 | 1.385 | 0.840 | 1.321 | 0.964 | 2.208 | 5.554 | 3.119 | 5.278 | 0.618 | 7.566 | 19.112 | 11.037 | 18.390 | 0.551 | 17.516 | 39.206 | 22.624 | 37.685 | **0.352** | 66.901 | 126.248 |
| TIGON | 1.074 | 1.135 | 0.656 | 0.915 | 0.687 | **0.851** | 0.952 | **0.501** | **0.760** | **0.480** | 0.821 | 0.884 | 0.496 | 0.720 | 0.441 | 0.813 | 0.860 | **0.444** | **0.663** | 0.431 | **0.842** | **0.916** | **0.483** | **0.677** | 0.430 | 54.515 | 42.844 |

*(b)* Repressilator dataset.

| Method | $\tau = 0$ | | | | | $\tau = 0.25$ | | | | | $\tau = 0.5$ | | | | | $\tau = 0.75$ | | | | | $\tau = 1$ | | | | | FKL | |
|---|---|---|---|---|---|---|---|---|---|---|---|---|---|---|---|---|---|---|---|---|---|---|---|---|---|---|---|
| | $EMD$ | $W2$ | SWD | MWD | MMD | $EMD$ | $W2$ | SWD | MWD | MMD | $EMD$ | $W2$ | SWD | MWD | MMD | $EMD$ | $W2$ | SWD | MWD | MMD | $EMD$ | $W2$ | SWD | MWD | MMD | $\mathrm{KL}(\nu^A\|\nu^B)$ | $\mathrm{KL}(\nu^B\|\nu^A)$ |
| VAL | 0.011 | 0.015 | 0.006 | 0.008 | 0.006 | 0.023 | 0.040 | 0.019 | 0.034 | 0.017 | 0.038 | 0.080 | 0.036 | 0.063 | 0.028 | 0.053 | 0.143 | 0.069 | 0.099 | 0.036 | 0.068 | 0.246 | 0.113 | 0.173 | 0.043 | 0.079 | 0.078 |
| SBIRR | 0.017 | 0.021 | 0.011 | 0.014 | 0.011 | **0.029** | **0.046** | **0.021** | **0.033** | 0.015 | **0.037** | **0.077** | **0.030** | **0.059** | **0.020** | 0.045 | **0.105** | **0.046** | **0.073** | 0.022 | **0.045** | **0.156** | **0.071** | **0.121** | **0.021** | 15.991 | 49.360 |
| vSB | **0.014** | **0.019** | 0.007 | **0.009** | 0.004 | **0.029** | 0.052 | 0.023 | **0.033** | 0.016 | 0.046 | 0.096 | 0.043 | 0.062 | 0.029 | 0.055 | 0.131 | 0.061 | 0.079 | 0.034 | 0.059 | 0.202 | 0.098 | 0.155 | 0.035 | 18.881 | 53.435 |
| MSBM | 0.015 | 0.021 | **0.007** | 0.009 | **0.003** | 0.043 | 0.062 | 0.027 | 0.050 | 0.021 | 0.070 | 0.111 | 0.053 | 0.089 | 0.043 | 0.096 | 0.160 | 0.084 | 0.131 | 0.057 | 0.119 | 0.254 | 0.126 | 0.226 | 0.067 | **9.641** | **17.055** |
| MFL | 0.194 | 0.690 | 0.482 | 0.515 | 0.051 | 0.287 | 0.516 | 0.352 | 0.385 | 0.054 | 0.284 | 0.445 | 0.296 | 0.316 | 0.106 | 0.184 | 0.387 | 0.251 | 0.264 | 0.062 | 0.105 | 0.397 | 0.254 | 0.285 | 0.023 | 42.660 | 68.191 |
| eAM | 0.016 | 0.022 | 0.008 | 0.011 | 0.008 | 0.102 | 0.115 | 0.059 | 0.081 | 0.028 | 0.167 | 0.194 | 0.095 | 0.138 | 0.067 | 0.183 | 0.224 | 0.114 | 0.163 | 0.082 | 0.166 | 0.283 | 0.149 | 0.205 | 0.088 | 12.328 | 31.641 |
| TIGON | 0.094 | 0.098 | 0.061 | 0.071 | 0.020 | 0.121 | 0.126 | 0.053 | 0.080 | **0.011** | 0.163 | 0.169 | 0.076 | 0.091 | 0.017 | 0.128 | 0.155 | 0.076 | 0.100 | **0.018** | 0.053 | 0.169 | 0.083 | 0.147 | 0.028 | 96.144 | 35.505 |

*(c)* Petal dataset.

*Table 2.* Marginal evaluation at each validation time $\tau$ and FKL results.

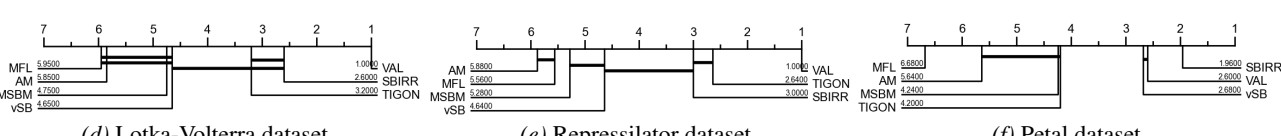

*(d)* Lotka-Volterra dataset.  *(e)* Repressilator dataset.  *(f)* Petal dataset.

*Figure 5.* Critical difference diagrams of marginal evaluation.

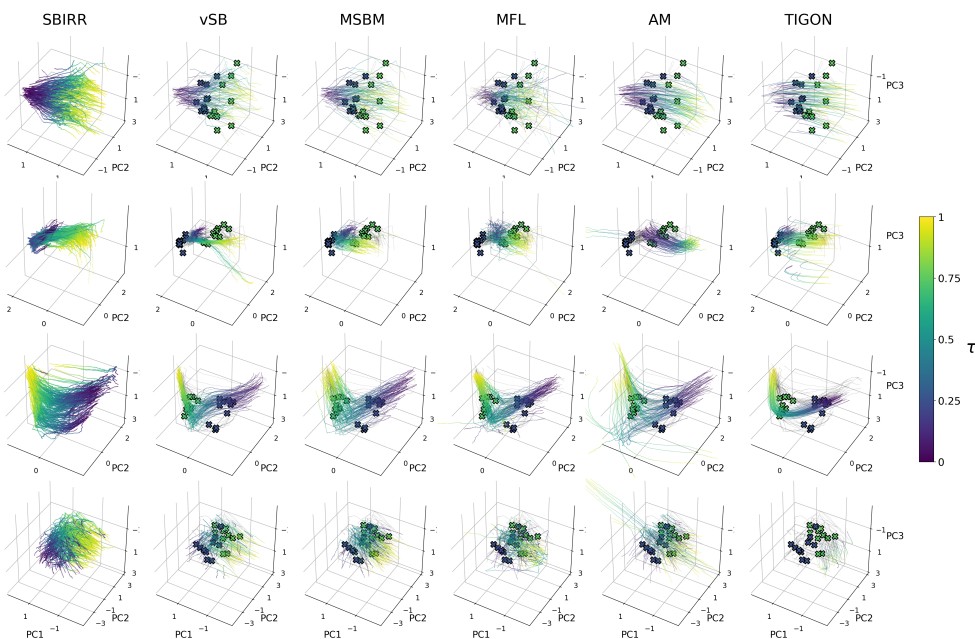

*Figure 6.* Trajectories generated by different TI methods across four real-world datasets, from top to bottom: EB, hESC, ME, and HF. Colored curves show generated trajectories, while black curves show reference trajectories inferred by SBIRR. Colored crosses mark sampled validation snapshots used to compute marginal metrics.

| Method | $\tau = 0.25$ | | | | | $\tau = 0.75$ | | | | | FKL | |
|---|---|---|---|---|---|---|---|---|---|---|---|---|
| | $EMD$ | $W_2$ | SWD | MWD | MMD | $EMD$ | $W_2$ | SWD | MWD | MMD | $\mathrm{KL}(\nu^A\|\nu^B)$ | $\mathrm{KL}(\nu^B\|\nu^A)$ |
| vSB | **0.699** | **0.763** | **0.215** | **0.342** | 0.195 | 0.856 | 0.920 | 0.290 | 0.502 | 0.213 | 23.778 | 27.727 |
| MSBM | 0.834 | 0.962 | 0.323 | 0.515 | 0.218 | **0.740** | **0.816** | **0.194** | **0.273** | **0.136** | 29.452 | **21.454** |
| MFL | 0.928 | 1.023 | 0.322 | 0.545 | 0.304 | 1.020 | 1.106 | 0.416 | 0.635 | 0.371 | **22.058** | 73.201 |
| eAM | 0.851 | 0.992 | 0.295 | 0.525 | 0.214 | 1.583 | 1.679 | 0.672 | 1.095 | 0.445 | 74.803 | 32.180 |
| TIGON | 1.040 | 1.250 | 0.334 | 0.654 | **0.187** | 1.094 | 1.226 | 0.378 | 0.610 | 0.206 | 122.486 | 41.076 |

*(a)* EB dataset.

| Method | $\tau = 0.25$ | | | | | $\tau = 0.75$ | | | | | FKL | |
|---|---|---|---|---|---|---|---|---|---|---|---|---|
| | $EMD$ | $W_2$ | SWD | MWD | MMD | $EMD$ | $W_2$ | SWD | MWD | MMD | $\mathrm{KL}(\nu^A\|\nu^B)$ | $\mathrm{KL}(\nu^B\|\nu^A)$ |
| vSB | 1.008 | 1.026 | 0.470 | 0.871 | 0.735 | **1.130** | **1.168** | **0.466** | **0.890** | 0.722 | 127.241 | 124.057 |
| MSBM | 1.011 | 1.036 | 0.444 | 0.891 | 0.703 | 1.179 | 1.216 | 0.508 | 0.951 | 0.736 | 111.151 | **81.697** |
| MFL | 1.167 | 1.193 | 0.508 | 0.951 | 0.794 | 1.183 | 1.223 | 0.481 | 0.956 | 0.757 | **97.134** | 117.901 |
| eAM | 2.194 | 2.243 | 1.086 | 2.068 | 1.023 | 2.206 | 2.242 | 1.025 | 1.875 | 1.047 | 145.552 | 283.293 |
| TIGON | **0.908** | **0.990** | **0.368** | **0.673** | **0.565** | 1.195 | 1.237 | 0.497 | 0.950 | **0.695** | 293.102 | 161.731 |

*(b)* hESC dataset.

| Method | $\tau = 0.25$ | | | | | $\tau = 0.75$ | | | | | FKL | |
|---|---|---|---|---|---|---|---|---|---|---|---|---|
| | $EMD$ | $W_2$ | SWD | MWD | MMD | $EMD$ | $W_2$ | SWD | MWD | MMD | $\mathrm{KL}(\nu^A\|\nu^B)$ | $\mathrm{KL}(\nu^B\|\nu^A)$ |
| vSB | 1.041 | 1.089 | 0.361 | 0.592 | 0.364 | **1.107** | **1.248** | **0.496** | **0.849** | **0.416** | 51.119 | 48.054 |
| MSBM | 0.943 | 1.011 | **0.288** | **0.444** | **0.290** | 1.209 | 1.298 | 0.501 | 0.921 | 0.417 | 65.571 | **37.563** |
| MFL | 1.054 | 1.125 | 0.385 | 0.563 | 0.417 | 1.398 | 1.517 | 0.603 | 1.014 | 0.586 | **42.268** | 79.306 |
| AM | **0.938** | **1.003** | 0.317 | 0.459 | 0.316 | 1.493 | 1.652 | 0.600 | 1.013 | 0.485 | 89.223 | 81.838 |
| TIGON | 1.091 | 1.189 | 0.472 | 0.803 | 0.425 | 1.432 | 1.524 | 0.656 | 1.094 | 0.563 | 250.619 | 82.386 |

*(c)* ME dataset.

| Method | $\tau = 0.25$ | | | | | $\tau = 0.75$ | | | | | FKL | |
|---|---|---|---|---|---|---|---|---|---|---|---|---|
| | $EMD$ | $W_2$ | SWD | MWD | MMD | $EMD$ | $W_2$ | SWD | MWD | MMD | $\mathrm{KL}(\nu^A\|\nu^B)$ | $\mathrm{KL}(\nu^B\|\nu^A)$ |
| vSB | 1.215 | 1.343 | 0.446 | 0.823 | 0.268 | 1.336 | 1.432 | 0.510 | 0.998 | 0.283 | 56.990 | 41.638 |
| MSBM | **0.956** | 1.063 | 0.362 | 0.674 | 0.216 | **1.168** | **1.275** | **0.426** | **0.765** | 0.279 | 58.914 | **26.784** |
| MFL | 1.151 | 1.266 | 0.477 | 0.827 | 0.323 | 1.368 | 1.476 | 0.542 | 0.930 | 0.379 | **29.706** | 69.830 |
| AM | 0.964 | **1.063** | **0.352** | **0.631** | **0.262** | 1.701 | 2.201 | 0.807 | 1.475 | **0.262** | 73.227 | 66.942 |
| TIGON | 2.371 | 2.482 | 1.097 | 2.138 | 0.592 | 2.031 | 2.199 | 0.907 | 1.688 | 0.441 | 197.769 | 69.540 |

*(d)* HF dataset.

*Table 3.* Marginal evaluation at each validation time $\tau$ and FKL results, with SBIRR trajectories used as the reference.

TIGON performs slightly better than MSBM under snapshot metrics, with W2 = 0.155 for TIGON versus 0.160 for MSBM, and a clearer advantage in MMD (0.018 versus 0.057), reflecting differences in local sample dispersion. Yet these marginal scores do not assess the underlying dynamics. In contrast, our path-level metric FKL, which compares full trajectory distributions rather than isolated time marginals, strongly separates the two methods (reverse FKL= 17.055 for MSBM and 35.505 for TIGON), revealing the different dynamics despite similar marginal fit. The zoomed inset in Figure 1 provides a visual counterpart to this observation: although snapshot distances at $\tau = 0.75$ are similar, the generated trajectories exhibit differences in their dynamics.

## 6. Conclusion

We introduced FKL, a general approach for estimating divergences between probability measures over function spaces. Grounded in the theory of infinite-dimensional flows, FKL yields a tractable estimator that can be learned from data and scales to realistic regimes in both the number of snapshots and the dimensionality of observations. We restricted our current FKL theory to mass-conserving dynamics, leaving the extension to unbalanced settings for future work.

We used FKL to re-examine trajectory inference in physics simulations and single-cell genomics. Current state-of-the-art methods are typically evaluated via marginal reconstruction on held-out snapshots, which does not directly assess the implied dynamics. Through an extensive empirical study on both synthetic and real datasets, we showed that these snapshot-based metrics can induce inconsistent method rankings and may even favor models whose assumptions conflict with the dynamics of the process generating the trajectories. By comparing full trajectory distributions to a reference distribution, FKL enables a coherent and principled assessment of TI methods. Across all datasets considered, this evaluation method aligns with qualitative expectations and resolves several counterintuitive outcomes produced by marginal criteria used in the literature.

## Impact Statement

This paper presents work whose goal is to advance the field of machine learning. There are many potential societal consequences of our work, none of which we feel must be specifically highlighted here.

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

# Appendices

## Contents

## A. Derivation of Functional KL Divergence

In this section, we present an expanded version of Section 3, with detailed proofs of the three lemmas included.

**To begin with, we clarify the probabilistic setting and state our objective.**

We extend the construction in Section 2 to accommodate *two different* probability spaces $(\Omega, \mathcal{F}, P^A)$ and $(\Omega, \mathcal{F}, P^B)$, both supporting independent random variables $X_0$ and $X_1$, such that $X_0$ has the same Gaussian law in both spaces whereas $X_1$ has laws $\mu_1^A = \nu^A$ and $\mu_1^B = \nu^B$, respectively. We then train one FFM model for each endpoint law, obtaining two parametrized pairs $(\mu_t^A, v_t^A)$ and $(\mu_t^B, v_t^B)$, each satisfying the weak continuity equation. We overload the notation previously introduced by considering superscripts to indicate whether the measures and fields of interest refer to the first or second probability space.

Our objective in this work is to estimate the KL divergence between two probability measures $\nu^A$ and $\nu^B$ on the Hilbert space $H$, which is formally defined as

$$
\mathrm{KL}\big(\nu^A \parallel \nu^B\big) := \begin{cases} \displaystyle\int_H \log\left(\frac{d\nu^A}{d\nu^B}\right) d\nu^A, & \text{if } \nu^A \ll \nu^B, \\ +\infty, & \text{otherwise.} \end{cases} \tag{8}
$$

**Next, we state the assumptions needed to construct our framework.**

**Assumption A.1** (Existence of Radon–Nikodym derivative)**.** We assume that $\nu^A \ll \nu^B$, so that the Radon–Nikodym derivative $\frac{d\nu^A}{d\nu^B}$ exists.

*Remark.* Indeed, as the goal of this work is to obtain KL estimates via a novel estimator, we need to assume that the underlying *estimand* (the true KL) is well-defined.

**Assumption A.2** (Cameron–Martin support)**.** The measures are fully supported on the Cameron–Martin space of the noise $(\nu^A(H_{\mu_0}) = \nu^B(H_{\mu_0}) = 1)$.

*Remarks.*

1. Assumption A.2 can be relaxed: together with Assumption A.1, it suffices to require that $\nu^B$ be fully supported on $H_{\mu_0}$, since $\nu^B(H_{\mu_0}) = 1$ paired with Assumption A.1 implies $\nu^A(H_{\mu_0}) = 1$. Indeed, by Assumption A.1, $\nu^A$ is absolutely continuous with respect to $\nu^B$ on $B(H)$, i.e., $\nu^B(\Gamma) = 0$ implies $\nu^A(\Gamma) = 0$ for all $\Gamma \in B(H)$. If $\nu^B(H_{\mu_0}) = 1$, then $\nu^B(H \setminus H_{\mu_0}) = 0$, and since the Hilbert subspace $H_{\mu_0}$ is Borel, Assumption A.1 yields $\nu^A(H \setminus H_{\mu_0}) = 0$, hence $\nu^A(H_{\mu_0}) = 1$.

2. Assumption A.2 can be satisfied by choosing noise that is rougher than the data distribution (so the data are fully supported on $H_{\mu_0}$) while still being trace-class (and hence in $H$). In practice, in $H := L^2(\mathbb{T}; \mathbb{R}^d)$ with Fourier orthonormal basis, we first estimate the data's Fourier spectrum using empirically determined variances for each Fourier coefficient, and then multiply each coefficient by the wavenumber magnitude $k = \|m\|_2$ to produce noise with a rougher spectrum while still in $H$ as discussed in (Chen & Vanden-Eijnden, 2025).

**Under these assumptions, we now develop a velocity-only representation of the KL divergence** $\mathrm{KL}(\nu^A \| \nu^B)$. The derivation is organized into three lemmas: Lemma A.3 establishes the absolute continuity needed to ensure the Radon–Nikodym derivatives are well-defined; Lemma A.4 applies weak continuity equation to express KL divergence with logarithmic Radon–Nikodym derivative and velocities. Lemma A.5 links the logarithmic gradients to the velocity fields.

**Lemma A.3.** *Under the Assumptions A.1 and A.2,*

*(a)* $\mu_t^{A,B} \ll \rho_t$, *where* $\rho_t = ((1-t)\mathrm{Id})_{\#}\mu_0$ *denotes the push-forward of* $\mu_0$ *by the map* $(1-t)\mathrm{Id}$ *for every* $t \in [0,1)$.

*(b)* $\mu_t^A \ll \mu_t^B$ *for every* $t \in [0,1]$.

*Proof.*

(a) By definition, $\rho_t$ remains a centered Gaussian measure on $H$ with covariance operator $C_t := (1-t)^2 C$. Using elementary Hilbert space theory, the Cameron-Martin space associated with $\mu_0$ is identical to that associated with $\rho_t$, equipped with an inner product scaled by $\frac{1}{(1-t)^2}$. Therefore, under Assumption A.2, we also have $\mu_1^{A,B}(H_{\rho_t}) = 1$.

For every fixed $t$, by the construction of linear interpolation (1), the conditional law of $X_t$ given $X_1 = x$ is the translation of $\rho_t$ by $tx$, i.e., $\mathcal{L}(X_t \mid X_1 = x) = (\tau_{tx})_{\#}\rho_t$ for $\mu_1$-almost any $x \in H$, where $\tau_h(\cdot) = \cdot + h$. Therefore, this conditional measure is also Gaussian, with mean $tx$ and covariance operator $C_t$. Since $\mu_1^{A,B}(H_{\rho_t}) = 1$ gives that $tx \in H_{\rho_t}$ for $\mu_1^{A,B}$-almost any $x \in H$, by the Cameron–Martin theorem (Da Prato & Zabczyk, 2014), we have that $(\tau_{tx})_{\#}\rho_t \sim \rho_t$, $x$ $\mu_1^{A,B}$-a.s. Furthermore, by definition of a conditional measure and Fubini's Theorem, for every $S \in B(H)$, we have

$$
\begin{aligned}
\mu_t^{A,B}(S) &= \int_H \left((\tau_{tx})_{\#}\rho_t\right)(S)\mu_1^{A,B}(dx) \\
&= \int_H \int_S \left((\tau_{tx})_{\#}\rho_t\right)(dy)\mu_1^{A,B}(dx) \\
&= \int_H \int_S \frac{d\left((\tau_{tx})_{\#}\rho_t\right)}{d\rho_t}(y)\rho_t(dy)\mu_1^{A,B}(dx) \\
&= \int_S \left(\int_H \frac{d\left((\tau_{tx})_{\#}\rho_t\right)}{d\rho_t}(y)\mu_1^{A,B}(dx)\right)\rho_t(dy).
\end{aligned}
$$

Hence $\mu_t^{A,B} \ll \rho_t$, and

$$
\frac{d\mu_t^{A,B}}{d\rho_t}(\cdot) = \int_H \frac{d\left((\tau_{tx})_{\#}\rho_t\right)}{d\rho_t}(\cdot)\mu_1^{A,B}(dx), \ \rho_t\text{-a.s.} \tag{9}
$$

(b) Denote $\frac{d\mu_t^{A,B}}{d\rho_t}$ by $f_t^{A,B}$, so $f_t^{A,B} \geq 0$. Let $S \in B(H)$ be such that $\mu_t^B(S) = \int_S f_t^B(y)\rho_t(dy) = 0$, then $\rho_t\left(S \cap \{f_t^B > 0\}\right) = 0$, so

$$\mu_t^A(S) = \int_S f_t^A(y)\rho_t(dy)$$

$$= \int_{S \cap \{f_B > 0\}} f_t^A(y)\rho_t(dy) + \int_{S \cap \{f_B = 0\}} f_t^A(y)\rho_t(dy) = \int_{S \cap \{f_B = 0\}} f_t^A(y)d\rho_t(dy)$$

Therefore, to get $\mu_t^A(S) = 0$ (hence $\mu_t^A \ll \mu_t^B$), it suffices to show that $f_t^A = 0$ on $\{y \in H : f_t^B(y) = 0\}$, $\rho_t$-a.s. To this end, observe that $f_t^{A,B}$ admit the representation $f_t^{A,B}(y) = \int_H g_t(x,y)\mu_1^{A,B}(dx)$, where $g_t(x,y) := \frac{d\left((\tau_{tx})_{\#}\rho_t\right)}{d\rho_t}(y) \geq 0$. Now fix $y \in H$. If $f_t^B(y) = 0$, then $\mu_1^B\left(H \cap \{g_t(x,y) > 0\}\right) = 0$. Under Assumption A.1, we also have $\mu_1^A\left(H \cap \{g_t(x,y) > 0\}\right) = 0$, and therefore $f_t^A(y) = \int_H g_t(x,y)\mu_1^A(dx) = 0$.

$\square$

Given the Radon–Nikodym derivatives are well-defined, next we apply the weak continuity equation using the Radon–Nikodym derivative and its logarithm as test functions to derive an integral representation of the KL divergence.

**Lemma A.4.** *Let* $r_t := \frac{d\mu_t^A}{d\mu_t^B}$, *which is well-defined by Lemma A.3* (b). *Then, under mild regularity conditions,*

$$\mathrm{KL}\left(\nu^A \,\|\, \nu^B\right) = \int_I \int_H \langle v_t^A(x) - v_t^B(x), \nabla_x \log r_t(x)\rangle \, d\mu_t^A(x). \tag{10}$$

*Proof.* Step 1: Admissibility of $r_t$ and $\log r_t$ as test functions in weak continuity equation.

Given $\{e_k\}_{k \in \mathbb{N}}$, let $\hat{\pi}^K = \sum_{i=1}^K e_i \otimes e_i$ be the orthogonal projector of $H$ onto the linear span $H_K = \lin\{e_1, \ldots, e_K\}$.

For all $K \in \mathbb{N}$, we first define the projected measures $\hat{\mu}_{t,K}^{A,B} := (\hat{\pi}^K)_{\#}\mu_t^{A,B}$. Clearly, $\hat{\mu}_{t,K}^{A,B} = E_{P^{A,B}}[\mu_t^{A,B}|\mathcal{G}_K]$, where $\mathcal{G}_K := \sigma(\hat{\pi}^K(X_t))$. Note also that $\{\mathcal{G}_K\}_{K \in \mathbb{N}}$ is a growing filtration with terminal value $\mathcal{G}_\infty = \sigma(X_t)$.

Next, we define the projected Radon-Nikodym derivative $r_t^K := \frac{d\hat{\mu}_{t,K}^A}{d\hat{\mu}_{t,K}^B} = E_{P^B}[r_t \mid \mathcal{G}_K]$, where $r_t := \frac{d\mu_t^A}{d\mu_t^B}$ is well-defined by Lemma A.3 (b). By definition of a Radon-Nikodym derivative, $r_t \in L^1(H \times I, \mu_t^B; \mathbb{R})$, so by Doob's martingale convergence theorem ((Øksendal, 2010) Corollary C.9), $r_t^K = E[r_t \mid \mathcal{G}_K] \xrightarrow{K \to \infty} E[r_t \mid \mathcal{G}_\infty] = r_t$, $\mu_t^B$-a.s. and in $L^1(\mu_t^B)$.

Now fix $K$. Since $r_t^K \in L^1(H^K \times I, \hat{\mu}_{t,K}^B; \mathbb{R})$, and smooth functions with compact support are dense in $L^1(H^K \times I, \hat{\mu}_{t,K}^B; \mathbb{R})$ (Nakai et al., 2004), therefore we can extract a sequence $(r_t^{K,n})_{n \in \mathbb{N}} \subset C_c^\infty(H^K \times I, \hat{\mu}_{t,K}^B; \mathbb{R})$ such that $r_t^{K,n} \xrightarrow{n \to \infty} r_t^K$ in $L^1(H^K \times I, \hat{\mu}_{t,K}^B; \mathbb{R})$.

Combining the two convergence together and using the reverse triangle inequality and the Cauchy–Schwarz inequality, we then have

$$\left|\left\|r_t^{K,n}\right\|_{L^1(\hat{\mu}_{t,K}^B)} - \|r_t\|_{L^1(\mu_t^B)}\right| \leq \left\|r_t^{K,n} - r_t\right\|_{L^1(\mu_t^B)} \leq \underbrace{\left\|r_t^{K,n} - r_t^K\right\|_{L^1(\hat{\mu}_{t,K}^B)}}_{=:A^{K,n}} + \underbrace{\left\|r_t^K - r_t\right\|_{L^1(\mu_t^B)}}_{=:B^K}.$$

For each $K$, since $A^{K,n} \to 0$ as $n \to \infty$, there exists $m_K \in \mathbb{N}$ such that for $n \geq m_K$, we have $A^{K,n} < \frac{1}{K}$; therefore, $A^{K,n} + B^K < \frac{1}{K} + B^K \xrightarrow{K \to \infty} 0$. Consequently, $r_t$ can be approximated arbitrarily well by smooth cylindrical test functions $\left(r_t^{K,n(K)}\right)$, and the interchange of the integral in the definition of the $L^1$-norm and the limit is justified.

Analogously, assume that $\partial_t \frac{d\mu_t^A}{d\mu_t^B} \in L^1(\mu_t^B)$ and $\nabla_x \frac{d\mu_t^A}{d\mu_t^B} \in L^1(\mu_t^B)$, then the $L^1\left(\mu_t^B\right)$-convergence and the interchange of integral and limit also hold for their corresponding smooth cylindrical approximations. Furthermore, assume that

$v_t^{A,B} \in L^\infty(\mu_t^{A,B})$, by Hölder's inequality, we get

$$\left\| \left\langle v_t^{A,B}, \nabla r_t^{K,n(K)} - \nabla r_t \right\rangle \right\|_{L^1(\mu_t^B)} \leq \left\| v_t^{A,B} \right\|_{L^\infty(\mu_t^{A,B})} \left\| \nabla r_t^{K,n(K)} - \nabla r_t \right\|_{L^1(\mu_t^B)} \to 0.$$

Therefore, letting $K \to \infty$ and $n(K) \to \infty$ in the weak continuity equation ((2)) tested against $r_t^{K,n(K)}$ yields

$$\int_H r_1(x) d\mu_1^{A,B}(x) - \int_H r_0(x) d\mu_0^{A,B}(x) = \int_I \int_H \left( \partial_t r_t(x) + \left\langle v_t^{A,B}(x), \nabla_x r_t(x) \right\rangle \right) d\mu_t^{A,B}(x) dt.$$

An analogous equation holds for $\log r_t$, under the assumption that $\log r_t \in L^1(\mu_t^B)$, $\partial_t \log r_t \in L^1(\mu_t^B)$, and $\nabla_x \log r_t \in L^1(\mu_t^B)$.

Step 2: KL identify from the weak continuity equation.

We first consider the weak continuity equation tested with $\log r_t(x)$ for the pair $(v_t^A, \mu_t^A)$. Using the boundary identities

$$\log r_1 = \log \frac{d\mu_1^A}{d\mu_1^B} = \mathrm{KL}\left(\nu^A \,\|\, \nu^B\right), \quad \log r_0 = \log \frac{d\mu_0^A}{d\mu_0^B} = \log 1 = 0, P^B\text{-a.s.}$$

the L.H.S. of (2) reduces to $\mathrm{KL}\left(\nu^A \,\|\, \nu^B\right)$, so

$$\mathrm{KL}\left(\nu^A \,\|\, \nu^B\right) = \int_I \int_H \partial_t \log r_t(x) d\mu_t^A(x) dt + \int_I \int_H \left\langle v_t^A(x), \nabla_x \log r_t(x) \right\rangle d\mu_t^A(x) dt \tag{11}$$

Next, we want to rewrite the time-derivative term on the R.H.S. of Equation (11) in terms of $v_t^B$ and $\nabla_x \log r_t$. To this end, we consider the weak continuity equation tested with $r_t(x)$ for the pair $(v_t^B, \mu_t^B)$. By definition $r_t := \frac{d\mu_t^A}{d\mu_t^B}$, we have

$$\int_H r_1 \, d\mu_1^B = \int_H d\mu_1^A = 1, \quad \int_H r_0 \, d\mu_0^B = \int_H d\mu_0^A = 1$$

the L.H.S. of (2) reduces to 0, so

$$\int_I \int_H \partial_t r_t(x) \, d\mu_t^B(x) \, dt = - \int_I \int_H \left\langle v_t^B(x), \nabla_x r_t(x) \right\rangle d\mu_t^B(x) \, dt.$$

Using $\mu_t^B(dx) = \frac{1}{r_t}(x)\mu_t^A(dx)$, we then get

$$\int_I \int_H \partial_t \log r_t(x) \, d\mu_t^A(x) \, dt = - \int_I \int_H \left\langle v_t^B(x), \nabla_x \log r_t(x) \right\rangle d\mu_t^A(x) \, dt \tag{12}$$

Finally, injecting (12) into the R.H.S. of Equation (11) gives the desired result.

$\square$

By Lemma A.3 $(a)$, we can rewrite the term $\nabla_x \log r_t(x)$ appearing in Lemma A.4 as

$$\nabla_x \log r_t(x) = \nabla_x \log \frac{d\mu_t^A}{d\rho_t}(x) - \nabla_x \log \frac{d\mu_t^B}{d\rho_t}. \tag{13}$$

The next lemma links this logarithmic gradients mismatch to velocity fields mismatch, which is useful to derive a velocity-only representation of $\mathrm{KL}\left(\nu^A \,\|\, \nu^B\right)$.

**Lemma A.5.** *For the linear interpolation used in FFM (Kerrigan et al., 2024), under the assumption that $v_t^A(x) - v_t^B(x) \in \mathrm{Dom}\left(C^{-1}\right) = \mathrm{Range}(C) \quad \mu_t^A(\mathrm{d}x) \, \mathrm{d}t$-a.e., we have*

$$\nabla_x \log \frac{d\mu_t^A}{d\rho_t}(x) - \nabla_x \log \frac{d\mu_t^B}{d\rho_t} = \frac{t}{1-t} C^{-1} \left( v_t^A(x) - v_t^B(x) \right) \tag{14}$$

*Proof.* Step 1: Expression of logarithmic gradients.

By Equation (9), the logarithmic gradients can be written as

$$\nabla_x \log \frac{d\mu_t^{A,B}}{d\rho_t}(x) = \frac{\int_H \nabla_x \frac{d(\tau_{th})_\# \rho_t}{d\rho_t}(x) d\mu_1^{A,B}(h)}{\int_H \frac{d(\tau_{th})_\# \rho_t}{d\rho_t}(x) d\mu_1^{A,B}(h)}. \tag{15}$$

Since $(\tau_{th})_\# \rho_t$ is the translate of $\rho_t = \mathcal{N}(0, (1-t)^2 C)$ by $th$, the Cameron-Martin theorem yields

$$\nabla_x \frac{d\left((\tau_{th})_\# \rho_t\right)}{d\rho_t}(x) = \frac{d\left((\tau_{th})_\# \rho_t\right)}{d\rho_t}(x) \frac{t}{(1-t)^2} C^{-1} h. \tag{16}$$

Substituting (16) into (15) gives

$$\nabla_x \log \frac{d\mu_t^{A,B}}{d\rho_t}(x) = \frac{t}{(1-t)^2} C^{-1} \int_H h \underbrace{\frac{\frac{d(\tau_{th})_\# \rho_t}{d\rho_t}(x) \mu_1^{A,B}(dh)}{\int_H \frac{d(\tau_{th})_\# \rho_t}{d\rho_t}(x) \mu_1^{A,B}(dh)}}_{=:(I)}. \tag{17}$$

To identify the ratio $(I)$ in (17), note that the conditional law of $X_t$ given $X_1 = h$ is: $\forall \Gamma \in B(H)$,

$$P^{A,B}\left(X_t \in \Gamma \mid X_1 = h\right) = (\tau_{th})_\# \rho_t(\Gamma) = \int_\Gamma \frac{d\left(\tau_{th}\right)_\# \rho_t}{d\rho_t}(x) \rho_t(dx),$$

hence the joint law of $(X_t, X_1)$ admits the factorization: $\forall \Gamma, \Lambda \in B(H)$,

$$P^{A,B}\left(X_t \in \Gamma, X_1 \in \Lambda\right) = \int_\Lambda P^{A,B}\left(X_t \in \Gamma \mid X_1 = h\right) \mu_1^{A,B}(dh) = \int_\Lambda \int_\Gamma \frac{d\left(\tau_{th}\right)_\# \rho_t}{d\rho_t}(x) \rho_t(dx) \mu_1^{A,B}(dh),$$

i.e.,

$$P^{A,B}\left(X_t = x, X_1 \in dh\right) = \frac{d\left(\tau_{th}\right)_\# \rho_t}{d\rho_t}(x) \mu_1^{A,B}(dh).$$

Therefore, by Bayes' rule, $(I)$ in (17) becomes

$$\frac{\frac{d(\tau_{th})_\# \rho_t}{d\rho_t}(x) \mu_1^{A,B}(dh)}{\int_H \frac{d(\tau_{th})_\# \rho_t}{d\rho_t}(x) \mu_1^{A,B}(dh)} = \frac{P^{A,B}\left(X_t = x, X_1 \in dh\right)}{P^{A,B}\left(X_t = x\right)} = P^{A,B}\left(X_1 \in dh \mid X_t = x\right)$$

Inserting the last display back into (17) gives

$$\begin{aligned} \nabla_x \log \frac{d\mu_t^{A,B}}{d\rho_t}(x) &= \frac{t}{(1-t)^2} C^{-1} \int_H h P^{A,B}\left(X_1 \in dh \mid X_t = x\right) \\ &= \frac{t}{(1-t)^2} C^{-1} E_{P^{A,B}}[X_1 | X_t = x]. \end{aligned} \tag{18}$$

Step 2: Expression of velocity fields.

For the linear interpolation used in FFM (Kerrigan et al., 2024), the velocity fields are given by

$$\begin{aligned} v_t^{A,B}(x) &= \mathbb{E}_{P^{A,B}}\left[\frac{X_1 - x}{1-t} \mid X_t = x\right] \\ &= \frac{1}{1-t} \mathbb{E}_{P^{A,B}}[X_1 \mid X_t = x] - \frac{x}{1-t} \end{aligned} \tag{19}$$

Step 3: Linking logarithmic gradients and velocities.

The domain condition $v_t^A(x) - v_t^B(x) \in \mathrm{Dom}\left(C^{-1}\right) = \mathrm{Range}(C)$   $\mu_t^A(\mathrm{d}x)\,\mathrm{d}t$-a.e. allows us to combine identities (18) and (19), yielding an expression for the logarithmic gradients mismatch in terms of the velocity fields mismatch:

$$\nabla_x \log \frac{d\mu_t^A}{d\rho_t}(x) - \nabla_x \log \frac{d\mu_t^B}{d\rho_t} = \frac{t}{1-t} C^{-1}\left(v_t^A(x) - v_t^B(x)\right)$$

$\square$

Finally, with $\nabla_x \log r_t$ in Lemma A.4 replaced by velocity fields mismatch in Lemma A.5, we obtain a velocity-only expression for what we label the FKL:

**Theorem A.6.** *For the linear interpolation used in FFM (Kerrigan et al., 2024), under Assumptions A.1–A.2 and mild regularity conditions, we have*

$$\mathrm{KL}\left(\nu^A \,\|\, \nu^B\right) = \int_0^1 \int_H \frac{t}{1-t} \left\|v_t^A(x) - v_t^B(x)\right\|_{H_{\mu_0}}^2 \, d\mu_t^A(x)\, dt. \tag{20}$$

## B. Implementation of Functional KL Estimation

Given the velocity-field-based expression of the FKL under the FFM framework (Equation 7), we now describe our implementation for estimating KL in function space.

**Parameterization of two velocity fields.**    Velocity field estimation in FFM is well established (Kerrigan et al., 2024). The key difference between (Kerrigan et al., 2024) and our setting is that, generative modeling typically approximates a single velocity field (e.g., $v^A$ by $v_\theta^A$ to generate samples from $\nu^A$), whereas our KL estimator requires approximating two velocity fields, $v^A$ and $v^B$.

To reduce parameters and simplify training, we use a single network with a binary conditioning flag $c \in \{0,1\}$: $v_\theta(c = 0, \cdot) \approx v^A$ and $v_\theta(c = 1, \cdot) \approx v^B$. The network is trained on shuffled function samples $x_1$ from $\nu^A$ and $\nu^B$, each paired with a label $c \in \{0,1\}$ indicating the target velocity field.

**Architecture and training.**    Our network is based on the state-of-the-art functional neural operator Mesh-Informed Neural Operator (MINO) (Shi et al., 2025), an encoder–decoder neural operator that uses Graph Neural Operator (GNO)s and Transformers to map between arbitrary irregular discretizations and a latent representation on a fixed regular grid. Compared with Fourier Neural Operator (FNO), which is used in many functional generative models including (Kerrigan et al., 2024), MINO naturally handles irregular grids and, in our preliminary experiments, performs substantially better on functions with non-periodic boundaries.

For stable FKL estimation, it is important to enforce $v(x, 1) = x$. Indeed, under $X_0 \sim \mathcal{N}(0, C)$ and $X_0 \perp X_1$, the velocity field satisfies $v(x, 1) = \mathbb{E}[X_1 - X_0 \mid X_1 = x] = x$. Hence, if both $v_\theta^A$ and $v_\theta^B$ satisfy this condition, their difference vanishes at $t = 1$, thereby canceling the singularity from $\frac{t}{1-t}$ in Equation (7). To impose this condition, we use the subtraction-based parameterization of (Hu et al., 2025):

$$v_\theta(x, t) = x + m_\theta(x, t) - m_\theta(x, 1), \tag{21}$$

which guarantees $v_\theta(x, 1) = x$.

We optimize the neural network with a conditional flow-matching objective (Kerrigan et al., 2023) that linearly combines the squared Hilbert and Cameron-Martin norms of the velocity error, with increasing emphasis on the Cameron-Martin term during training to progressively refine high-frequency modes at later stages. In practice, all fields are represented in a truncated spectral basis, so both norms are well defined.

**Estimation of FKL.**    Given the two learned velocity fields, we estimate FKL via Monte Carlo approximation of Equation (7). Specifically, we sample $x_1^A \sim \nu^A$, $t \sim \mathcal{U}[0, 1]$, and $x_0 \sim \mu_0 = \mathcal{N}(0, C)$, and form $x_t^A = tx_1^A + (1-t)x_0$, so that $x_t^A \sim \mu_t^A$. Evaluating both fields at $(x_t^A, t)$ yields $v_\theta^A(x_t^A, t)$ and $v_\theta^B(x_t^A, t)$. We then compute the integrand $\frac{t}{1-t}\left\|v_\theta^A(x_t^A, t) -\right.$

$v_\theta^B(x_t^A, t)\big\|_{H_{\mu_0}}^2$, with the norm computed after projecting the velocity fields onto a suitable orthonormal basis, as in prior function-space literature (Kerrigan et al., 2024; Franzese et al., 2023). Averaging over repeated samples yields the FKL estimate. Importantly, this Monte Carlo approximation procedure does not require simulating the full generative dynamics via ODE integration.

## C. Special Cases

In this section, we consider two analytically tractable special cases to validate our functional-space KL formulation. These settings admit closed-form KL values, which serve as ground truth and enable direct verification of both our theoretical derivation and the resulting KL estimation pipeline. Empirically, our estimates closely match the closed-form values, providing strong evidence that the derivation and the resulting estimator are accurate in practice. Section C.1 derives the closed-form KL expressions for the Gaussian-measure and linear-SDE cases, while Section C.2 reports experimental details and results.

### C.1. Closed-form Expression

#### C.1.1. SPECIAL CASE 1: GAUSSIAN MEASURES

We consider the target measures $\nu^{A,B}$ on some separable Banach space $H$ to be Gaussian measures $\mathcal{N}(m^{A,B}, \mathcal{R})$. Provided $m^A - m^B \in H_{\mathcal{R}}$, the Cameron Martin space, the KL divergence has known analytical expression $\frac{1}{2}\|m^A - m^B\|_{H_C}^2$. We consider $H = L^2(\mathbb{T}; \mathbb{R})$ and Matérn covariance $\mathcal{R} = \sigma^2(-\Delta_P + \tau^2 I)^{-\alpha}$, $\sigma, \tau, \alpha > 0$. For simplicity, we select $m^A = \cos(2\pi x) \in H_C$ and $m^B = 0$. For the analytical computation, we select as ONB $\{e_k(x) = e^{2\pi i k \cdot x} : k \in \mathbb{Z}\}$, where $\mathcal{K}e_k = \lambda_k e_k$, $\lambda_k = \sigma^2(4\pi^2 k^2 + \tau^2)^{-\alpha}$.

Furthermore, the velocity field mismatch also admits an analytic expression. Recall that the velocity field is defined as $v_t(r) = \mathbb{E}[\dot{X}_t \mid X_t = r]$. Applying the Gaussian conditioning formula in Hilbert space (Mandelbaum, 1984) to the linear interpolation yields $v_t(r) = m + (tC - (1-t)K)((1-t)^2 K + t^2 C)^{-1}(r - tm)$. Equivalently, in Fourier coordinates,

$$v_{t,k}(r_k) = m_k + \underbrace{\frac{tc_k - (1-t)\kappa_k}{(1-t)^2 \kappa_k + t^2 c_k}}_{=:a_k(t)} (r_k - tm_k). \tag{22}$$

Let $v_t^A$ and $v_t^B$ denote the velocity field corresponding to $\nu_A = \mathcal{N}(m, C)$ and $\nu_B = \mathcal{N}(0, C)$ respectively, then $v_{t,k}^A(r_k) = m_k + a_k(t)(r_k - tm_k)$ and $v_{t,k}^B(r_k) = a_k(t)r_k$, so their difference simplifies to a deterministic (input-independent) quantity:

$$v_{t,k}^{\text{diff}} := v_{t,k}^A - v_{t,k}^B = \frac{(1-t)\kappa_k}{(1-t)^2 \kappa_k + t^2 c_k} m_k. \tag{23}$$

#### C.1.2. SPECIAL CASE 2: SDEs

Let $(Y_t)_{t \in [0,1]}$ be an $\mathbb{R}^D$-valued process and consider the two SDEs

$$\begin{cases} \text{A:} & dY_t = c_A Y_t \, dt + g \, dW_t, \\ \text{B:} & dY_t = c_B Y_t \, dt + g \, dW_t, \end{cases} \qquad Y_0 \sim \mathcal{N}(m_0, \Sigma_0), \tag{24}$$

where $c_A, c_B \in \mathbb{R}$ are scalar drift coefficients and we assume $c_A \neq 0$, $g > 0$ is a constant diffusion, and $W_t$ is a standard $D$-dimensional Wiener process. Denote the induced path measures on $[0, 1]$ by $\nu^A$ and $\nu^B$, and define $S_0 := \text{Tr}(\Sigma_0)$, $M_0 := \|m_0\|^2$.

Since the two SDEs have the same diffusion and initial law, Girsanov's theorem gives

$$\text{KL}(\nu_A \| \nu_B) = \frac{1}{2} \int_0^1 \mathbb{E}_A\left[\left\|\frac{(c_A - c_B)Y_t}{g}\right\|^2\right] dt = \frac{(c_A - c_B)^2}{2g^2} \int_0^1 \mathbb{E}_A\left[\|Y_t\|^2\right] dt. \tag{25}$$

**Step 1: compute $\mathbb{E}_A[\|Y_t\|^2]$.** Under the SDE A, the explicit solution is

$$Y_t = e^{c_A t}Y_0 + g \int_0^t e^{c_A(t-s)} \, dW_s.$$

Using independence of $Y_0$ and $(W_s)_{s \geq 0}$, the cross-term vanishes and Itô isometry yields

$$\mathbb{E}_A[\|Y_t\|^2] = e^{2c_A t} \mathbb{E}[\|Y_0\|^2] + g^2 \mathbb{E}\left[\left\|\int_0^t e^{c_A(t-s)} dW_s\right\|^2\right]$$

$$= e^{2c_A t}(M_0 + S_0) + g^2 \int_0^t e^{2c_A(t-s)} \mathbb{E}[\|dW_s\|^2]$$

$$= e^{2c_A t}(M_0 + S_0) + g^2 D \int_0^t e^{2c_A(t-s)} ds. \tag{26}$$

For $c_A \neq 0$,

$$\int_0^t e^{2c_A(t-s)} ds = \frac{e^{2c_A t} - 1}{2c_A},$$

hence

$$\mathbb{E}_A[\|Y_t\|^2] = e^{2c_A t}(M_0 + S_0) + \frac{g^2 D}{2c_A}\left(e^{2c_A t} - 1\right). \tag{27}$$

**Step 2: integrate over $t \in [0, 1]$.** Plugging (27) into (25) gives, for $c_A \neq 0$,

$$\text{KL}(\nu_A \| \nu_B) = \frac{(c_A - c_B)c^2}{2g^2} \int_0^1 \left[e^{2c_A t}(M_0 + S_0) + \frac{g^2 D}{2c_A}(e^{2c_A t} - 1)\right] dt$$

$$= \frac{(c_A - c_B)^2}{2g^2} \left[(M_0 + S_0) \int_0^1 e^{2c_A t} dt + \frac{g^2 D}{2c_A} \left(\int_0^1 e^{2c_A t} dt - 1\right)\right]$$

$$= \frac{(c_A - c_B)^2}{2g^2} \left[(M_0 + S_0)\frac{e^{2c_A} - 1}{2c_A} + \frac{g^2 D}{2c_A} \left(\frac{e^{2c_A} - 1}{2c_A} - 1\right)\right] \tag{28}$$

where the last equality follows from the identity $\int_0^1 e^{2c_A t} dt = \frac{e^{2c_A} - 1}{2c_A}$.

### C.2. Experimental Details

The hyperparameters are reported in Table 4 for the Gaussian case and in Table 5 for the SDE case.

## D. Additional Experiments

### D.1. Ablation on Noise Covariance

In our function-space formulation, the FFM-based FKL estimator is accurate and resolution-invariant only when the noise covariance operator $C$ satisfies two conditions: *(i)* $C$ is trace-class on the Hilbert space $H$; and *(ii)* $C$ satisfies the Cameron-Martin support assumption (Assumption 3.2), i.e., the Gaussian noise $\mathcal{N}(0, C)$ must be rougher than the data measure. To verify this empirically, we conduct an ablation study on a Gaussian special case (Case 2 in Table 1 (a) with data smoothness $\alpha_1 = 3.5$). We test three choices of $C$ and evaluate the FKL estimate on trajectories sampled at resolutions $M \in \{128, 256, 512, 1024\}$, while keeping the FFM trained at resolution $M = 256$. Results are shown in Figure 7.

When $C = \text{Id}$ (white noise), condition *(i)* fails because $C$ is not trace-class; consequently, the estimator is not function-space resolution-invariant, with the estimated FKL diverging as the inference resolution increases. When $C$ is Matérn with smoothness $\alpha_0 = 6.0$, condition *(i)* holds, but condition *(ii)* fails because the noise is smoother than the data measure ($\alpha_0 > \alpha_1$), leading to unstable and divergent FKL estimates across all resolutions. In contrast, Matérn $C$ with $\alpha_0 = 0.5$ satisfies both conditions, yielding accurate and resolution-invariant forward and reverse FKL estimates, which highlights the importance of choosing a noise covariance that meets both conditions for FKL evaluation.

For the empirical comparison between white noise and trace-class Matérn noise with smoothness $\alpha_0 = 0.5$, in addition to demonstrating the improved robustness of KL estimation under super-resolution afforded by trace-class noise (see Figure 7c), we further show that trace-class noise also provides more robust generation quality under super-resolution (see Figure 8). A theoretical discussion of this choice is given in (Franzese, 2025).

| Name | Value |
|---|---|
| Training function $X_1^A$ | $\begin{cases} f \sim \mathcal{N}(\mu, C) \\ \mu(x) = s\sin(2\pi f_0 x),\ s \in \{0.5,\ 1.5,\ 3.0\},\ f_0 \in \{1,\ 3,\ 5\} \\ C : \text{Matérn w. } \nu_C = 3.5,\ \ell_C = 0.05,\ \sigma_C^2 = 0.15 \end{cases}$ |
| Training function $X_1^B$ | $f \sim \mathcal{N}(0, C)$ |
| Training functions' input time points $M$ | 128 |
| Training functions' output dimension $D$ | $\in \{1, 2, 3, 5, 10\}$ |
| Noise function $X_0$ | $f \sim \mathcal{N}(0, K),\ K : \text{Matérn w. } \nu_K = 0.5,\ \ell_K = 0.1,\ \sigma_K^2 = 1.0$ |
| Num. modes $N$ summed at KL estimation | 64 |
| $t$ sampling scheme at training | Logit-normal |
| $t$ sampling scheme at KL estimation | Importance sampling $t/(1-t)$ |
| Num. $t$ sampled at KL estimation | 100 |
| Num. functions at training | 50,000 |
| Training batch size | 1024 |
| Training iterations | 30,000 |
| Num. functions at KL estimation | 500 |
| Optimizer | Adam |
| EMA rate | 0.999 |
| Learning Rate (LR) | 1e−3 |
| LR scheduler | Cosine annealing |
| FFM training loss | $L_{\text{FFM}}$ |
| Model | MINO-T |
| Encoder (dim / depth / heads) | 64 / 2 / 4 |
| Decoder (dim / depth / heads) | 64 / 2 / 4 |
| Supernode radius | 5e−3 |
| GPUs for Training | $1 \times$ NVIDIA A100 |

*Table 4.* FKL hyperparameters for Gaussian Measures.

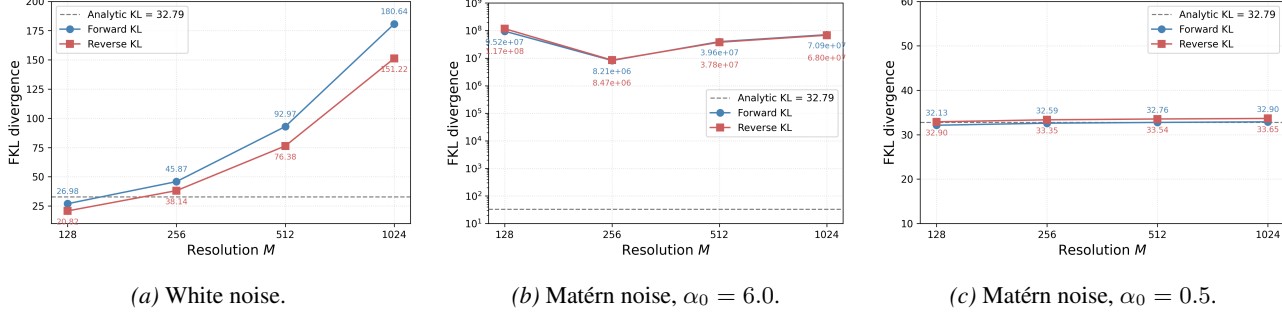

*(a)* White noise.  *(b)* Matérn noise, $\alpha_0 = 6.0$.  *(c)* Matérn noise, $\alpha_0 = 0.5$.

*Figure 7.* **Evaluation of resolution invariance with different noise covariances** $C$. All models were trained at resolution $M = 256$ and evaluated at varying test resolutions.

| Name | Value |
|---|---|
| Training function $X_1^A$ | $\begin{cases} dY_t = c_A\,Y_t\,dt + g\,dW_t \\ Y_0 \sim \mathcal{N}(m_0, \Sigma_0),\ m_0{=}2.0,\ \Sigma_0{=}0.2 \end{cases}$ |
| Training function $X_1^B$ | $\begin{cases} dY_t = c_B\,Y_t\,dt + g\,dW_t \\ Y_0 \sim \mathcal{N}(m_0, \Sigma_0),\ m_0{=}2.0,\ \Sigma_0{=}0.2 \end{cases}$ |
| Training functions' input time points $M$ | 128 |
| Training functions' output dimension $D$ | $\in \{1,\ 2,\ 3,\ 5\}$ |
| Noise function $X_0$ | Rougher empirical GT Fourier-spectrum |
| Num. modes $N$ summed at KL estimation | 64 |
| $t$ sampling scheme at training | Importance sampling $t/(1-t)$ |
| $t$ sampling scheme at KL estimation | Importance sampling $t/(1-t)$ |
| Num. $t$ sampled at KL estimation | 100 |
| Num. functions at training | 50,000 |
| Training batch size | 1024 |
| Training iterations | 30,000 |
| Num. functions at KL estimation | 500 |
| Optimizer | Adam |
| EMA rate | 0.999 |
| LR | 4.06e−3 |
| LR scheduler | Cosine annealing |
| FFM training loss | $w\,L_{\text{FFM}} + (1-w)\,L_{\text{FKL}},\ w$ linearly decayed from 1 to 0.2 |
| Model | MINO-T |
| Encoder (dim / depth / heads) | 32 / 2 / 8 |
| Decoder (dim / depth / heads) | 32 / 2 / 8 |
| Supernode radius | 5e−4 |
| GPUs for Training | $1 \times$ NVIDIA A100 |

*Table 5.* FKL hyperparameters for SDEs.

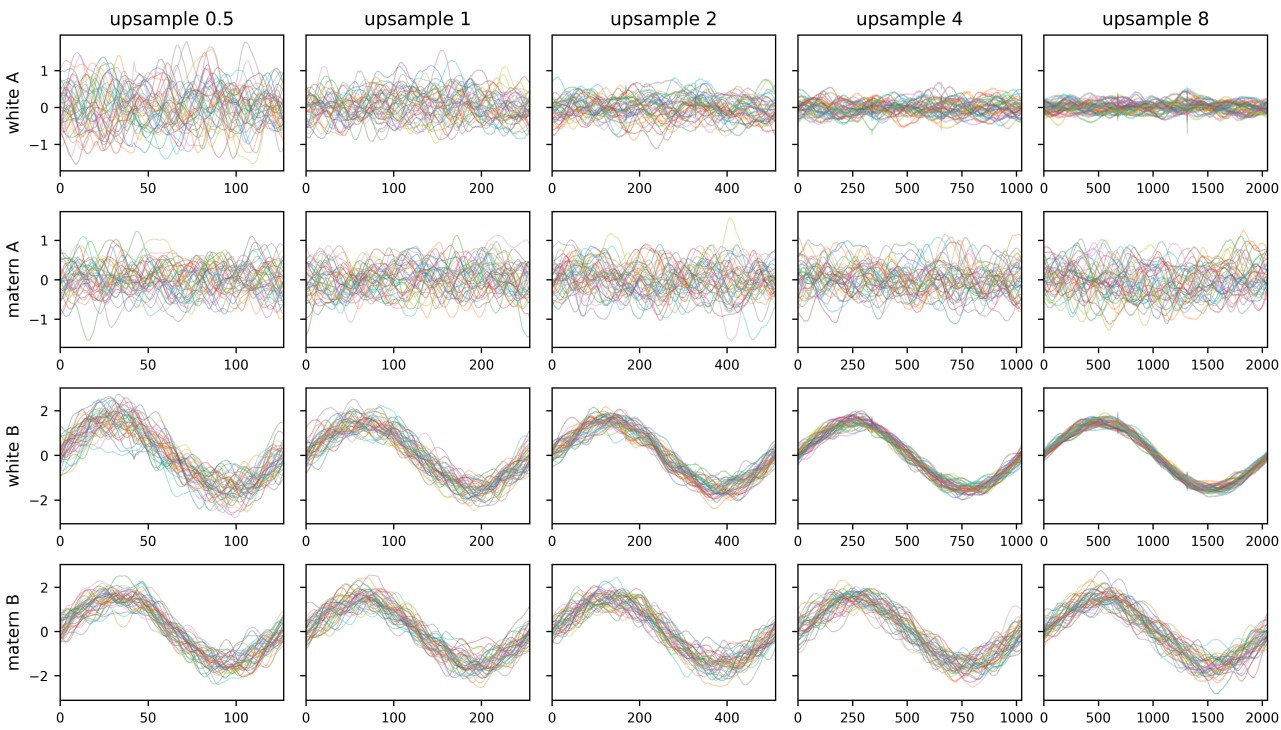

*Figure 8.* Real vs. generated samples across upsample ratios, for the Gaussian measures special case.

## D.2. FKL Ranking Across Diffusion Coefficients

**Problem setup.** We test how the proposed estimator behaves when the distributions $\nu^{\mathcal{A}}$ and $\nu^{\mathcal{B}}$ are generated by SDEs with same drift but different diffusion coefficients $\sigma$, thereby changing the dynamics. We focus on the Lotka-Volterra dataset. Both distributions include 50000 trajectories, sampled on a grid of $M=128$ time points. The reference trajectories are sampled with diffusion $\sigma_{\mathcal{A}}=0.10$, whereas the controlled trajectories are sampled with $\sigma_{\mathcal{B}} \in \{0.00, 0.03, 0.05, 0.10, 0.20, 0.30, 0.40\}$. For every value of $\sigma_{\mathcal{B}}$, we train a conditional MINO-T model for 30k iterations using importance sampling on $t/(1-t)$ during training and estimation, muon optimizer and learning rate $lr = 0.004$. We then compute the forward and reverse FKL using 100 sampled timepoints and 500 trajectory samples. Trajectories are represented in Figure 9.

**Results.** As illustrated in Figure 10, the forward and reverse FKL divergences exhibit a distinct U-shaped profile, reaching their minimum where the measures $\nu^A$ and $\nu^B$ share the same diffusion coefficient of 0.1. This U-shaped behavior aligns with theoretical expectations: the FKL minimizes when the measures share identical support and dynamics, effectively quantifying any discrepancies in the underlying controlled dynamics. As $\sigma_B$ deviates further from the reference $\sigma_A$, the values of forward and reverse FKL correspondingly increase.

## E. Experimental Details for TI Evaluation

### E.1. Marginal Metrics

To quantitatively assess the quality of the generated trajectories and to compare our new evaluation metric, we considered a set of established metrics. These metrics capture the difference in marginal reconstruction on validation data.

**Optimal Transport metrics.** We quantify geometric distances between generated and target distributions using Wasserstein distances (Peyré & Cuturi, 2019). For $p \in \{1, 2\}$, the $p$-Wasserstein distance between probability measures $\mu$ and $\nu$ is

$$W_p(\mu, \nu) = \left( \inf_{\gamma \in \Pi(\mu, \nu)} \int_{\mathbb{R}^d \times \mathbb{R}^d} \|x - y\|^p \, \mathrm{d}\gamma(x, y) \right)^{1/p}, \tag{29}$$

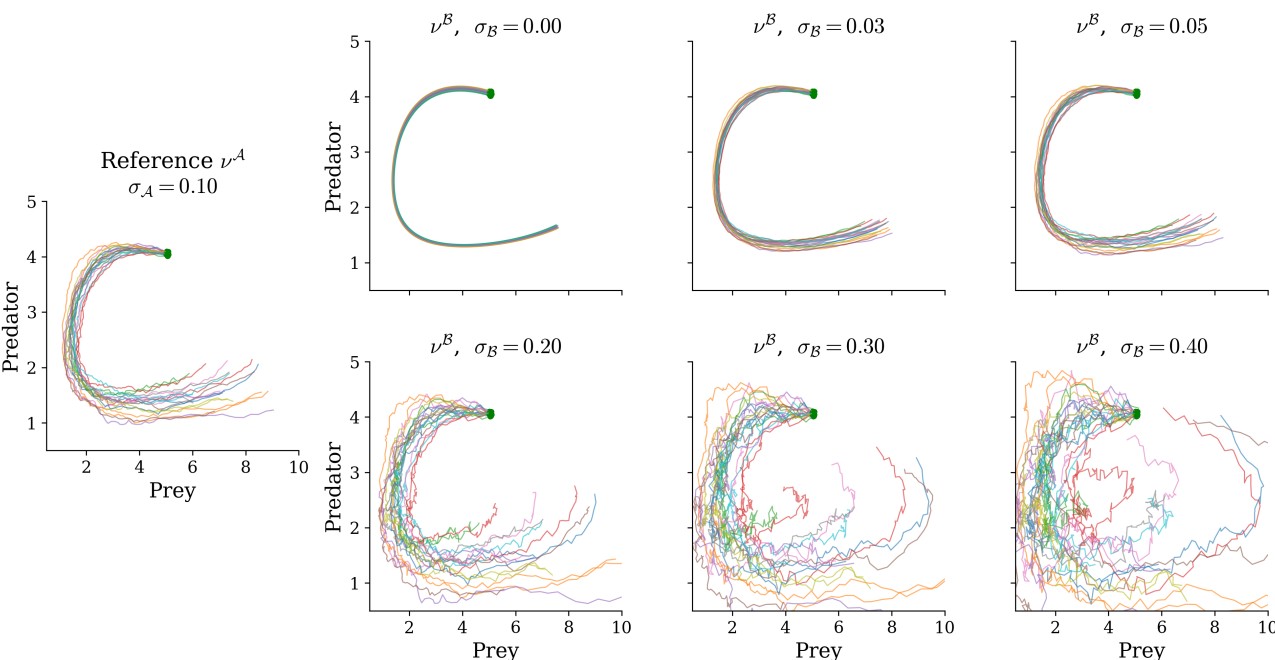

*Figure 9.* Lotka Volterra trajectories, different diffusion coeffient $\sigma_B$.

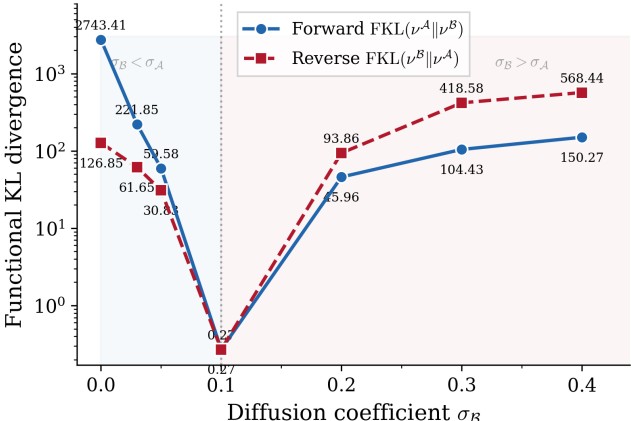

*Figure 10.* Analysis of FKL and distribution support. Forward and Reverse FKL vs diffusion coefficient.

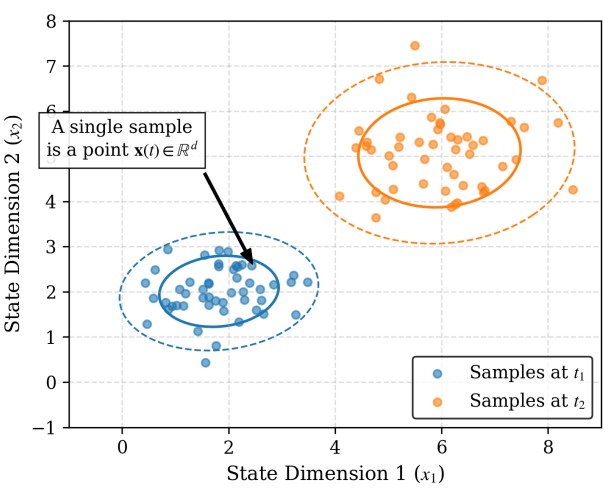
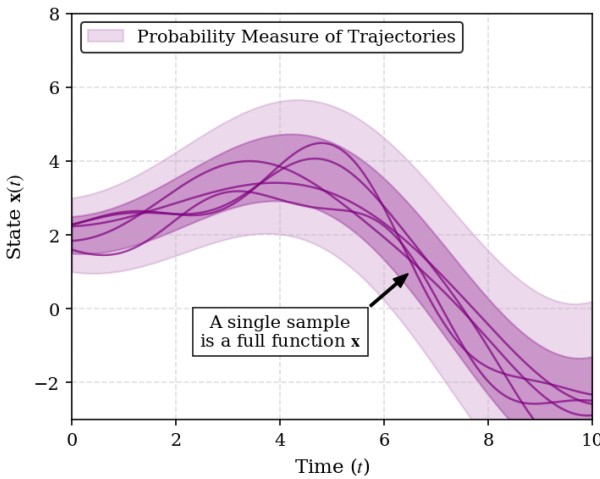

*(a)* Finite-dimensional setting: distributions over states $\mathbf{x}(t) \in \mathbb{R}^d$ at individual time points.

*(b)* Function-space setting: distributions over trajectories $\mathbf{x} : [0,1] \to \mathbb{R}^d$.

*Figure 11.* Conceptual comparison between finite-dimensional and function space modeling.

where $\Pi(\mu, \nu)$ is the set of couplings with marginals $\mu$ and $\nu$. We consider Earth Mover's Distance (EMD, i.e., $W_1$) and Wasserstein-2 ($W_2$). To remain comparable to prior work, we also include the Sliced Wasserstein Distance (SWD):

$$SW_p(\mu, \nu) = \left( \int_{\mathbb{S}^{d-1}} W_p^p(\theta_{\#}\mu, \theta_{\#}\nu) \, \mathrm{d}\lambda(\theta) \right)^{1/p}, \tag{30}$$

which averages 1D Wasserstein distances over random projections over the unit sphere $\theta \in \mathbb{S}^{d-1}$. Similarly, we report the Max-Sliced Wasserstein Distance (MWD).

**Kernel-based metrics.** In addition to OT metrics, we also report the Maximum Mean Discrepancy (MMD), a kernel based two-sample test (Gretton et al., 2012) that detects differences between distributions by comparing their mean embeddings in a Reproducing Kernel Hilbert Space (RKHS). Formally,

$$\mathrm{MMD}^2(\mu, \nu) = \mathbb{E}_{x,x' \sim \mu}[k(x, x')] - 2\mathbb{E}_{x \sim \mu, y \sim \nu}[k(x, y)] + \mathbb{E}_{y,y' \sim \nu}[k(y, y')], \tag{31}$$

where $k$ is a positive definite kernel. For all the experiments, we consider the Radial Basis Function (RBF) kernel with kernel bandwidth $\sigma = 1$.

It is important to note that, such marginal metrics are intrinsically limited because marginals do not determine how states at different times are coupled. Figure 11 highlights this distinction: while marginal evaluation only compares snapshot distributions in finite-dimensional space, the underlying inference target is a probability measure over full trajectories in function space.

### E.2. TI Evaluation on Synthetic Datasets

#### E.2.1. LOTKA-VOLTERRA

**Dataset.** We evaluate the proposed method on the dynamics of a stochastic Lotka-Volterra predator-prey model (Shen et al., 2025). The system describes the evolution of prey ($X_t$) and predator ($Y_t$) populations governed by the following system of stochastic differential equations (SDEs):

$$\begin{aligned}
\mathrm{d}X_t &= (\alpha X_t - \beta X_t Y_t)\mathrm{d}t + \sigma \mathrm{d}W_{x,t}, \\
\mathrm{d}Y_t &= (\gamma X_t Y_t - \delta Y_t)\mathrm{d}t + \sigma \mathrm{d}W_{y,t},
\end{aligned} \tag{32}$$

where $W_t = [W_{x,t}, W_{y,t}]^\top$ denotes a standard 2-dimensional Brownian motion. We define the diffusion coefficient as $\sigma = 0.1$ and fix the model parameters to $\alpha = 1$, $\beta = 0.4$, $\gamma = 0.1$, and $\delta = 0.4$.

To generate the synthetic dataset, we simulate the system over $K = 8$ unit time intervals. The initial states are sampled uniformly such that $X_0 \sim \mathcal{U}(5, 5.1)$ and $Y_0 \sim \mathcal{U}(4, 4.1)$. Numerical integration is performed using the Euler-Maruyama scheme with a discretization step of $\Delta t = 0.02$. Generated trajectories are considered as GT.

**TI methods configuration.** To train the trajectory inference methods, we considered 9 equally spaced snapshots in the trajectory time $[0, 1]$. Odd snapshots are used as training, whereas even snapshots as validation. For each snapshot, we consider 100 points for training TI methods.

Specifications of hyperparameters for TI methods:

- SBIRR-vSB: we run SBIRR and vSB methods using default parameters, but considering the new data. For a fair comparison, the number of iterations for vSB has been set to the same number of SBIRR (i.e. 10).

- MSBM: num_stage $= 20$; num_epoch $= 1$; num_itr $= 1000$; num_ResNet $= 1$; learning_rate $= 1 \times 10^{-3}$; var $= 0.1$; interval $= 101$, BS$= 34$, time_scale $= 8$

- MFL: lambda_reg $= 0.0075$; initial position of the particles (cx, cy) $= (3.0, 2.5)$; n_sinkhorn $= 500$; sigma$=2.0$, sigma_final $= 0.8$; t_final $= 8.0$; eta_final $=0.1$; n_iter $= 2500$; M (number of particles) $= 500$; tau_final $= 1.0$. All the other hyperparameters are set to the default values. Notice that we had to change the initial position of the particles (cx, cy $\neq 0, 0$) is order to make them closer to the correct positions of the marginals.

- AM: T_final $= 8.0$, BS $= 100$; SIGMA $= 0.1$; lr $= 5e$-$5$; num_iterations $= 2\_000$. Moreover, due to the non-overlapping of the training snapshots, we needed to use the "interpolation trick" to make the dynamics continuous.

- TIGON: learning rate $= 5e - 4$; training time points $t \in [0, 2, 4, 6, 8]$; initial gaussian kernel bandwidth $\sigma_{\text{now}} = 1$; decay $= 0.9\sigma$; and a regularization parameter $\lambda_d = 10^4$. All the other parameters have been set to the default values.

**FKL configuration.** We report FKL hyperparameters in Table 6.

| Name | Value |
|---|---:|
| Training function $X_1^A$ | GT |
| Training function $X_1^B$ | TI methods: SBIRR, vSB, MSBM, MFL, AM, TIGON |
| Training functions' input time points $M$ | 401 |
| Training functions' output dimension $D$ | 2 |
| Covariance operator of noise function $X_0$ | Rougher empirical GT Fourier-spectrum |
| Num. modes $N$ summed at KL estimation | 16 |
| $t$ sampling scheme at training | Curriculum: logit-normal (mean=0.8, std=1) for first $40\%$, then uniform |
| $t$ sampling scheme at KL estimation | Importance sampling $t/(1 - t)$ |
| Num. $t$ sampled at KL estimation | 100 |
| Num. functions at training | 500 |
| Training batch size | 32 |
| Training iterations | 20,000 |
| Num. functions at KL estimation | 500 |
| Optimizer | Muon |
| EMA rate | 0.999 |
| LR | 6.21e$-$4 |
| LR scheduler | Cosine annealing |
| FFM training loss | $w\,L_{\text{FFM}} + (1 - w)\,L_{\text{FKL}}$, $w$ linearly decayed from 1 to 0.2 |
| Model | MINO-T |
| Encoder (dim / depth / heads) | 64 / 4 / 4 |
| Decoder (dim / depth / heads) | 64 / 3 / 4 |
| Supernode radius | 0.002 |
| GPUs for Training | $1 \times$ NVIDIA A100 |

*Table 6.* FKL hyperparameters for Lotka-Volterra.

### E.2.2. REPRESSILATOR

**Dataset.**    Repressilator (Shen et al., 2025) is a synthetic genetic regulatory network designed to exhibit stable oscillatory behavior. The system consists of three genes connected in a feedback loop, where each gene expresses a protein that represses the next gene in the cycle. The protein concentrations $X_t = [X_{1,t}, X_{2,t}, X_{3,t}]^\top$ can be modeled using the following system of SDEs:

$$
\begin{aligned}
\mathrm{d}X_{1,t} &= \left( \frac{\beta}{1 + (X_{3,t}/k)^n} - \gamma X_{1,t} \right) \mathrm{d}t + \sigma \mathrm{d}W_{1,t}, \\
\mathrm{d}X_{2,t} &= \left( \frac{\beta}{1 + (X_{1,t}/k)^n} - \gamma X_{2,t} \right) \mathrm{d}t + \sigma \mathrm{d}W_{2,t}, \\
\mathrm{d}X_{3,t} &= \left( \frac{\beta}{1 + (X_{2,t}/k)^n} - \gamma X_{3,t} \right) \mathrm{d}t + \sigma \mathrm{d}W_{3,t},
\end{aligned}
\tag{33}
$$

where $\mathbf{W}_t = [W_{1,t}, W_{2,t}, W_{3,t}]^\top$ denotes a standard 3-dimensional Brownian motion. We set the parameters to $\beta = 10$, $n = 3$, $k = 1$, and degradation rate $\gamma = 1$. We set the diffusion coefficient $\sigma = 0.1$.

We simulate trajectories over 7.5 unit time intervals using the Euler-Maruyama scheme with a step size of $\Delta t = 0.01$. The system is initialized with $X_{1,0}, X_{2,0} \sim \mathcal{U}(1, 1.1)$ and $X_{3,0} \sim \mathcal{U}(2, 2.1)$. Generated trajectories are considered as GT trajectories.

**TI methods configuration.**    To train the trajectory inference methods, we considered 11 equally spaced snapshots in the trajectory time $[0, 1]$. Odd snapshots are used as training, whereas even snapshots as validation. For each snapshot, we consider 100 points for training.

Specifications of hyperparameters for TI methods:

- SBIRR-vSB: As in the Lotka Volterra case, we run SBIRR and vSB methods using default parameters, but considering the new data. Again, for a fair comparison, the number of iterations has been set to the same number (i.e. 10).

- MSBM hyperparameters: num_stage $= 20$; num_epoch $= 5$; num_itr $= 1000$; num_ResNet $= 3$; learning_rate $= 1 \times 10^{-3}$; var $= 0.1$; interval $= 151$, time_scale $= 7.5$, BS=32.

- MFL: lambda_reg $= 0.0075$; initial position of the particles (cx, cy, cz) $= (2.5, 2.5, 2.5)$; n_sinkhorn $= 500$; sigma=1.0, sigma_final $= 0.5$; t_final $= 7.5$; eta_final $=0.1$; n_iter $= 2500$; M (number of particles) $= 500$, tau_final $= 1.0$. All the other hyperparameters are set to the default values. Notice that, also in this case, we had to change the initial position of the particles (cx, cy, cz $\neq 0, 0, 0$).

- AM: T_final $= 7.5$, BS $= 100$; SIGMA $= 0.1$; lr $= 5e$-$6$; num_iterations $= 10\_000$. Even in this case we employ the linear interpolation trick, given that training marginals are located far away in space.

- TIGON: learning rate $= 5e - 4$; training time points $t \in [0, 2, 4, 6, 8, 10]$; initial gaussian kernel bandwidth $\sigma_{\mathrm{now}} = 1$; decay $= 0.9\sigma$ with a stopping condition of $\sigma > 0.02$; and a regularization parameter $\lambda_d = 10^7$. We changed the neural network architecture for the drift, considering 8 hidden layers, each with dimension 32. All the other parameters have been set to the default values.

**FKL configuration.**    We report FKL hyperparameters in Table 7.

### E.2.3. PETAL

**Dataset.**    Petal (Huguet et al., 2022; Neklyudov et al., 2023) is a 2D dataset designed to evaluate the models ability to handle complex branching dynamics. The data mimics a biological differentiation process where trajectories originate from a single source and evolve into distinct lineages. The geometry consists of 8 sinusoidal branches radiating from a central origin, creating a flower-like structure.

The particle dynamics are defined in an intrinsic coordinate system $(u, z)$ relative to a specific branch $k \in \{1, \ldots, 8\}$. The longitudinal position $u_t$ represents the progress along the branch, while the transverse component $z_t$ represents the deviation

| Name | Value |
|---|---|
| Training function $X_1^A$ | GT |
| Training function $X_1^B$ | TI methods: SBIRR, vSB, MSBM, MFL, AM, TIGON |
| Training functions' input time points $M$ | 751 |
| Training functions' output dimension $D$ | 3 |
| Covariance operator of noise function $X_0$ | Rougher empirical AM Fourier-spectrum |
| Num. modes $N$ summed at KL estimation | 16 |
| $t$ sampling scheme at training | Curriculum: logit-normal (mean=0.5, std=1) for first 20%, then uniform |
| $t$ sampling scheme at KL estimation | Importance sampling $t/(1-t)$ |
| Num. $t$ sampled at KL estimation | 100 |
| Num. functions at training | 500 |
| Training batch size | 32 |
| Training iterations | 40,000 |
| Num. functions at KL estimation | 500 |
| Optimizer | Muon |
| EMA rate | 0.999 |
| LR | 3.57e−3 |
| LR scheduler | Cosine annealing |
| FFM training loss | $w\,L_{\text{FFM}} + (1-w)\,L_{\text{FKL}}$, $w$ linearly decayed from 0.2 to 0.04 |
| Model | MINO-T |
| Encoder (dim / depth / heads) | 32 / 2 / 8 |
| Decoder (dim / depth / heads) | 32 / 2 / 8 |
| Supernode radius | 5e−4 |
| GPUs for Training | $1 \times$ NVIDIA A100 |

*Table 7.* FKL hyperparameters for Repressilator.

from the branch spine (thickness). The evolution is governed by the following system:

$$
\begin{aligned}
\mathrm{d}u_t &= v\mathrm{d}t \\
\mathrm{d}z_t &= -\kappa z_t \mathrm{d}t + \sigma_z \mathrm{d}W_t
\end{aligned}
\tag{34}
$$

Here, the longitudinal progress is deterministic with a constant drift velocity $v$ shared by all particles. The transverse dynamics follow an Ornstein-Uhlenbeck process with mean reversion rate $\kappa$, confining particles within a "tube" around the branch line driven by diffusion $\sigma_z$. A deterministic mapping function $\Psi_k(u_t, z_t)$ then projects these coordinates into the 2D Cartesian space based on the sinusoidal geometry of branch $k$.

We define the branches using a reference length $L = 1.0$ and a curvature amplitude $\alpha = 0.25$. The dynamics are configured with a restoring force $\kappa = 0.5$ and transverse diffusion $\sigma_z = 0.04$ (resulting in a stationary tube width of 0.04). The drift velocity is set to $v = 0.2$.

We simulate GT trajectories over the time interval $t \in [0, 4.0]$. The simulation uses a time step of $\Delta t = 0.04$ (100 steps), from which we extract 5 equidistant snapshots for evaluation. Trajectories are initialized as a Gaussian blob centered at the origin ($\sigma_{\text{init}} = 0.1$).

**TI methods configuration.** For trajectory inference methods, we considered the experimental setup of Action Matching (Neklyudov et al., 2023) where, instead of considering held-out marginals, we train the system on all 5 snapshots and evaluate the generated trajectories on the validation points. Given the more complex dynamics due to branching, we consider 2000 points for each training snapshot and 2000 for validation.

Specifications of hyperparameters for the TI methods:

- SBIRR-vSB: for the Petal dataset, we use a custom reference drift `PetalReference` that softly combines the 8

branches. For a state $\mathbf{x}$, we compute for each branch $k$ a spine point $\mathbf{c}_k(\mathbf{x})$ and a unit tangent $\mathbf{t}_k(\mathbf{x})$, and assign weights

$$w_k(\mathbf{x}) = \frac{\exp\left(-\|\mathbf{x} - \mathbf{c}_k(\mathbf{x})\|^2/\tau\right)}{\sum_{j=1}^{8} \exp(-\|\mathbf{x} - \mathbf{c}_j(\mathbf{x})\|^2/\tau)}, \tag{35}$$

with temperature $\tau$ (initialized to 0.01). Let $\bar{\mathbf{c}}(\mathbf{x}) = \sum_k w_k(\mathbf{x})\,\mathbf{c}_k(\mathbf{x})$ and $\bar{\mathbf{t}}(\mathbf{x}) = \sum_k w_k(\mathbf{x})\,\mathbf{t}_k(\mathbf{x})$, and define $\tilde{\mathbf{t}}(\mathbf{x}) = \bar{\mathbf{t}}(\mathbf{x})/(\|\bar{\mathbf{t}}(\mathbf{x})\| + \varepsilon)$. The reference drift is

$$f_{\text{ref}}(\mathbf{x}) = s\,\tilde{\mathbf{t}}(\mathbf{x}) + \lambda\big(\bar{\mathbf{c}}(\mathbf{x}) - \mathbf{x}\big). \tag{36}$$

The diffusion coefficient is set to match the manifold width $\sigma = 0.04$, and the solver discretization $\Delta t = 0.04$ ($N = 25$ steps per snapshot interval). For SBIRR, we use an informative prior that encodes the geometric structure of the data. We initialize the parameters with a tangential speed $s = 0.2$ and a restoring force $\lambda = 0.5$, providing the bridge optimization with a starting process that already respects the flow and the petal structure. For the vSB, we simulate a standard, uninformative Schrödinger Bridge by considering a Brownian motion reference process.

- MSBM: default hyperparameters.

- MFL: t_final = 4.0, lambda_reg = 0.0075; n_sinkhorn = 250; sigma=1.0, sigma_final = 0.35; eta_final =0.1; n_iter = 2500; M (number of particles) = 2000; tau_final = 1.0. All the other hyperparameters are set to the default values.

- AM: T = 4.0, BS = 512; SIGMA = 0.04; lr = 1e-5; num_iterations = 20_000. Given that in this case the training snapshots are overlapping, we followed the same procedure as in the original paper, considering mixture of points to have data which is more dense in time. We do not use any interpolation trick in this case.

- TIGON: learning rate $= 5e - 4$; training time points $t \in [0, 0.25, 0.5, 0.75, 1]$; initial gaussian kernel bandwidth $\sigma_{\text{now}} = 1$; decay $= 0.9\sigma$ with a stopping condition of $\sigma > 0.02$; and a regularization parameter $\lambda_d = 10^7$. We changed the neural network architecture for the drift, considering 8 hidden layers, each with dimension 32. All the other parameters have been set to the default values.

**KL configuration.** We report FKL hyperparameters in Table 8.

| Name | Value |
|---|---|
| Training function $X_1^A$ | GT |
| Training function $X_1^B$ | TI methods: SBIRR, vSB, MSBM, MFL, AM, TIGON |
| Training functions' input time points $M$ | 101 |
| Training functions' output dimension $D$ | 2 |
| Covariance operator of noise function $X_0$ | Rougher empirical GT Fourier-spectrum |
| Num. modes $N$ summed at KL estimation | 16 |
| $t$ sampling scheme at training | Curriculum: logit-normal (mean=0.5, std=1.5) for first 60%, then uniform |
| $t$ sampling scheme at KL estimation | Importance sampling $t/(1-t)$ |
| Num. $t$ sampled at KL estimation | 100 |
| Num. functions at training | 2000 |
| Training batch size | 64 |
| Training iterations | 50,000 |
| Num. functions at KL estimation | 500 |
| Optimizer | Muon |
| EMA rate | 0.999 |
| LR | 1.78e−3 |
| LR scheduler | Cosine annealing |
| FFM training loss | $w\,L_{\text{FFM}} + (1 - w)\,L_{\text{FKL}}$, $w$ linearly decayed from 1 to 0.2 |
| Model | MINO-T |
| Encoder (dim / depth / heads) | 32 / 2 / 8 |
| Decoder (dim / depth / heads) | 32 / 2 / 8 |
| Supernode radius | 5e−4 |
| GPUs for Training | $1 \times$ NVIDIA A100 |

*Table 8.* FKL hyperparameters for Petal.

### E.2.4. CRITICAL DIFFERENCES (CD) DIAGRAM.

We summarize methods performance in terms of marginal metrics using CD (Ismail Fawaz et al., 2019) diagrams based on average ranks. For each task, methods are ranked according to the evaluation score (with ties handled by average ranks), and ranks are averaged across tasks. Statistical differences are assessed via a Friedman test followed by a Wilcoxon-Holm post-hoc comparison: two methods are considered significantly different if their average-rank gap exceeds the CD. In the diagram, methods connected by a horizontal bar are not significantly different at the chosen significance level, whereas unconnected groups indicate statistically distinguishable performance.

We report the diagrams in Figure 5.

### E.3. TI Evaluation on Real-World Datasets

As real-world data we consider two different single-cell RNA sequencing (scRNA-seq) datasets that capture cellular differentiation processes over time: the Embryoid Body (EB) dataset and the Human Embryonic Stem Cell (hESC) dataset, both preprocessed as in (Shen et al., 2025). Both datasets provide snapshots of gene expression profiles at multiple time points during differentiation, making them suitable for evaluating trajectory inference methods.

We consider SBIRR trajectories as reference GT, training the model over all the available snapshots. The other TI methods are trained on odd-index snapshots, and tested on SBIRR validation marginals.

### E.3.1. EMBRYOID BODY

**TI methods configuration.** Specifications of hyperparameters for the TI methods:

- SBIRR: default parameters, trained on all snapshots. Time horizon $\tau \in [0, 1]$.

- vSB: we run vSB methods with default parameters, but we changed the discretization to $dt = 0.01$ and $dts = [0, 0.5, 1]$ in order to be consistent with the other trajectory inference methods.

- MSBM: num_stage $= 11$; num_epoch $= 10$; num_itr $= 1000$; num_ResNet $= 1$; learning_rate $= 2 \times 10^{-4}$; batch_size $= 256$; var $= 0.1$; interval $= 51$.

- MFL: lambda_reg = 0.05; n_sinkhorn = 250; sigma=2.0, sigma_final = 1.0; t_final = 1; eta_final =0.1; n_iter = 1500; M (number of particles) = 1000; tau_final = 1.0. All the other hyperparameters are set to the default values.

- AM hyperparameters: omega = 0.1; BS = 100; SIGMA = 0.1; lr = 1e-6; num_iterations = 10_000. We followed the same procedure as in the original paper, for which we considered mixture of points to have data which is more dense in time. We do not employ any interpolation trick in this case.

- TIGON: training time points $t \in \{0, 0.5, 1\}$; initial kernel bandwidth $\sigma_{\text{now}} = 1$; decay $= 0.5\sigma$ with a stopping condition of $\sigma > 0.02$; and a regularization parameter $\lambda_d = 10^7$. All the other parameters have been set to the default values.

**KL configuration.** We report FKL hyperparameters in Table 9.

### E.3.2. HUMAN EMBRYONIC STEM CELL

**TI methods configuration.** Specifications of hyperparameters for TI methods:

- SBIRR: default parameters, trained on all snapshots. Time horizon $\tau \in [0, 1]$.

- vSB: we run vSB with default parameters. As for the EB dataset, we changed the time horizon in order to be limited in the interval $[0, 1]$, to be consistent with the other methods.

- MSBM hyperparameters: num_stage $= 100$; num_epoch $= 1$; num_itr $= 1000$; num_ResNet $= 1$; learning_rate $= 1 \times 10^{-3}$; batch_size $= 256$; var $= 0.1$; interval $= 30$.

- MFL: lambda_reg = 0.025; n_sinkhorn = 500; sigma=2.0, sigma_final = 1.0; t_final = 1; eta_final =0.1; n_iter = 2500; M (number of particles) = 500; tau_final = 1.0. All the other hyperparameters are set to the default values.

| Name | Value |
|---|---|
| Training function $X_1^A$ | SBIRR |
| Training function $X_1^B$ | TI methods: vSB, MSBM, MFL, AM, TIGON |
| Training functions' input time points $M$ | 101 |
| Training functions' output dimension $D$ | 5 |
| Covariance operator of noise function $X_0$ | Rougher empirical SBIRR Fourier-spectrum |
| Num. modes $N$ summed at KL estimation | 16 |
| $t$ sampling scheme at training | Curriculum: logit-normal (mean=0.5, std=1.5) for first 40%, then uniform |
| $t$ sampling scheme at KL estimation | Importance sampling $t/(1-t)$ |
| Num. $t$ sampled at KL estimation | 100 |
| Num. functions at training | 300 |
| Training batch size | 32 |
| Training iterations | 20,000 |
| Num. functions at KL estimation | 500 |
| Optimizer | Muon |
| EMA rate | 0.999 |
| LR | 1.28e−3 |
| LR scheduler | Cosine annealing |
| FFM training loss | $w\, L_{\text{FFM}} + (1-w)\, L_{\text{FKL}}$, $w$ linearly decayed from 1 to 0.2 |
| Model | MINO-T |
| Encoder (dim / depth / heads) | 32 / 2 / 8 |
| Decoder (dim / depth / heads) | 32 / 2 / 8 |
| Supernode radius | 5e−3 |
| GPUs for Training | $1 \times$ NVIDIA A100 |

*Table 9.* FKL hyperparameters for Embryoid Body.

- AM: BS = 50; SIGMA = 0.1; lr = 1e-6; num_iterations = 20_000; MLP with hidden dimension equal to 256. In this case, we used gradient accumulation. We followed the same procedure as in the original paper, for which we considered mixture of points to have data which is more dense in time. We do not employ any interpolation trick in this case, even if the data present jumps in space between marginals.

- TIGON: training time points $t \in \{0, 0.5, 1\}$; initial kernel bandwidth $\sigma_{\text{now}} = 1$; decay $= 0.5\sigma$ with a stopping condition of $\sigma > 0.02$; and a regularization parameter $\lambda_d = 10^7$. All the other parameters have been set to the default values. In this case, given the different number of cells for each snapshot, we also included the growth term in the model.

**KL configuration.** We report FKL hyperparameters in Table 10.

### E.3.3. MOUSE ERYTHROID

**TI methods configuration.** Specifications of hyperparmeters for the TI methods:

- SBIRR: 300 training points per snapshot; n_epochs = 40; lr = 2e-2.

- vSB: same as SBIRR.

- MSBM: num_stage = 20; num_epoch = 1; num_itr = 1000; num_ResNet = 1; learning_rate = $1 \times 10^{-3}$; batch_size = 256; var = 0.1; interval = 512, time_scale = 1.0.

- MFL: lambda_reg = 0.0075; n_sinkhorn = 250; sigma=2.0, sigma_final = 0.5; t_final = 1.0; eta_final =0.1; n_iter = 10_000; M (number of particles) = 1000; tau_final = 1.0.

- AM: omega = 0.1; BS = 100; SIGMA = 0.1; lr = 1e-5; num_iterations = 50_000.

- TIGON: same as EB dataset, but we changed the neural network architecture for the drift, considering 8 hidden layers, each with dimension 32.

We report the generated trajectories in Figure 6 and the hyperparameters for training FFM in Table 11.

| Name | Value |
|---|---|
| Training function $X_1^A$ | SBIRR |
| Training function $X_1^B$ | TI methods: vSB, MSBM, MFL, AM, TIGON |
| Training functions' input time points $M$ | 121 |
| Training functions' output dimension $D$ | 5 |
| Covariance operator of noise function $X_0$ | Rougher empirical SBIRR Fourier-spectrum |
| Num. modes $N$ summed at KL estimation | 16 |
| $t$ sampling scheme at training | Curriculum: logit-normal (mean=0.5, std=1.5) for first 40%, then uniform |
| $t$ sampling scheme at KL estimation | Importance sampling $t/(1-t)$ |
| Num. $t$ sampled at KL estimation | 100 |
| Num. functions at training | 296 |
| Training batch size | 32 |
| Training iterations | 20,000 |
| Num. functions at KL estimation | 500 |
| Optimizer | Muon |
| EMA rate | 0.999 |
| LR | 1.28e−3 |
| LR scheduler | Cosine annealing |
| FFM training loss | $w\,L_{\text{FFM}} + (1-w)\,L_{\text{FKL}}$, $w$ linearly decayed from 1 to 0.2 |
| Model | MINO-T |
| Encoder (dim / depth / heads) | 32 / 2 / 8 |
| Decoder (dim / depth / heads) | 32 / 2 / 8 |
| Supernode radius | 5e−3 |
| GPUs for Training | $1 \times$ NVIDIA A100 |

*Table 10.* FKL hyperparameters for HESC.

**KL configuration.** We report FKL hyperparameters in Table 11.

### E.3.4. HUMAN FIBROBLAST

**TI methods configuration.** Specifications for the TI methods:

- SBIRR: same as Mouse Erythroid.

- vSB: same as SBIRR.

- MSBM hyperparameters: same as Mouse Erythroid.

- MFL hyperparameters: lambda_reg = 0.05; n_sinkhorn = 250; sigma=2.0, sigma_final = 0.5; t_final = 1.0; eta_final =0.1; n_iter = 10_000; M (number of particles) = 1000; tau_final = 1.0.

- AM hyperparameters: omega = 0.1; BS = 100; SIGMA = 0.1; lr = 1e-5; num_iterations = 50_000.

- TIGON configuration: same as Mouse Erythroid.

We report the generated trajectories in Figure 6 and the hyperparameters for training FFM in Table 12.

**KL configuration.** We report FKL hyperparameters in Table 12.

### E.4. Uncertainty Estimation

#### E.4.1. FKL

To ensure the reliability of our performance rankings and account for the inherent stochasticity in Monte Carlo sampling and velocity fields training, we evaluate all models across multiple independent runs. By employing 3 different random

| Name | Value |
|---|---|
| Training function $X_1^A$ | SBIRR |
| Training function $X_1^B$ | TI methods: vSB, MSBM, MFL, AM, TIGON |
| Training functions' input time points $M$ | 1024 |
| Training functions' output dimension $D$ | 5 |
| Covariance operator of noise function $X_0$ | Rougher empirical SBIRR Fourier-spectrum |
| Num. modes $N$ summed at KL estimation | 16 |
| $t$ sampling scheme at training | Curriculum: logit-normal (mean=0.5, std=1.5) for first 40%, then uniform |
| $t$ sampling scheme at KL estimation | Importance sampling $t/(1-t)$ |
| Num. $t$ sampled at KL estimation | 100 |
| Num. functions at training | 900 |
| Training batch size | 64 |
| Training iterations | 20,000 |
| Num. functions at KL estimation | 500 |
| Optimizer | Muon |
| EMA rate | 0.999 |
| LR | 1.28e−3 |
| LR scheduler | Cosine annealing |
| FFM training loss | $w\,L_{\text{FFM}} + (1-w)\,L_{\text{FKL}}$, $w$ linearly decayed from 1 to 0.2 |
| Model | MINO-T |
| Encoder (dim / depth / heads) | 32 / 2 / 8 |
| Decoder (dim / depth / heads) | 32 / 2 / 8 |
| Supernode radius | 2e−4 |
| GPUs for Training | $1 \times$ NVIDIA A100 |

*Table 11.* FKL hyperparameters for Mouse Erythroid.

| Name | Value |
|---|---|
| Training function $X_1^A$ | SBIRR |
| Training function $X_1^B$ | TI methods: vSB, MSBM, MFL, AM, TIGON |
| Training functions' input time points $M$ | 1024 |
| Training functions' output dimension $D$ | 5 |
| Covariance operator of noise function $X_0$ | Rougher empirical SBIRR Fourier-spectrum |
| Num. modes $N$ summed at KL estimation | 16 |
| $t$ sampling scheme at training | Curriculum: logit-normal (mean=0.5, std=1.5) for first 40%, then uniform |
| $t$ sampling scheme at KL estimation | Importance sampling $t/(1-t)$ |
| Num. $t$ sampled at KL estimation | 100 |
| Num. functions at training | 1000 |
| Training batch size | 64 |
| Training iterations | 20,000 |
| Num. functions at KL estimation | 500 |
| Optimizer | Muon |
| EMA rate | 0.999 |
| LR | 1.28e−3 |
| LR scheduler | Cosine annealing |
| FFM training loss | $w\,L_{\text{FFM}} + (1-w)\,L_{\text{FKL}}$, $w$ linearly decayed from 1 to 0.2 |
| Model | MINO-T |
| Encoder (dim / depth / heads) | 32 / 2 / 8 |
| Decoder (dim / depth / heads) | 32 / 2 / 8 |
| Supernode radius | 2e−4 |
| GPUs for Training | $1 \times$ NVIDIA A100 |

*Table 12.* FKL hyperparameters for Human Fibroblast.

| Models | LV | | Repr | | Petal | |
|---|---|---|---|---|---|---|
| | $\mathrm{KL}(\nu^A\|\nu^B)$ | $\mathrm{KL}(\nu^B\|\nu^A)$ | $\mathrm{KL}(\nu^A\|\nu^B)$ | $\mathrm{KL}(\nu^B\|\nu^A)$ | $\mathrm{KL}(\nu^A\|\nu^B)$ | $\mathrm{KL}(\nu^B\|\nu^A)$ |
| Val | $0.271 \pm 0.008$ | $0.268 \pm 0.009$ | $0.015 \pm 0.001$ | $0.014 \pm 0.001$ | $0.079 \pm 0.006$ | $0.078 \pm 0.005$ |
| SBIRR | $\mathbf{43.352 \pm 1.456}$ | $\mathbf{42.779 \pm 0.629}$ | $\mathbf{23.519 \pm 1.087}$ | $\mathbf{25.242 \pm 0.314}$ | $15.991 \pm 0.892$ | $49.360 \pm 2.334$ |
| vSB | $165.057 \pm 8.938$ | $126.886 \pm 5.601$ | $82.933 \pm 2.302$ | $79.014 \pm 0.874$ | $18.881 \pm 0.658$ | $53.435 \pm 4.186$ |
| MSBM | $79.872 \pm 2.644$ | $46.023 \pm 1.504$ | $90.011 \pm 4.888$ | $49.395 \pm 1.399$ | $\mathbf{9.641 \pm 0.307}$ | $\mathbf{17.055 \pm 1.186}$ |
| MFL | $43.929 \pm 2.094$ | $130.579 \pm 13.905$ | $63.077 \pm 2.686$ | $84.621 \pm 4.619$ | $42.660 \pm 0.957$ | $68.191 \pm 3.010$ |
| AM | $44.914 \pm 2.233$ | $55.488 \pm 3.059$ | $66.901 \pm 0.437$ | $126.248 \pm 8.847$ | $12.328 \pm 0.387$ | $31.641 \pm 2.890$ |
| TIGON | $179.367 \pm 2.205$ | $65.442 \pm 5.152$ | $54.515 \pm 3.738$ | $42.844 \pm 1.857$ | $96.144 \pm 11.848$ | $35.505 \pm 3.089$ |

*Table 13.* Variation in FKL across 3 seeds on synthetic datasets.

| Models | EB | | hESC | | ME | | HF | |
|---|---|---|---|---|---|---|---|---|
| | $\mathrm{KL}(\nu^A\|\nu^B)$ | $\mathrm{KL}(\nu^B\|\nu^A)$ | $\mathrm{KL}(\nu^A\|\nu^B)$ | $\mathrm{KL}(\nu^B\|\nu^A)$ | $\mathrm{KL}(\nu^A\|\nu^B)$ | $\mathrm{KL}(\nu^B\|\nu^A)$ | $\mathrm{KL}(\nu^A\|\nu^B)$ | $\mathrm{KL}(\nu^B\|\nu^A)$ |
| vSB | $23.778 \pm 0.982$ | $27.727 \pm 0.392$ | $127.241 \pm 3.511$ | $124.057 \pm 3.868$ | $51.119 \pm 1.330$ | $48.054 \pm 3.561$ | $56.990 \pm 5.092$ | $41.638 \pm 0.481$ |
| MSBM | $29.452 \pm 0.757$ | $\mathbf{21.454 \pm 0.878}$ | $111.151 \pm 3.632$ | $\mathbf{81.697 \pm 1.934}$ | $65.571 \pm 2.524$ | $\mathbf{37.563 \pm 3.349}$ | $58.914 \pm 2.043$ | $\mathbf{26.784 \pm 0.437}$ |
| MFL | $\mathbf{22.058 \pm 0.838}$ | $73.201 \pm 2.341$ | $\mathbf{97.134 \pm 2.747}$ | $117.901 \pm 3.346$ | $\mathbf{42.268 \pm 2.123}$ | $79.306 \pm 4.288$ | $\mathbf{29.706 \pm 0.731}$ | $69.830 \pm 7.687$ |
| AM | $74.803 \pm 5.319$ | $32.180 \pm 0.480$ | $145.552 \pm 18.465$ | $283.293 \pm 32.506$ | $89.223 \pm 3.304$ | $81.838 \pm 5.794$ | $73.227 \pm 6.770$ | $66.942 \pm 4.031$ |
| TIGON | $122.486 \pm 4.452$ | $41.076 \pm 1.958$ | $293.102 \pm 4.104$ | $161.731 \pm 5.677$ | $250.619 \pm 29.298$ | $82.386 \pm 7.848$ | $197.769 \pm 4.872$ | $69.540 \pm 5.794$ |

*Table 14.* Variation in FKL across 3 seeds on real-world datasets.

seeds, we provide quantitative uncertainty estimates for the forward and backward FKL, across the three synthetic datasets (Table 13) and four real-world datasets (Table 14).

The reported performance metrics exhibit high consistency across multiple independent trials. The variance observed between different random seeds suggests that the model is robust to stochastic initialization and training noise, ensuring the reproducibility of our findings.

### E.4.2. MARGINAL METRICS.

In Tables 15 to 17 we estimate the uncertainty of marginal metrics via bootstrapping, on the three synthetic datasets. For Lotka-Volterraand Repressilator we considered 10 runs with a subsample size of 100, whereas for Petal we consider 10 runs and subsample size equal to 500.

The low variance present in most of the results show robustness in Monte Carlo sampling of the data. Notably, in the case of AM on the Repressilator dataset (Table 16), we observe that the variance scales positively with $\tau$. This behavior is expected, as larger values of $\tau$ correspond to trajectories that propagate further into the state space, naturally leading to a higher dispersion of samples and a subsequent increase in the system's variance.

| $\tau$ | Metric | VAL | SBIRR | vSB | MSBM | MFL | AM | TIGON |
|---|---|---|---|---|---|---|---|---|
| 0.125 | $EMD$ | $0.041 \pm 0.005$ | $\mathbf{0.182 \pm 0.016}$ | $1.007 \pm 0.018$ | $0.786 \pm 0.014$ | $0.997 \pm 0.051$ | $0.861 \pm 0.017$ | $0.434 \pm 0.025$ |
| | $W_2$ | $0.054 \pm 0.008$ | $\mathbf{0.191 \pm 0.015}$ | $1.015 \pm 0.033$ | $0.788 \pm 0.014$ | $1.131 \pm 0.071$ | $0.864 \pm 0.016$ | $0.481 \pm 0.024$ |
| | SWD | $0.029 \pm 0.007$ | $\mathbf{0.129 \pm 0.012}$ | $0.758 \pm 0.025$ | $0.596 \pm 0.011$ | $0.794 \pm 0.051$ | $0.654 \pm 0.012$ | $0.328 \pm 0.016$ |
| | MWD | $0.038 \pm 0.009$ | $\mathbf{0.182 \pm 0.016}$ | $1.009 \pm 0.027$ | $0.786 \pm 0.015$ | $0.897 \pm 0.073$ | $0.862 \pm 0.016$ | $0.375 \pm 0.030$ |
| | MMD | $0.019 \pm 0.008$ | $\mathbf{0.175 \pm 0.016}$ | $0.874 \pm 0.007$ | $0.717 \pm 0.011$ | $0.617 \pm 0.019$ | $0.762 \pm 0.012$ | $0.287 \pm 0.021$ |
| 0.375 | $EMD$ | $0.058 \pm 0.007$ | $\mathbf{0.098 \pm 0.006}$ | $0.522 \pm 0.023$ | $0.322 \pm 0.020$ | $0.411 \pm 0.048$ | $0.329 \pm 0.029$ | $0.321 \pm 0.032$ |
| | $W_2$ | $0.073 \pm 0.006$ | $\mathbf{0.117 \pm 0.009}$ | $0.545 \pm 0.066$ | $0.331 \pm 0.019$ | $0.696 \pm 0.105$ | $0.373 \pm 0.057$ | $0.375 \pm 0.029$ |
| | SWD | $0.037 \pm 0.005$ | $\mathbf{0.069 \pm 0.006}$ | $0.363 \pm 0.052$ | $0.235 \pm 0.015$ | $0.498 \pm 0.080$ | $0.272 \pm 0.044$ | $0.246 \pm 0.019$ |
| | MWD | $0.046 \pm 0.006$ | $\mathbf{0.080 \pm 0.009}$ | $0.515 \pm 0.027$ | $0.326 \pm 0.019$ | $0.645 \pm 0.093$ | $0.347 \pm 0.058$ | $0.345 \pm 0.032$ |
| | MMD | $0.022 \pm 0.012$ | $\mathbf{0.049 \pm 0.009}$ | $0.478 \pm 0.011$ | $0.305 \pm 0.018$ | $0.245 \pm 0.019$ | $0.273 \pm 0.023$ | $0.170 \pm 0.025$ |
| 0.625 | $EMD$ | $0.090 \pm 0.009$ | $\mathbf{0.248 \pm 0.028}$ | $0.311 \pm 0.020$ | $0.490 \pm 0.051$ | $0.421 \pm 0.061$ | $0.544 \pm 0.074$ | $0.270 \pm 0.035$ |
| | $W_2$ | $0.113 \pm 0.014$ | $\mathbf{0.276 \pm 0.027}$ | $0.360 \pm 0.056$ | $0.511 \pm 0.051$ | $0.605 \pm 0.091$ | $0.980 \pm 0.234$ | $0.361 \pm 0.036$ |
| | SWD | $0.063 \pm 0.012$ | $\mathbf{0.194 \pm 0.022}$ | $0.248 \pm 0.042$ | $0.366 \pm 0.038$ | $0.415 \pm 0.065$ | $0.727 \pm 0.176$ | $0.238 \pm 0.026$ |
| | MWD | $0.083 \pm 0.018$ | $\mathbf{0.250 \pm 0.031}$ | $0.304 \pm 0.063$ | $0.506 \pm 0.052$ | $0.517 \pm 0.078$ | $0.969 \pm 0.236$ | $0.311 \pm 0.036$ |
| | MMD | $0.039 \pm 0.012$ | $0.189 \pm 0.036$ | $0.183 \pm 0.016$ | $0.429 \pm 0.040$ | $0.271 \pm 0.037$ | $0.252 \pm 0.032$ | $\mathbf{0.136 \pm 0.028}$ |
| 0.875 | $EMD$ | $0.181 \pm 0.025$ | $0.445 \pm 0.100$ | $0.293 \pm 0.046$ | $0.575 \pm 0.077$ | $0.639 \pm 0.080$ | $1.136 \pm 0.146$ | $\mathbf{0.289 \pm 0.053}$ |
| | $W_2$ | $0.221 \pm 0.028$ | $0.514 \pm 0.102$ | $\mathbf{0.349 \pm 0.052}$ | $0.665 \pm 0.078$ | $0.924 \pm 0.105$ | $1.915 \pm 0.284$ | $0.380 \pm 0.066$ |
| | SWD | $0.133 \pm 0.022$ | $0.345 \pm 0.079$ | $\mathbf{0.221 \pm 0.040}$ | $0.470 \pm 0.057$ | $0.664 \pm 0.079$ | $1.412 \pm 0.213$ | $0.254 \pm 0.053$ |
| | MWD | $0.178 \pm 0.032$ | $0.472 \pm 0.110$ | $\mathbf{0.273 \pm 0.065}$ | $0.646 \pm 0.080$ | $0.837 \pm 0.113$ | $1.898 \pm 0.284$ | $0.331 \pm 0.072$ |
| | MMD | $0.081 \pm 0.025$ | $0.234 \pm 0.054$ | $0.169 \pm 0.030$ | $0.396 \pm 0.045$ | $0.240 \pm 0.021$ | $0.227 \pm 0.031$ | $\mathbf{0.127 \pm 0.041}$ |

*Table 15.* Distances from GT Lotka-Volterra trajectories to other methods at each snapshot $\tau$ (mean±std over resampling runs).

| $\tau$ | Metric | VAL | SBIRR | vSB | MSBM | MFL | AM | TIGON |
|---|---|---|---|---|---|---|---|---|
| 0.1 | $EMD$ | $0.063 \pm 0.003$ | $\mathbf{0.392 \pm 0.006}$ | $1.882 \pm 0.006$ | $1.467 \pm 0.014$ | $1.795 \pm 0.098$ | $1.456 \pm 0.016$ | $1.084 \pm 0.041$ |
| | $W_2$ | $0.073 \pm 0.004$ | $\mathbf{0.416 \pm 0.007}$ | $1.887 \pm 0.006$ | $1.470 \pm 0.014$ | $1.915 \pm 0.114$ | $1.459 \pm 0.015$ | $1.147 \pm 0.043$ |
| | SWD | $0.026 \pm 0.003$ | $\mathbf{0.211 \pm 0.004}$ | $1.020 \pm 0.003$ | $0.803 \pm 0.007$ | $1.116 \pm 0.074$ | $0.799 \pm 0.009$ | $0.617 \pm 0.025$ |
| | MWD | $0.040 \pm 0.004$ | $\mathbf{0.395 \pm 0.005}$ | $1.811 \pm 0.006$ | $1.384 \pm 0.014$ | $1.515 \pm 0.085$ | $1.375 \pm 0.016$ | $0.887 \pm 0.031$ |
| | MMD | $0.024 \pm 0.008$ | $\mathbf{0.367 \pm 0.005}$ | $1.268 \pm 0.002$ | $1.120 \pm 0.005$ | $0.931 \pm 0.019$ | $1.111 \pm 0.009$ | $0.690 \pm 0.016$ |
| 0.3 | $EMD$ | $0.118 \pm 0.013$ | $\mathbf{0.856 \pm 0.037}$ | $1.234 \pm 0.019$ | $1.336 \pm 0.023$ | $1.734 \pm 0.060$ | $1.364 \pm 0.040$ | $0.871 \pm 0.060$ |
| | $W_2$ | $0.143 \pm 0.020$ | $\mathbf{0.893 \pm 0.039}$ | $1.258 \pm 0.018$ | $1.366 \pm 0.022$ | $1.845 \pm 0.066$ | $1.402 \pm 0.045$ | $0.974 \pm 0.069$ |
| | SWD | $0.063 \pm 0.015$ | $\mathbf{0.522 \pm 0.023}$ | $0.747 \pm 0.011$ | $0.790 \pm 0.014$ | $1.031 \pm 0.038$ | $0.804 \pm 0.029$ | $0.542 \pm 0.043$ |
| | MWD | $0.100 \pm 0.026$ | $0.870 \pm 0.043$ | $1.135 \pm 0.025$ | $1.315 \pm 0.027$ | $1.519 \pm 0.064$ | $1.307 \pm 0.049$ | $\mathbf{0.770 \pm 0.078}$ |
| | MMD | $0.047 \pm 0.024$ | $0.677 \pm 0.031$ | $0.880 \pm 0.013$ | $0.982 \pm 0.013$ | $0.866 \pm 0.017$ | $0.959 \pm 0.023$ | $\mathbf{0.478 \pm 0.030}$ |
| 0.5 | $EMD$ | $0.174 \pm 0.026$ | $\mathbf{0.451 \pm 0.044}$ | $0.955 \pm 0.061$ | $1.014 \pm 0.039$ | $1.455 \pm 0.097$ | $2.253 \pm 0.448$ | $0.829 \pm 0.070$ |
| | $W_2$ | $0.209 \pm 0.029$ | $\mathbf{0.491 \pm 0.055}$ | $0.996 \pm 0.056$ | $1.114 \pm 0.047$ | $1.593 \pm 0.095$ | $5.650 \pm 1.121$ | $0.899 \pm 0.070$ |
| | SWD | $0.102 \pm 0.023$ | $\mathbf{0.263 \pm 0.035}$ | $0.578 \pm 0.036$ | $0.658 \pm 0.029$ | $0.892 \pm 0.056$ | $3.070 \pm 0.646$ | $0.471 \pm 0.046$ |
| | MWD | $0.157 \pm 0.039$ | $\mathbf{0.375 \pm 0.070}$ | $0.935 \pm 0.055$ | $1.050 \pm 0.042$ | $1.316 \pm 0.106$ | $5.375 \pm 1.033$ | $0.730 \pm 0.079$ |
| | MMD | $0.067 \pm 0.025$ | $\mathbf{0.290 \pm 0.027}$ | $0.596 \pm 0.032$ | $0.707 \pm 0.018$ | $0.642 \pm 0.027$ | $0.615 \pm 0.036$ | $0.433 \pm 0.035$ |
| 0.7 | $EMD$ | $0.229 \pm 0.026$ | $\mathbf{0.787 \pm 0.108}$ | $1.090 \pm 0.036$ | $1.019 \pm 0.062$ | $1.380 \pm 0.110$ | $7.523 \pm 1.679$ | $0.818 \pm 0.079$ |
| | $W_2$ | $0.272 \pm 0.027$ | $\mathbf{0.854 \pm 0.118}$ | $1.158 \pm 0.035$ | $1.157 \pm 0.073$ | $1.530 \pm 0.105$ | $10.616 \pm 1.687$ | $0.875 \pm 0.079$ |
| | SWD | $0.125 \pm 0.020$ | $0.480 \pm 0.068$ | $0.622 \pm 0.018$ | $0.642 \pm 0.041$ | $0.861 \pm 0.067$ | $10.616 \pm 1.687$ | $\mathbf{0.453 \pm 0.051}$ |
| | MWD | $0.205 \pm 0.042$ | $0.800 \pm 0.125$ | $0.999 \pm 0.044$ | $1.015 \pm 0.084$ | $1.305 \pm 0.112$ | $18.451 \pm 2.929$ | $\mathbf{0.684 \pm 0.076}$ |
| | MMD | $0.075 \pm 0.021$ | $\mathbf{0.402 \pm 0.043}$ | $0.620 \pm 0.013$ | $0.620 \pm 0.028$ | $0.537 \pm 0.021$ | $0.563 \pm 0.019$ | $0.425 \pm 0.034$ |
| 0.9 | $EMD$ | $0.287 \pm 0.042$ | $1.872 \pm 0.185$ | $0.954 \pm 0.073$ | $1.228 \pm 0.077$ | $1.228 \pm 0.104$ | $17.566 \pm 3.948$ | $\mathbf{0.852 \pm 0.087}$ |
| | $W_2$ | $0.339 \pm 0.047$ | $1.943 \pm 0.189$ | $1.083 \pm 0.070$ | $1.403 \pm 0.089$ | $1.369 \pm 0.103$ | $38.787 \pm 5.272$ | $\mathbf{0.932 \pm 0.096}$ |
| | SWD | $0.169 \pm 0.032$ | $1.132 \pm 0.111$ | $0.625 \pm 0.040$ | $0.781 \pm 0.054$ | $0.715 \pm 0.066$ | $22.178 \pm 3.052$ | $\mathbf{0.502 \pm 0.062}$ |
| | MWD | $0.264 \pm 0.066$ | $1.894 \pm 0.189$ | $0.912 \pm 0.077$ | $1.284 \pm 0.101$ | $1.134 \pm 0.116$ | $38.447 \pm 5.252$ | $\mathbf{0.712 \pm 0.118}$ |
| | MMD | $0.095 \pm 0.027$ | $0.563 \pm 0.054$ | $0.538 \pm 0.033$ | $0.680 \pm 0.025$ | $0.479 \pm 0.026$ | $\mathbf{0.367 \pm 0.022}$ | $0.428 \pm 0.040$ |

*Table 16.* Distances from GT Repressilator trajectories to other methods at each snapshot $\tau$ (mean±std over resampling runs)

### E.5. Multi-reference Evaluation

In the main we reported the results on real scRNA-seq data considering SBIRR as reference: this choice is biologically motivated by Waddington's epigenetic-landscape view of differentiation. At the same time, our framework is not restricted to this choice and can accommodate other reasonable references. We add in this section experiments changing the reference for real data. More precisely, we considered MSBM and MFL as reference.

We report the results in Table 18 and Table 19.

| $\tau$ | Metric | VAL | SBIRR | vSB | MSBM | MFL | AM | TIGON |
|---|---|---|---|---|---|---|---|---|
| 0 | $EMD$ | $0.025 \pm 0.003$ | $0.027 \pm 0.003$ | $\mathbf{0.025 \pm 0.002}$ | $0.026 \pm 0.003$ | $0.203 \pm 0.023$ | $0.026 \pm 0.002$ | $0.093 \pm 0.004$ |
| | $W_2$ | $0.031 \pm 0.002$ | $0.034 \pm 0.003$ | $\mathbf{0.031 \pm 0.002}$ | $0.033 \pm 0.003$ | $0.694 \pm 0.061$ | $0.033 \pm 0.002$ | $0.098 \pm 0.004$ |
| | SWD | $0.013 \pm 0.003$ | $0.016 \pm 0.003$ | $\mathbf{0.013 \pm 0.002}$ | $0.014 \pm 0.002$ | $0.489 \pm 0.047$ | $0.013 \pm 0.002$ | $0.056 \pm 0.003$ |
| | MWD | $0.017 \pm 0.004$ | $0.019 \pm 0.004$ | $\mathbf{0.016 \pm 0.003}$ | $0.018 \pm 0.003$ | $0.541 \pm 0.060$ | $0.017 \pm 0.003$ | $0.068 \pm 0.003$ |
| | MMD | $0.010 \pm 0.006$ | $0.013 \pm 0.004$ | $\mathbf{0.009 \pm 0.004}$ | $0.009 \pm 0.003$ | $0.054 \pm 0.009$ | $0.011 \pm 0.004$ | $0.022 \pm 0.003$ |
| 0.25 | $EMD$ | $0.048 \pm 0.011$ | $0.048 \pm 0.004$ | $\mathbf{0.048 \pm 0.011}$ | $0.059 \pm 0.005$ | $0.292 \pm 0.012$ | $0.111 \pm 0.005$ | $0.123 \pm 0.004$ |
| | $W_2$ | $0.071 \pm 0.019$ | $\mathbf{0.070 \pm 0.007}$ | $0.071 \pm 0.018$ | $0.080 \pm 0.008$ | $0.529 \pm 0.046$ | $0.126 \pm 0.005$ | $0.130 \pm 0.005$ |
| | SWD | $0.034 \pm 0.011$ | $0.033 \pm 0.004$ | $\mathbf{0.031 \pm 0.010}$ | $0.041 \pm 0.006$ | $0.366 \pm 0.034$ | $0.067 \pm 0.004$ | $0.060 \pm 0.003$ |
| | MWD | $0.050 \pm 0.016$ | $0.050 \pm 0.010$ | $\mathbf{0.047 \pm 0.018}$ | $0.058 \pm 0.010$ | $0.407 \pm 0.040$ | $0.086 \pm 0.005$ | $0.083 \pm 0.003$ |
| | MMD | $0.028 \pm 0.013$ | $0.027 \pm 0.004$ | $0.021 \pm 0.012$ | $0.031 \pm 0.009$ | $0.057 \pm 0.004$ | $0.037 \pm 0.009$ | $\mathbf{0.019 \pm 0.012}$ |
| 0.5 | $EMD$ | $0.070 \pm 0.017$ | $\mathbf{0.066 \pm 0.008}$ | $0.071 \pm 0.017$ | $0.090 \pm 0.010$ | $0.291 \pm 0.011$ | $0.176 \pm 0.009$ | $0.170 \pm 0.007$ |
| | $W_2$ | $0.119 \pm 0.028$ | $\mathbf{0.114 \pm 0.012}$ | $0.122 \pm 0.026$ | $0.136 \pm 0.017$ | $0.460 \pm 0.047$ | $0.205 \pm 0.010$ | $0.180 \pm 0.009$ |
| | SWD | $0.060 \pm 0.015$ | $\mathbf{0.056 \pm 0.007}$ | $0.056 \pm 0.016$ | $0.072 \pm 0.012$ | $0.308 \pm 0.033$ | $0.106 \pm 0.008$ | $0.091 \pm 0.005$ |
| | MWD | $0.086 \pm 0.022$ | $0.086 \pm 0.012$ | $\mathbf{0.084 \pm 0.024}$ | $0.101 \pm 0.015$ | $0.348 \pm 0.045$ | $0.143 \pm 0.015$ | $0.121 \pm 0.007$ |
| | MMD | $0.044 \pm 0.017$ | $0.038 \pm 0.008$ | $0.037 \pm 0.018$ | $0.055 \pm 0.013$ | $0.109 \pm 0.005$ | $0.076 \pm 0.015$ | $\mathbf{0.035 \pm 0.013}$ |
| 0.75 | $EMD$ | $0.089 \pm 0.021$ | $\mathbf{0.080 \pm 0.009}$ | $0.084 \pm 0.020$ | $0.118 \pm 0.020$ | $0.196 \pm 0.012$ | $0.196 \pm 0.017$ | $0.147 \pm 0.014$ |
| | $W_2$ | $0.176 \pm 0.031$ | $\mathbf{0.157 \pm 0.017}$ | $0.159 \pm 0.033$ | $0.189 \pm 0.033$ | $0.397 \pm 0.049$ | $0.239 \pm 0.022$ | $0.192 \pm 0.025$ |
| | SWD | $0.087 \pm 0.018$ | $\mathbf{0.076 \pm 0.007}$ | $0.076 \pm 0.019$ | $0.100 \pm 0.022$ | $0.255 \pm 0.036$ | $0.121 \pm 0.015$ | $0.094 \pm 0.011$ |
| | MWD | $0.134 \pm 0.036$ | $\mathbf{0.115 \pm 0.019}$ | $0.117 \pm 0.031$ | $0.142 \pm 0.034$ | $0.301 \pm 0.040$ | $0.170 \pm 0.025$ | $0.137 \pm 0.026$ |
| | MMD | $0.055 \pm 0.020$ | $0.044 \pm 0.009$ | $0.045 \pm 0.019$ | $0.070 \pm 0.018$ | $0.069 \pm 0.007$ | $0.091 \pm 0.021$ | $\mathbf{0.039 \pm 0.015}$ |
| 1 | $EMD$ | $0.103 \pm 0.025$ | $0.088 \pm 0.013$ | $\mathbf{0.087 \pm 0.026}$ | $0.143 \pm 0.032$ | $0.135 \pm 0.022$ | $0.182 \pm 0.025$ | $0.090 \pm 0.022$ |
| | $W_2$ | $0.272 \pm 0.049$ | $0.237 \pm 0.034$ | $\mathbf{0.232 \pm 0.056}$ | $0.289 \pm 0.063$ | $0.422 \pm 0.049$ | $0.300 \pm 0.045$ | $0.241 \pm 0.049$ |
| | SWD | $0.137 \pm 0.026$ | $0.117 \pm 0.015$ | $\mathbf{0.115 \pm 0.028}$ | $0.151 \pm 0.038$ | $0.261 \pm 0.034$ | $0.155 \pm 0.024$ | $0.116 \pm 0.025$ |
| | MWD | $0.227 \pm 0.058$ | $0.196 \pm 0.033$ | $\mathbf{0.193 \pm 0.055}$ | $0.237 \pm 0.062$ | $0.332 \pm 0.044$ | $0.225 \pm 0.036$ | $0.193 \pm 0.046$ |
| | MMD | $0.062 \pm 0.020$ | $0.047 \pm 0.011$ | $0.048 \pm 0.018$ | $0.080 \pm 0.023$ | $\mathbf{0.043 \pm 0.016}$ | $0.095 \pm 0.022$ | $0.047 \pm 0.015$ |

*Table 17.* Distances from GT Petal trajectories to other methods at each snapshot $\tau$ (mean±std over resampling runs).

*Table 18.* Reference: MSBM. Estimation of FKL over 3 seeds, MINO-T (values divided by number of dims).

| Models | EB | | hESC | | ME | | HF | |
|---|---|---|---|---|---|---|---|---|
| | KL$(\nu^A\|\nu^B)$ | KL$(\nu^B\|\nu^A)$ | KL$(\nu^A\|\nu^B)$ | KL$(\nu^B\|\nu^A)$ | KL$(\nu^A\|\nu^B)$ | KL$(\nu^B\|\nu^A)$ | KL$(\nu^A\|\nu^B)$ | KL$(\nu^B\|\nu^A)$ |
| vSB | $17.512 \pm 0.956$ | $\mathbf{23.739 \pm 0.981}$ | $84.116 \pm 3.265$ | $88.905 \pm 3.552$ | $48.074 \pm 1.632$ | $\mathbf{57.059 \pm 2.495}$ | $60.559 \pm 8.194$ | $\mathbf{63.432 \pm 10.192}$ |
| MFL | $\mathbf{6.953 \pm 0.056}$ | $67.359 \pm 4.940$ | $\mathbf{5.547 \pm 0.393}$ | $\mathbf{39.017 \pm 0.559}$ | $\mathbf{10.619 \pm 0.701}$ | $88.158 \pm 5.204$ | $\mathbf{7.674 \pm 0.200}$ | $109.134 \pm 0.679$ |
| AM | $43.950 \pm 0.493$ | $29.795 \pm 0.221$ | $71.038 \pm 9.118$ | $153.293 \pm 2.358$ | $35.251 \pm 2.960$ | $66.027 \pm 0.807$ | $32.381 \pm 1.047$ | $114.165 \pm 6.470$ |
| TIGON | $81.588 \pm 4.501$ | $29.320 \pm 0.714$ | $200.108 \pm 13.899$ | $56.783 \pm 3.982$ | $343.852 \pm 7.354$ | $91.126 \pm 6.188$ | $175.696 \pm 1.165$ | $98.617 \pm 5.035$ |
| SB-IRR | $17.946 \pm 1.127$ | $26.692 \pm 1.339$ | $82.547 \pm 2.144$ | $103.798 \pm 2.158$ | $58.931 \pm 0.859$ | $104.039 \pm 4.359$ | $28.898 \pm 0.010$ | $71.011 \pm 0.111$ |

*Table 19.* Reference: MFL. Estimation of FKL over 3 seeds, MINO-T (values divided by number of dims).

| Models | EB | | hESC | | ME | | HF | |
|---|---|---|---|---|---|---|---|---|
| | KL$(\nu^A\|\nu^B)$ | KL$(\nu^B\|\nu^A)$ | KL$(\nu^A\|\nu^B)$ | KL$(\nu^B\|\nu^A)$ | KL$(\nu^A\|\nu^B)$ | KL$(\nu^B\|\nu^A)$ | KL$(\nu^A\|\nu^B)$ | KL$(\nu^B\|\nu^A)$ |
| vSB | $\mathbf{57.713 \pm 3.204}$ | $16.421 \pm 0.256$ | $111.940 \pm 4.222$ | $64.712 \pm 0.888$ | $66.469 \pm 0.818$ | $28.579 \pm 0.763$ | $124.758 \pm 16.314$ | $8.731 \pm 0.360$ |
| MSBM | $60.120 \pm 0.907$ | $\mathbf{5.458 \pm 0.040}$ | $\mathbf{38.210 \pm 1.106}$ | $\mathbf{6.270 \pm 0.159}$ | $\mathbf{54.617 \pm 1.533}$ | $\mathbf{7.838 \pm 0.184}$ | $72.245 \pm 3.685$ | $\mathbf{4.484 \pm 0.107}$ |
| AM | $150.758 \pm 4.526$ | $17.975 \pm 0.799$ | $89.820 \pm 4.511$ | $59.907 \pm 2.242$ | $90.623 \pm 4.266$ | $37.277 \pm 1.686$ | $105.329 \pm 1.490$ | $38.085 \pm 1.332$ |
| TIGON | $188.422 \pm 6.251$ | $24.308 \pm 0.385$ | $196.647 \pm 3.424$ | $38.173 \pm 0.578$ | $309.645 \pm 13.862$ | $53.703 \pm 3.573$ | $263.589 \pm 2.370$ | $19.846 \pm 0.804$ |
| SB-IRR | $66.442 \pm 2.047$ | $22.463 \pm 0.758$ | $114.122 \pm 4.109$ | $93.741 \pm 0.672$ | $79.151 \pm 5.821$ | $46.969 \pm 1.531$ | $\mathbf{60.049 \pm 2.317}$ | $30.179 \pm 1.132$ |

