# OpenReview forum: "Relative Entropy Estimation in Function Space: Theory and Applications to Trajectory Inference"
_ICML.cc/2026/Conference — ICML 2026 regular_

### Official Review · Reviewer_vrmv · 2026-03-10

**Soundness:** 2
**Presentation:** 3
**Significance:** 2
**Originality:** 2
**Overall Recommendation:** 4
**Confidence:** 3

**Summary:**

Most existing evaluations of trajectory inference use held-out marginals. Motivated by this limitation, the authors propose Functional KL to compare trajectory measures in path space. The simulation results show that rankings under different marginal metrics can be unstable, while the proposed path-level metric appears more consistent with the implicit assumptions of different methods.

**Compliance With Llm Reviewing Policy:**

Affirmed.

**Final Justification:**

The author's rebuttal has addressed my major concerns.

**Key Questions For Authors:**

1. In the real-data setting, there is no ground-truth trajectory distribution, yet the paper uses the SBIRR-induced trajectory measure as a reference. As a result, the reported ranking seems to reflect closeness to SBIRR’s modeling bias rather than closeness to the true biological dynamics. What exactly is the intended application of FKL in real applications then?

2. Have the authors considered multi-reference evaluation or reference sensitivity analysis in real scNRA-seq data? If the ranking changes substantially when the reference changes, then the real-data conclusions may be difficult to interpret.

3.  Does the current FKL theory extend to unnormalized trajectory measures or unbalanced dynamics? If not, then the theoretical scope of the paper may not fully match the empirical benchmarking setup for methods like TIGON or Action Matching.

4.  Can the authors provide a systematic sensitivity or stability analysis of FKL calculation, showing how these approximations affect the final metric values and method rankings?

5. Since the main contribution is a new evaluation criterion and benchmarking perspective, the datasets as well as the methods tested here seem limited. Could the authors expand the coverage, and more importantly, discuss the rigorous rationale to design the comparisons?

**Limitations:**

no, see my questions

**Strengths And Weaknesses:**

## Strengths

-  The paper is generally clearly written and easy to follow.

## Weaknesses

- On real data, there are no ground-truth trajectories, while the paper uses the SBIRR-induced trajectory measure as a proxy reference without analyzing the robustness of this choice. As a result, the ranking seems to measure closeness to SBIRR’s modeling bias rather than closeness to the true biological process. This makes the practical value of the proposed metric in real applications unclear.

- Several compared methods, such as TIGON and Action Matching, are designed for non-conservative or mass-varying dynamics. It is therefore unclear whether the current theory applies to unnormalized dynamics, and this theoretical scope should be discussed more carefully.

- The estimation of FKL relies on additional velocity-field fitting and approximation steps, but the paper does not provide sufficient error analysis, sensitivity analysis, or stability theory for the resulting metric.

- Since the main contribution is a new evaluation metric, the empirical coverage of methods/datasets used seems quite limited now. The number of compared methods, datasets, and experimental settings is relatively small, so the results currently read more like a few illustrative cases of metric mismatch than a broadly supported benchmarking conclusion.

---

> ### Author Rebuttal · Authors · 2026-03-31
>
> We thank the reviewer for the constructive feedback and address the concerns below.
>
> **Q1: Intended Use of FKL on Real Data**
>
> We agree FKL is reference-dependent. On real data, it measures agreement w.r.t. a chosen reference path measure, which in our experiments is SBIRR. We do not treat SBIRR as biological ground truth. Rather, since SBIRR uses gradient fields motivated by Waddington’s analogy of cellular differentiation as a marble rolling down a potential landscape, we use it as a biologically motivated reference dynamics. Thus, FKL assesses how well a TI method agrees with a biologically interpretable view of developmental dynamics.
>
> **W1 & Q2: Reference Sensitivity Analysis on Real Data**
>
> We use SBIRR as the reference because its dynamics are biologically motivated and align with Waddington’s epigenetic landscape. For this reason, we did not include other references in the real-data experiments. We will clarify that the real-data FKL results should be interpreted relative to this reference. At the same time, multi-reference evaluation is entirely feasible, since the FKL estimator is not restricted to this choice and can accommodate other reasonable references. In the revised version, we will expand the evaluation to include multiple references.
>
> **W2 & Q3: Applicability to Unnormalized Dynamics**
>
> Thank you for raising this point. Our current FKL theory is restricted to balanced, mass-conserving dynamics and does not directly cover unbalanced settings. Extending the analysis to mass-varying dynamics would require a distinct theoretical formulation[1], which is beyond the scope of this work but is an interesting direction for future research. In our experiments, we use the mass-conserving variants of TIGON and Action Matching, so the empirical setup is consistent with the theoretical scope. We will make this limitation and scope clearer in the revision.
>
> **W3 & Q4: Sensitivity of FKL Estimate**
>
> We refer the reviewer to our response to Reviewer 9caM, "Q1: Sensitivity of FKL Estimate".
>
> **W4 & Q5: Benchmark Scope and Design Rationale**
>
> We agree the benchmark should be representative and well motivated. Below, we clarify our choices and add results on two more real datasets.
>
> (a) Methods. Our benchmark follows a principled categorization of TI methods by dynamical formulation: deterministic flow-based and stochastic process-based methods, with Schrödinger Bridge (SB) methods as a major subclass of the latter. We select recent representative methods from each family. For SB-based methods, we include vSB, SBIRR, MSBM, and MFL. While vSB, SBIRR, and MSBM learn forward/backward diffusion dynamics under multiple marginal constraints, MFL formulates SB in path space via a mean-field Langevin approach. We also include entropic AM as a non-SB stochastic method and TIGON as a deterministic method.
>
> (b) Datasets. We cover both (i) synthetic data, where the underlying SDEs are known and FKL can be evaluated against ground truth, and (ii) real scRNA-seq data, where only destructive snapshots are available, matching the biological setting motivating our work. We use three synthetic datasets (Lotka-Volterra, Repressilator, Petal) spanning diverse dynamical patterns and dimensions, and two real scRNA-seq datasets (EB, hESC). Together, they enable balanced evaluation of fidelity to known dynamics and real-world applicability.
>
> (c) Two additional real datasets. We further include two real scRNA-seq datasets: Mouse Erythroid (ME) [2] and Human Fibroblast (HF) [3]. Using the same preprocessing and evaluation pipeline as in the main experiments, we consider a 5-dim state space and 5 temporal snapshots (odd-indexed for training, even-indexed for marginal evaluation). Trajectories are shown in Fig. 4 of [4].
>
> Table 2 of [4] shows that marginal metrics produce substantially different rankings depending on the validation timepoint and the metric choice, indicating that they may be unstable for evaluating TI methods on real scRNA-seq datasets. By contrast, our path-space FKL yields rankings that align with method behavior: forward KL favors mode coverage and ranks MFL best on both datasets, while reverse KL favors mode seeking and ranks MSBM best on both datasets. On the ME dataset, although vSB is sometimes ranked best by marginal metrics, it is outperformed by MFL in forward FKL and by MSBM in reverse FKL. Overall, these results support that our path-space FKL metrics provide a reasonable evaluation of TI methods on real scRNA-seq datasets, consistent with the benchmark results in our submission.
>
> [1] Baradat et al. Regularized unbalanced optimal transport as entropy minimization with respect to branching Brownian motion. arXiv:2111.01666, 2021.
>
> [2] Pijuan-Sala et al. A single-cell molecular map of mouse gastrulation and early organogenesis. Nature, 2019.
>
> [3] Riba et al. Cell cycle gene regulation dynamics revealed by rna velocity and deep-learning. Nature, 2022.
>
> [4] https://anonymous.4open.science/r/15118/figtabs.pdf

---

> > ### Author Rebuttal · Reviewer_vrmv · 2026-04-03
> >
> > The authors have addressed my questions and I will increase my score.

---

> > > ### Author Response · Authors · 2026-04-03
> > >
> > > Dear reviewer vrmv,
> > >
> > > Thank you for your positive feedback! We are glad that our revisions have addressed your questions, and we greatly appreciate your increased score! We will reflect these improvements clearly in the final version.
> > >
> > > Best regards,
> > >
> > > Authors

---

### Official Review · Reviewer_9caM · 2026-03-10

**Soundness:** 3
**Presentation:** 3
**Significance:** 4
**Originality:** 4
**Overall Recommendation:** 5
**Confidence:** 3

**Summary:**

This paper studies a gap in trajectory inference evaluation rather than proposing a new TI model. The starting point is that snapshot data only identify time-indexed marginals, not the underlying path measure, so held-out marginal accuracy can miss large differences in inferred dynamics. The paper proposes FKL, an estimator of KL divergence between probability measures on function space, derived using functional flow matching and implemented through learned velocity fields. It then uses this estimator to benchmark TI methods on synthetic systems with known trajectories and on real single-cell datasets, arguing that path-space evaluation gives a more coherent picture than marginal-only metrics.

**Compliance With Llm Reviewing Policy:**

Affirmed.

**Final Justification:**

This paper addresses an important problem in trajectory inference and the contribution is technically strong and broadly useful. I am keeping my recommendation at “accept”.

**Key Questions For Authors:**

1. How sensitive are the FKL rankings to basis truncation, velocity-field estimation, and Monte Carlo approximation? Since these are part of the estimator it would help to know how stable the rankings are in practice.
2. Assumptions 2.1 and 2.2 are central to the theory. How should a reader interpret FKL in other TI settings where these assumptions may hold only approximately?
3. On the real datasets, how much do the conclusions depend on using SBIRR as the reference trajectory distribution? Would a different reasonable reference substantially change the ranking?

**Limitations:**

Mostly yes. The paper could be a bit more explicit about the fact that the real-data evaluation is relative to a chosen reference rather than to known true trajectories, but it does not necessarily detract from the paper.

**Strengths And Weaknesses:**

Strengths: The paper addresses a well-defined problem in current TI evaluation. Many TI methods are built to represent distributions over paths, but they are usually judged using only marginal agreement across time. The paper’s main contribution is valuable because it targets this mismatch directly. It does not just argue that marginal metrics are insufficient; it also provides a concrete path-space criterion that can be estimated and used in practice, which gives the work broader relevance. The technical contribution is also substantive. The paper derives an estimator for KL divergence on function space rather than restricting attention to finite-dimensional marginals. Under the stated assumptions, the derivation leads to a computable quantity through learned velocity fields. This is important because without such a construction, the paper would remain a conceptual critique of current practice rather than an operational alternative. The empirical study is also strong on the synthetic benchmarks and supports the paper’s central claim more directly than a purely real-data evaluation would.

Weaknesses: The theory depends on assumptions that are mathematically clean but fairly strong in practice, including absolute continuity with bounded density ratio and support in the Cameron-Martin space, possibly after deterministic smoothing. These assumptions make the KL well-defined, but as a result it is less clear how literally the estimator should be interpreted when practical TI outputs only approximately satisfy those conditions. Second, the real data experiments are necessarily reference-dependent. On EB and hESC, the paper evaluates relative to SBIRR trajectories used as the reference rather than to true developmental trajectories. That is a reasonable choice, but I think it changes the interpretation of the result since on real data, FKL measures agreement with the chosen reference rather than closeness to ground truth. I think this should be made more explicit in the paper. Overall, though this is quite an impressive and technically sound paper with a clear contribution.

---

> ### Author Rebuttal · Authors · 2026-03-31
>
> We sincerely thank the reviewer for the constructive feedback and for recognizing the significance of our KL-based path-space TI criterion. Below, we address each concern.
>
> **W1 & Q2: Assumptions**
>
> We thank the reviewer for this thoughtful question.
>
> For Assumption 2.1, absolute continuity is essentially the condition under which the KL is finite; otherwise, the KL is infinite. Since our goal is to estimate KL via a novel estimator, we assume that the target quantity is well-defined. The bounded density-ratio condition is a technical assumption used in the derivation of FKL. In practice, it can be approximately enforced by a mild regularization, such as slight smoothing of both data measures toward each other.
>
> For Assumption 2.2, the Cameron-Martin support condition is needed in the FFM framework. In practice, it can be satisfied by pre-processing of the data via a deterministic smoothing map $T: H \rightarrow H_C$.
>
> When the assumptions hold only approximately, presmoothing modifies the original measures. Therefore, the resulting FKL should be interpreted as a well-defined bound on the KL divergence of the original pair. We agree that further studying this bound is an interesting direction for future work.
>
> **W2 & Q3: Reference-Dependent Evaluation**
>
> We thank the reviewer for this insightful comment. We agree that the reference dependence of FKL on real data should be stated more explicitly. On real datasets, FKL measures agreement relative to a chosen reference path measure; in our experiments, we use SBIRR as reference because it provides biologically motivated dynamics consistent with Waddington’s epigenetic-landscape view of differentiation. At the same time, our framework is not restricted to this choice and can accommodate other reasonable references. We will clarify this in the revised paper and expand the evaluation to assess the sensitivity of the conclusions to the choice of reference.
>
> **Q1: Sensitivity of FKL Estimate**
>
> We sincerely thank the reviewer for this positive and helpful suggestion. To assess the sensitivity of FKL, we conduct a systematic study on a Gaussian special case (detailed in Appendix A.1.1) where the analytic KL is available, allowing direct comparison between the estimated and true KL.
>
> (a) Basis truncation. We discretize the Gaussian processes on $M=128$ uniformly spaced time points and implement the method using Fourier representations with 8, 16, 32, and 64 modes. Fig. 3a in [1] shows that the estimation error decreases as more modes are retained.
>
> (b) Velocity-field estimation. In all TI-ranking experiments, although the analytic marginal velocity field is intractable, the FFM conditional flow-matching loss quantifies conditional velocity-field estimation error. Empirically, this loss decreases rapidly and converges to a very small value, indicating accurate velocity estimation.
>
> In the Gaussian special case, the analytic marginal velocity field is available, enabling direct evaluation of the error in estimating the marginal velocity mismatch ove $t \in[0,1]$. As shown in Fig. 3b of [1], this error remains uniformly small across all $t$, and further decreases as $t$ increases, reaching zero at $t=1$. This directly supports accurate velocity-field estimation. Moreover, since the factor $\frac{t}{1-t}$ in the integrand of Eq. (9) increases monotonically with $t$, the velocity mismatch at larger $t$ is weighted more heavily in the integral. Thus, the most influential region in Eq. (9) is where the estimation error is smallest, further supporting overall FKL precision.
>
> (c) Monte Carlo approximation. We vary the number of trajectories used in the Monte Carlo estimate of FKL, from 10 to 2000. Fig. 3c in [1] shows that the estimates remain highly precise and stable across this range.
>
> (d) Temporal discretization. For the integral over $t\in[0,1]$ in Eq. (9), we vary the number $n$ of sampled time points used to approximate the integral. For each $n$, we repeat the experiment with 5 random seeds and report mean and standard deviation. Fig. 3d in [1] shows that the mean estimate remains close to the ground truth across all $n$, indicating consistently good accuracy. The standard deviation decreases markedly as $n$ increases and is negligible at $n=80$, confirming the strong stability of our estimator.
>
> Together, these results show that FKL estimation is numerically stable and support our discretization choices for the integral in the FKL expression. This numerical stability also underlies the robustness of the resulting TI rankings. In our preliminary ranking experiments, variation across random seeds are much smaller than the pairwise differences in estimated FKL between TIs, so the rankings remain unchanged. In the revised manuscript, we will report standard deviations over multiple random seeds for all TI-ranking tables.
>
> [1] https://anonymous.4open.science/r/15118/figtabs.pdf

---

> > ### Author Rebuttal · Reviewer_9caM · 2026-04-02
> >
> > All concerns have been addressed by the authors and I would recommend that this paper be accepted. The authors clarified how the assumptions should be interpreted in practice and explained more clearly the reference-dependent nature of the real-data evaluation. The sensitivity analysis is very much appreciated.

---

> > > ### Author Response · Authors · 2026-04-02
> > >
> > > Dear reviewer 9caM,
> > >
> > > We sincerely thank you for your positive feedback and recommendation for acceptance! We are pleased that our revisions have addressed your concerns. We also appreciate your recognition of our clarifications regarding the practical interpretation of the assumptions, the reference-dependence of the real-data evaluation, and the added sensitivity analysis. These points will be clearly reflected in the final version.
> > >
> > > Best regards,
> > >
> > > Authors

---

### Official Review · Reviewer_Zm1E · 2026-03-11

**Soundness:** 3
**Presentation:** 3
**Significance:** 3
**Originality:** 4
**Overall Recommendation:** 5
**Confidence:** 4

**Summary:**

This paper introduces FKL, a novel estimator for trajectory inference methods. FKL formulates the estimator in function spaces and estimates the divergence between samples from distributions over trajectories, rather than between marginal snapshots. Empirical results support the use of FKL as a criterion for comparing various trajectory inference methods.

**Compliance With Llm Reviewing Policy:**

Affirmed.

**Final Justification:**

I find FKL to be a novel work with a clear and well-motivated objective. As cell-dynamics modeling becomes more popular, many researchers still rely on distributional distances between snapshots. This paper argues that a more appropriate metric is the KL divergence in path space. The authors explain how this metric can be used and demonstrate its effectiveness clearly. Regarding my concerns about function-space modeling and computational issues, the authors addressed them satisfactorily. I believe this paper is worthy of publication.

**Key Questions For Authors:**

See weaknesses.

**Limitations:**

yes

**Strengths And Weaknesses:**

**Strengths**
* The authors propose a metric called FKL to measure the distributional discrepancy between distributions of inferred trajectories.
* The motivation is novel and well justified.
* The paper is clearly written and easy to follow. However, for general audience who is not familiar to function space modeling, it would benefit from a conceptual figure depicting the difference between snapshot distributions in finite-dimensional spaces and distributions over trajectories in function spaces.

**Weaknesses**
* The proposed FKL appears to rely on a function-space generative model (e,g, FFM). However, as the authors cited, there have been many recent function-space generative models. Can those models also be used to estimate FKL? If so, how would the metric change for different choice of function-space generative model? Moreover, in function-space modeling, the choice of the trace-class covariance operator $C$ is important. Would different choices of $C$ affect the evaluation results?
* Evaluating each method with FKL requires training an FFM to simulate trajectories for each method. How much time and computational cost does it take to train these models for each method?

---

> ### Author Rebuttal · Authors · 2026-03-31
>
> We sincerely thank the reviewer for the thoughtful feedback and for recognizing the novelty of FKL and the soundness of its justification. Below, we address each concern.
>
> **W1: Sensitivity to Model and Covariance**
>
>  We thank the reviewer for this insightful question, which highlights two key aspects of our framework: (a) the choice of generative model and (b) the choice of covariance operator $C$.
>
> (a) Choice of function-space generative model. Our derivation of the velocity-only representation for the KL divergence (Eq. (9)) is specialized to ODE-based function-space generative models, namely FFM, because the derivation relies on the weak continuity equation and the link between logarithmic-gradient mismatch and velocity-field mismatch.
>
> That said, the underlying FKL quantity itself is not specific to FFM. In principle, other function-space generative models could also be used to estimate the same quantity. For example, for SDE-based function-space generative models, such as functional diffusion processes [1], the derivation would proceed via Girsanov’s theorem. We agree that this is an interesting direction, but it lies beyond the scope of the current work.
>
> Therefore, the choice of function-space generative model does not change the definition and value of FKL itself; rather, it changes the derivation and implementation used to obtain a computable estimator. In our work, we adopt FFM because it yields a direct and tractable derivation.
>
> (b) Choice of noise covariance operator $C$. In our function-space formulation, the FFM-based FKL estimator is accurate and resolution-invariant only when the noise covariance satisfies two conditions: (i) $C$ is trace-class on the Hilbert space $H$; and (ii) $C$ satisfies the Cameron-Martin support assumption, i.e., the Gaussian noise $\mathcal{N}(0, C)$ must be rougher than the data measure.
>
> To verify this empirically, we conduct an ablation on a Gaussian-measure case (extending the experiment in Appendix A.2): $\nu^A=\mathcal{N}(1.5 \sin (2 \pi x), \mathcal{R})$, $\nu^B=\mathcal{N}(0, \mathcal{R})$, where $\mathcal{R}$ is a Matérn covariance with smoothness $\alpha_1=3.5$. In this case, the analytic FKL is available: $\operatorname{KL}\left(\nu^A \| \nu^B\right) = \operatorname{KL}\left(\nu^B \| \nu^A \right)=32.79$. We test three choices of $C$ and evaluate the FKL estimate on trajectories sampled at resolutions $M \in \\{128,256,512,1024\\}$, while keeping the FFM trained at resolution $M=256$. Results are shown in Fig. 2 of [2].
>
> When $C=\text{Id}$ (white noise), condition (i) fails because $C$ is not trace-class; consequently, the estimator is not function-space resolution-invariant, with the estimated FKL diverging as the inference resolution increases. When $C$ is Matérn with smoothness $\alpha_0=6.0$, condition (i) holds, but condition (ii) fails because the noise is smoother than the data measure ($\alpha_0>\alpha_1$), leading to unstable and divergent FKL estimates across all resolutions. In contrast, Matérn $C$ with $\alpha_0=0.5$ satisfies both conditions, yielding accurate and resolution-invariant forward and reverse FKL estimates, which highlights the importance of choosing a noise covariance that meets both conditions for FKL evaluation.
>
> **W2: Computational Cost**
>
> We thank the reviewer for raising this important practical question. Because the FKL is estimated using only 500 trajectories generated by each TI method, which we found to be sufficient in our sensitivity study, the overall training overhead is low. Specifically, on a single NVIDIA A100 GPU, the training requires only 13.69 seconds per epoch, corresponding to 2.38 hours in total for 625 epochs, with a memory usage of 1110 MB.
>
> **S3: Conceptual Figure for Snapshot vs. Trajectory Distributions**
>
> We thank the reviewer for this valuable suggestion. In the revised manuscript, we will include a conceptual figure illustrating the difference between snapshot distributions in finite-dimensional spaces and trajectory distributions in function spaces, as shown in Fig. 5 of [2].
>
> [1] Franzese et al. Continuous-Time Functional Diffusion Processes. NeurIPS, 2023.
>
> [2] https://anonymous.4open.science/r/15118/figtabs.pdf

---

> > ### Author Rebuttal · Reviewer_Zm1E · 2026-04-01
> >
> > I thank the authors for addressing my questions. I find FKL to be a novel work with a clear and well-motivated objective. The authors have addressed all of my questions, and I believe that, if these clarifications are properly reflected in the final version, the paper would merit acceptance. I am happy to raise my score from 4 -> 5.

---

> > > ### Author Response · Authors · 2026-04-01
> > >
> > > Dear reviewer Zm1E,
> > >
> > > We are grateful for your positive feedback! We are glad that our responses addressed your questions, and we appreciate your recognition of the novelty of FKL and its clear motivation. We also appreciate your reconsideration and updated score. We will reflect these clarifications in the final version.
> > >
> > > Best regards,
> > >
> > > Authors

---

### Official Review · Reviewer_E5yq · 2026-03-12

**Soundness:** 3
**Presentation:** 2
**Significance:** 2
**Originality:** 3
**Overall Recommendation:** 3
**Confidence:** 2

**Summary:**

## Summary:

This work focuses on estimating the KL divergence between processes in function space, which the authors call Functional KL divergence (FKL). Assuming that both the processes have the same initial distribution, the authors adopt a linear interpolation model for the underlying process which is associated with the velocity field $(v_t(x))$ and the push-forward measure $\mu_t$. Using the weak continuity equation and projection on a finite basis, the authors show that KL divergence can be estimated in terms of integral of the norm of the difference of the velocity fields associated with the two path measures.

Further, the article proposes FKL to be an improved metric for ranking trajectory inference methods compared to traditional evaluation metrics. Demonstration of the FKL as a ranking  metric is presented using synthetic and real data on various SOTA algorithms.

**Compliance With Llm Reviewing Policy:**

Affirmed.

**Key Questions For Authors:**

- Can the authors provide details on the current state of the art on estimating KL divergence in function space?

- Is the formulation of equation (9) a new result that the authors have presented for the first time or are there similar existing formulations in literature?

- It is mentioned in the article that the integral of equation (9) is intractable and the authors are approximating it via Monte Carlo sampling. Can the authors comment on the accuracy of the sampled estimator?

- In Section 3, Implementation Details, the authors mention training two neural networks $v_\theta^A$ and $v_\theta^B$, but it is not clear how these NN are trained; here please clarify what forms the labeled data, the architecture of the NN used, and the loss function used. Can the authors provide details on how this is done?

- In Section 3, Estimation, can the authors clarify how they obtain the $v_\theta^A$ and $v_\theta^B$ from interpolated samples. Also can the authors provide more details on how the estimate of the integral of the equation (9) is obtained?

- In the Experiments section the authors mention using the inference by the SBIRR method as a proxy for the ground truth of the scRNAseq data. Is this choice common in the TI literature of the scRNAseq data? Can the authors provide details on what is a commonly used approach in this field when ground truth is not available?

- Can the authors provide insights on how their method could be adapted if the underlying dynamics do not follow the linear interpolation model of equation (1)?

**Limitations:**

The limitations of the method should be discussed clearly

**Strengths And Weaknesses:**

### Strengths:

- Based on KL divergence in function space, a metric is provided to evaluate trajectory inference algorithms.

- Empirical validation of the KL divergence based ranking metric on various trajectory inference methods are presented using synthetic and real data.

### Weaknesses:

- A better comparison of the proposed KL divergence estimation approach with  existing KL divergence estimation methods needs to be presented.

- No detail is provided on the error between the estimate of the KL divergence and its actual value.  The KL divergence is being estimated using (9); apart from the approximation of the true velocity vector field with a NN proxy, in the estimation phase to carry out the integral in (9) it seems sampling is used to get an estimate. How well and the complexity of the sampling carried out needs to be quantitatively discussed.

- Parts of the article need reorganization. For instance, in the Results section the authors present a discussion about the CD metric. However, the results that are being referred there are in the Appendix and not in the main paper. To improve the presentation the CD diagrams should be included in the main text or the discussion should be moved to Appendix.

- The implementation details are not clear in Section 3. More details on the implementations should be provided. Please see questions for more details.

- **Minor Comments:** Some notations are not explained properly and there are multiple typographical errors. Below are some instances:

    - In equation (3) the set $I$ is not defined.

    - There seems to be a typo in the second paragraph of Section 3 in the expression of $r_t^K$.

    - There are typos in the equations below (6).

    - There are typographical errors and omissions in the equations below equation (6)

    - In equation (9) the norm notation $||\cdot||_{C^\frac{1}{2}H}$ is used but the notation has not been explained.

---

> ### Author Rebuttal · Authors · 2026-03-31
>
> We thank the reviewer for the comprehensive feedback and address the concerns below.
>
> **W1 & Q1 & Q2: Comparison to Existing KL Estimators and Novelty**
>
> Eq. (9) is, to the best of our knowledge, the first KL estimator for function-space measures. Closest prior work is the flow-matching-based KL estimators for finite-dimensional distributions [1,2], but they are not appropriate baselines because we study discretization-invariant KL estimation in function space. We clarify the math distinction and practical advantage below.
>
> The finite- and infinite-dim settings differ fundamentally. [1,2] are derived in finite-dim Euclidean space with $X_0\sim\mathcal{N}(0,\mathrm{Id})$, which is unavailable in an infinite-dim Hilbert space $H$ since $\mathrm{Id}$ is not trace-class. Moreover, $H$ has no Lebesgue measure, so densities w.r.t. Lebesgue measure are undefined. Our derivation instead relies on absolute continuity w.r.t. suitable reference measures, making the relevant Radon-Nikodym derivatives well-defined. The continuity equation in infinite dimensions is also available only in weak form via smooth cylindrical test functions. We therefore first derive the result on $K$-dim projections, then take $K\to\infty$ using Doob's martingale convergence theorem.
>
> Our infinite-dim formulation also has a practical advantage: resolution invariance. Unlike finite-dim methods tied to a fixed discretization, our functional formulation and neural operator estimate KL between path-space measures using trajectories on arbitrary grids, without retraining. We provide experimental evidence in Appendix A.2 and will include an extensive empirical study in the revision.
>
> **W2 & Q3 & Q5: Monte Carlo Integration and Accuracy**
>
> (a) Monte Carlo integration. Evaluating Eq. (9) analytically is intractable. Nevertheless, it admits a Monte Carlo approximation: samples from $\mu_t^A$ can be obtained via $X_1\sim\mu^A$, $X_t\sim\mu_t^{\delta_{X_1}}$, where $\mu_t^{\delta_{X_1}}$ denotes the conditional measure at time $t$ given $X_1$. Likewise, the outer time integral can be approximated by sampling $t\sim\mathcal{U}(0,1)$, using $\int_0^1(\cdot) \mathrm{d} t=\mathbb{E}_{t \sim \mathcal{U}(0,1)}[(\cdot)]$. In practice, our sensitivity analysis shows that 500 Monte Carlo samples and 100 sampled time instants are sufficient.
>
> (b) Accuracy of the sampled estimator. We validate our KL estimator in two settings where the analytic KL is available: (i) Gaussian measures and (ii) linear SDEs; derivations are given in Appendix A. Across diverse settings, our estimates closely match the true KL (Fig. 1, Table 1 in [3]), supporting the validity and accuracy of our approach.
>
> **W3 & W5: Organization and Typo**
>
> We will improve the presentation by moving the discussion closer to the plots, correcting typos, and clarifying relevant notation.
>
> **W4 & Q4: Implementation Details**
>
> (a) Labeled training data. For the estimation of $\mathrm{KL}\left(\nu^A \| \nu^B\right)$, training samples drawn from $\nu^A$ and $\nu^B$ are equipped with a label $c \in\\{0,1\\}$ indicating their distribution of origin.
>
> (b) NN architecture. In the experiments reported, $v_\theta$ is parameterized with Fourier neural operator. We will also examine the mesh-informed neural operator in the revision, as it is well suited to functions on arbitrary grids.
>
> (c) Loss function. We use the conditional flow matching loss introduced in FFM; we will provide pseudocode in the appendix.
>
> **Q6: Evaluation Protocol in scRNA-seq TI**
>
> For real scRNA-seq data, ground-truth trajectories are unavailable because measurements are destructive. Evaluation therefore typically uses marginal metrics to assess reconstruction quality on held-out snapshots. As shown in Section 5, rankings based on these marginal metrics vary substantially depending on the validation time point and the choice of metric. Moreover, matching snapshot marginals does not ensure correct dynamics. This motivates FKL, which evaluates path-space dynamics beyond marginals.
>
> FKL necessarily depends on a reference path measure. In our experiments, SBIRR is used to define this reference dynamics because it provides biologically motivated dynamics consistent with Waddington’s epigenetic-landscape view of differentiation.
>
> **Q7: Eq. (1): Linear Interpolation**
>
> We appreciate this question. The time variable in Eq. (1) is an auxiliary generation time for FFM, orthogonal to the physical time of the single-cell dynamics. Accordingly, Eq. (1) should not be interpreted as assuming linear evolution of the process. The linear interpolation between $X_0$ and $X_1$ is only a simple and stable choice for FFM, while the target measure at $X_1$ is induced by the single-cell dynamics and can capture arbitrary nonlinear processes.
>
> [1] Su et al. On Flow Matching KL Divergence. arXiv:2511.05480, 2025.
>
> [2] Butakov et al. FMMI: Flow Matching Mutual Information Estimation. arXiv:2511.08552, 2025.
>
> [3] https://anonymous.4open.science/r/15118/figtabs.pdf

---

> > ### Author Rebuttal · Reviewer_E5yq · 2026-04-02
> >
> > I still have concerns on the applicability of the results to practice; via the integral in Equation 9. Also, the results seem related to Girsanov's theorem extension to infinite dimensional spaces. I will stick with my score.

---

> > > ### Author Response · Authors · 2026-04-03
> > >
> > > We thank the reviewer for the additional questions. We address them below.
> > >
> > > (1) *I still have concerns on the applicability of the results to practice; via the integral in Equation 9.*
> > >
> > > The numerical evaluation of Eq. (9) via Monte Carlo integration does not limit its practical applicability, as the integral can be accurately approximated in practice.
> > >
> > > As discussed in our rebuttal, in settings where the KL divergence is available in closed form, including Gaussian measures and SDEs, the Monte Carlo estimate of the functional KL consistently provides an accurate approximation to the exact value.
> > >
> > > More importantly, beyond these controlled comparisons, our experiments on complex synthetic SDEs as well as real-world scRNA-seq datasets indicate that Eq. (9) is numerically stable and does not present practical difficulties. This is further supported by the sensitivity analysis provided in our response to Reviewer 9caM (“Q1: Sensitivity of FKL Estimate”), where we examine the effects of basis truncation, velocity-field estimation, Monte Carlo approximation, and temporal discretization. Taken together, these results provide consistent evidence that the evaluation of functional KL is numerically accurate and robust in practice.
> > >
> > > (2) *the results seem related to Girsanov's theorem extension to infinite dimensional spaces.*
> > >
> > > We respectfully clarify that our result does not rely on Girsanov’s theorem. Girsanov’s theorem applies to changes of measure for **stochastic** processes, whereas our framework is entirely **deterministic**. Specifically, our method is based on functional flow matching, an ODE in an infinite-dimensional space. Accordingly, the derivation of Eq. (9) relies on the weak form of the continuity equation, which describes the evolution of measures under a deterministic velocity field in function space. Identifying admissible test functions for this weak-form continuity equation and using them to establish the link between velocity-field mismatch and logarithmic-gradient mismatch are among the main technical challenges addressed in our work.
> > >
> > > We hope this clarifies both the practical computability of Eq. (9) and the distinction between our result and Girsanov-type arguments.

---

### Decision · Program_Chairs · 2026-04-30

**Decision:**

Accept (regular)

**Comment:**

The most significant contribution is a well-motivated framework for estimating KL divergence between probability measures in function space, which addresses a recognized gap in trajectory inference evaluation. Three of four reviewers found the derivation via functional flow matching and the weak continuity equation to be technically substantive and original, and the synthetic experiments on Gaussian measures and linear SDEs provide convincing validation of the estimator's accuracy and stability across basis truncation, Monte Carlo sample size, and temporal discretization choices.

Authors are expected in the final version to include the conceptual figure distinguishing snapshot distributions from path-space distributions, make the reference-dependence of the real-data evaluation explicit and expand to multiple references for scRNA-seq experiments, clarify that the theory applies only to balanced mass-conserving dynamics and does not cover methods like TIGON or Action Matching in their unbalanced variants, and clearly position the contribution relative to finite-dimensional flow-matching KL estimators; one reviewer maintained a weak reject citing residual concerns about practical applicability of the Monte Carlo integral and the relationship to Girsanov-type arguments, but these concerns are outweighed by the majority positive assessment.